# Oxytocin facilitates social behavior of female rats via selective modulation of interneurons in the medial prefrontal cortex

Stephanie Schimmer[1,2,24], Alan Kania [1,2,24], Arthur Lefevre [1,2,3,24],
Konstantinos Afordakos [1,2], Kai-Yi Wang[4], Julia Lebedeva [1,5,6],
Andrei Rozov [7,8,9], Androniki Raftogianni [1,10], Rishika Tiwari[11], Shai Netser [11],
Ana Zovko[1,2], Huma Shaheen[1,2], Jonas Schimmer[1,2], Ryan Patwell [1,2],
Clémence Denis[4], Valentin Grelot[4], Hugues Petitjean[1,2,4], Lan Geng[12],
Dimitri Hefter[13,14], Arjen Boender [15,16,17], Yuval Podpecan[18],
Franziska Schommer [18], Tim Schubert [18], Anna Sanetra [19],
Aleksandra Trenk [19], Anna Gugula [19], René Hurlemann[20], Shlomo Wagner [11],
Yulong Li [12], Ferdinand Althammer[18], Anna Blasiak [19], Sarah Melzer [21],
Hannah Monyer [22], Alexandre Charlet [4], Marina Eliava[1,2] ✉ &
Valery Grinevich [1,2,23] ✉

The hypothalamic neuropeptide oxytocin is best known for its prosocial behavioral effects. However, the precise anatomical and cellular targets for oxytocin in the cortex during social behavior remain elusive. Here we show that oxytocin neurons project directly to the medial prefrontal cortex where evoked axonal oxytocin release facilitates social behaviors in adult female rats. In conjunction, we report that local oxytocin receptor-expressing (OTR⁺) cells are predominantly interneurons, whose activation promotes social interaction. Notably, this prosocial effect persists even under physiological challenge (hunger), pointing to a dedicated prosocial circuit capable of overriding primary survival drives. We further demonstrate that activation of these OTR⁺ interneurons inhibits principal cells specifically projecting to the basolateral amygdala, thus providing a putative mechanism of selective oxytocin action in this sociability-promoting cortical network.

The neuropeptide oxytocin (OT) plays a central role in modulating various types of social behavior and is primarily synthesized in the hypothalamic paraventricular (PVN), supraoptic (SON), and accessory nuclei[1,2]. Hypothalamic OT is subsequently transported to the posterior lobe of the pituitary gland and released into the blood where it elicits milk ejection during lactation and uterine contractions during childbirth[3]. Beyond its peripheral effects, OT is actively involved in modulating neural circuits within the brain. Particularly, direct social interaction strongly activates the OT system via OT neurons in the PVN[4]. Axonal collaterals of OT neurons extend to various forebrain regions[5], which coincides with the expression of OT receptors (OTRs) at the site of putative neuropeptide release[6]. These areas, including the central nucleus of the amygdala, nucleus accumbens, ventromedial hypothalamic nucleus, lateral septum, and hippocampus[7,8], are pivotal in processing social information, emotional responses, and stress regulation[9]. The infusion of OT, evoked OT release, or activation of OTR neurons in these regions trigger a broad spectrum of region-specific social behaviors, such as social fear[5,10,11], social transmission and contingency[12], social memory[13], social avoidance[14], and aggression[15]. While most studies show that OT's behavioral effects

**Fig. 1 | OT neurons from both PVN and SON project to the infralimbic cortex.**
**A** The rat brain regions of interest: PVN and SON in the hypothalamus and the ILC as part of the mPFC. **B** OT innervation in the medial prefrontal cortex. **B1** Immuno-histochemical staining for OT in the hypothalamus showing OTergic nuclei, the PVN and SON. 3 V: third ventricle, opt: optic tract. Scalebar 500 μm. **B2** Coronal section of the mPFC subdivisions (Cgt, PrL, ILC) and high-magnification images of OT fiber distribution for each sub region. Scalebar left: 500 μm, middle: 300 μm, right 100 μm. **B3** Density of OT-positive fibers quantified in digitized images. The number of OT-positive fibers per mm² was significantly higher in the ILC, compared to the mPFC dorsal subdivisions (11 sections obtained from $n$ = 5 rats). **C** Innervation from the PVN and SON. **C1** Injection of a rAAV expressing Venus under the

control of the OT promoter (rAAV-OTp-Venus) into either the PVN or SON in separate animals, with respective injection sites. Scalebar 250 μm. **C2** Automated analysis of fibers in the ILC quantifying their density by the sum of their volume originating from the PVN or SON revealed stronger innervation from the SON. **C3** Localized analysis of fiber innervation originating from the PVN or SON depending on the layer in the ILC (layer 1, layer 2/3 or layer 5/6) revealed most fibers in layer 5/6 ($n$ = 6 sections obtained from 3 animals per group), $p$ = 0.017 (layer x nucleus). Statistical significance is indicated as * $p < 0.05$, ** $p < 0.01$, *** $p < 0.001$, **** $p < 0.0001$. Statistical significance of post-hoc tests is indicated as # $p < 0.05$, ## $p < 0.01$, ### $p < 0.001$, #### $p < 0.0001$. Error bars show mean±sd. For details on statistical tests please refer to Supplementary Data 1.

arise from subcortical circuits, emerging work has started to uncover its role in cortical signaling. OT signaling in the murine auditory cortex and its role in nursing, has been extensively studied by Froemke and colleagues[16–18]. Moreover, OT has been shown to drive top-down modulation of early sensory processing in the olfactory bulb to enhance social recognition[19]. Additionally, several studies have directly implicated cortical OT signaling in the regulation of socio-sexual behaviors: in the medial prefrontal cortex (mPFC), Heintz and colleagues identified OTR-expressing (OTR⁺) interneurons that gate female sociosexual behavior and revealed a sex-dimorphic OTR–CRH interaction shaping social and anxiety-related responses[20,21]. Moreover, OTR-expressing glutamatergic neurons in the mPFC have been shown to selectively regulate social recognition memory[22].

The mPFC is instrumental in orchestrating various higher order and computational aspects of social behavior in mammals, including but not limited to social motivation[23], recognition of social cues[24,25], and decision-making[26]. The connections of the mPFC to other brain areas like the amygdala[27] and hypothalamus[28] suggest its role in top-down executive control, such as guiding goal-directed social behaviors. In rodents, the mPFC can be divided into three distinct brain subregions, each with unique functions and connectivity: the infralimbic cortex (ILC), the prelimbic cortex (PrL) and the cingulate cortex (Cgt)[29]. Considering the widespread influence of OT across various brain regions and its known input to the mPFC[5], it is likely that OT modulates mPFC functions.

Despite the inspiring demonstrations that OT action and/or OTR⁺ neurons in the mPFC modulate paternal[30] and sexual behavior[20] in voles and mice, respectively, the OT-modulated circuit that mediates same-sex affiliative (non-sexual) social behavior in the mPFC remains largely unexplored. To address this gap, we aimed to describe the pattern of OT innervation in the mPFC (1), the composition of the local OT receptive cells (2), and their downstream target structures (3), which might promote social behavior in adult female rats.

## Results
### Evoked OT release in the infralimbic cortex facilitates social behavior in female rats

Immunohistochemical (IHC) labeling revealed the presence of OT projections through the entire mPFC (Figs. 1A, B1–2). The highest density of OT axons was found in the infralimbic cortex (ILC) compared to the cingulate (Cgt) and prelimbic (PrL) cortices (49.9 ± 9.2 vs. 1.5 ± 0.8 axonal segments per mm² in 50 μm coronal slices; this and following data are presented as mean ± sd; $p$ = 0.0002, details of statistical tests are provided in Supplementary Data 1; sample sizes are reported in the figure legends) (Fig. 1B3). Based on this finding, we focused specifically on the ILC in all follow-up experiments. Considering that the IHC approach labeled all OT fibers independent of origin, we next employed a viral approach to label specific nuclei in the hypothalamus to determine the source of OT inputs to the ILC. Therefore, we injected a recombinant adeno-associated virus (rAAV)

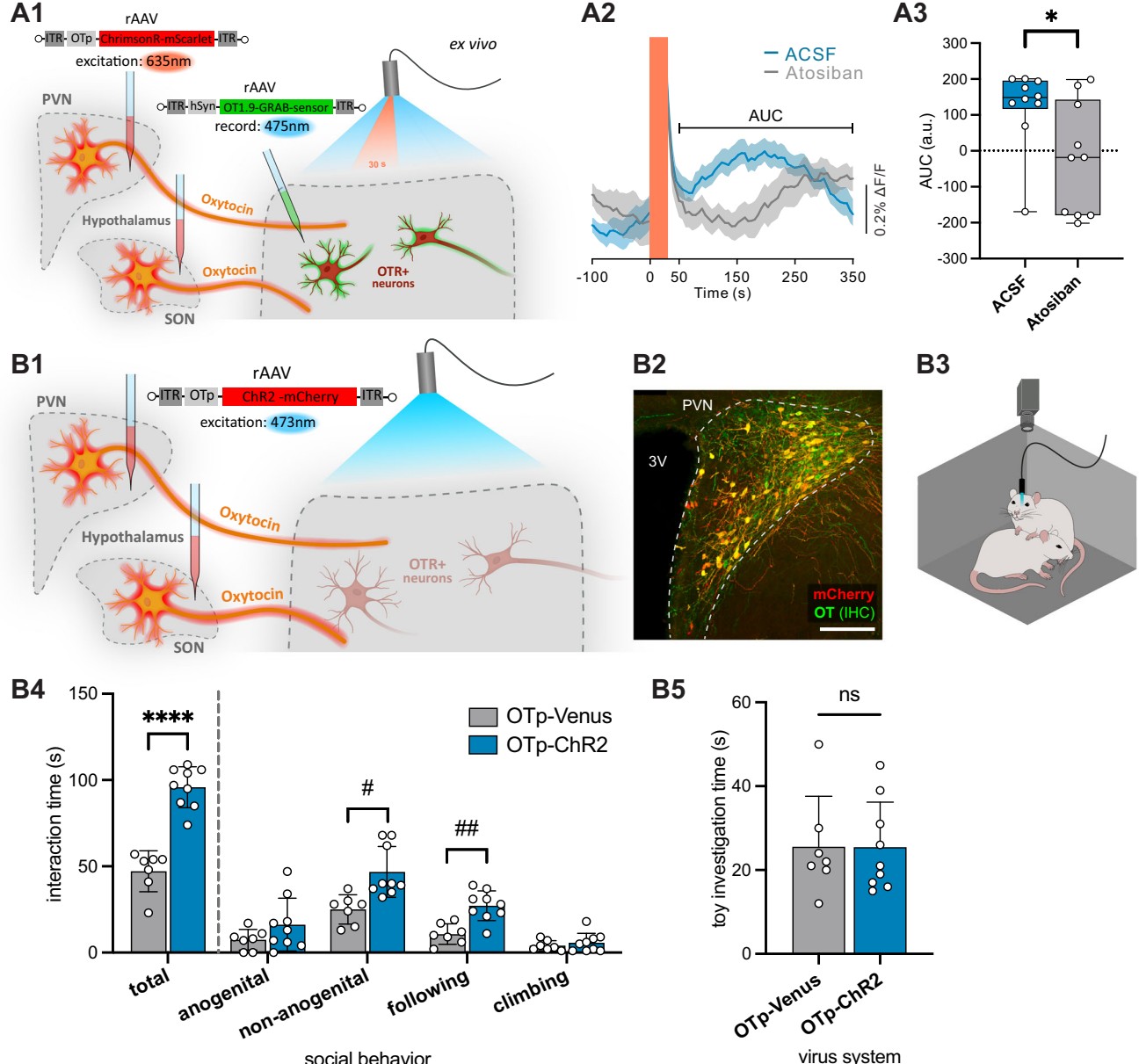

**Fig. 2 | Evoked OT release in the ILC facilitates social behavior in female rats.** **A** Axonal OT release in the ILC ex vivo. **A1** Injection of a rAAV expressing the excitatory ChrimsonR under the control of the OT promoter (rAAV-OTp-ChrimsonR-mScarlet) into the hypothalamic nuclei of female rats and injection of a rAAV expressing an OT-sensor (rAAV-OT1.9-GRAB-sensor) in the ILC. Stimulation of the OT fibers in the ILC (625 nm light, 30 Hz, 10 ms light on, for 30 s). **A2** Peri-stimulus GRAB-OT1.9 fluorescence changes in the ILC in ACSF-only solution (blue) or in the presence of Atosiban (OTR-antagonist, gray). Orange square marks the stimulation period. Bracket indicates the timeframe for area under the curve (AUC) analysis in panel A3. Data presented as mean ± sem. **A3** Recordings in ACSF condition showed higher AUC, calculated between 50 and 350 seconds after light stimulation, compared to recordings in the Atosiban condition ($n = 10$ sections). **B** Optogenetic activation of OT axons in the ILC during free social interaction. **B1** Injection of a rAAV expressing the excitatory ChR2 under the control of the OT promoter (rAAV-OTp-

ChR2-mCherry) into the hypothalamic nuclei of female rats and implantation of an optic fiber above the ILC. **B2** Injection site showing high specificity of the rAAV-OTp-ChR2-mCherry (red) after counterstaining for OT (green). Scalebar 200 μm. **B3** 5-min social interaction paradigm with an unknown conspecific; blue light (BL) stimulation present for the first two min of the session. **B4** Animals injected with OTp-ChR2 ($n = 9$ animals) showed significantly ($p < 0.0001$) increased social interaction time compared to control animals (injected with OTp-Venus, $n = 7$ animals), especially increased non-anogenital sniffing and following behavior. **B5** Investigation time of a toy rat in the open field remained unaltered by optogenetic activation. Statistical significance is indicated as * $p < 0.05$, ** $p < 0.01$, *** $p < 0.001$, **** $p < 0.0001$. Statistical significance of post-hoc tests is indicated as # $p < 0.05$, ## $p < 0.01$, ### $p < 0.001$, #### $p < 0.0001$. Error bars show mean±sd. Box plots show the median, 25th–75th percentiles, and whiskers from minimum to maximum. For details on statistical tests please refer to Supplementary Data 1.

equipped with the OT promoter (rAAV-OTp-Venus) either in the PVN or the SON of female rats (Fig. 1 C1), and analyzed fiber density in the ILC with a semi-automated Imaris approach. We found significantly denser innervation originating from the SON than from the PVN ($135.4 \pm 65.6$ vs. $51.8 \pm 32.1$ $10^3 \times$ μm³ sum volume of fibers, $p = 0.0253$) (Fig. 1C2). For both hypothalamic nuclei we found that they mostly innervate layer 5/6, with additional substantial innervation in layer 1,

while the least innervation was found in layer 2/3 (Fig. 1 C3) ($p = 0.0007$ for layer 1 vs. layer 5/6 SON projections, $p < 0.0001$ for layer 2/3 vs. layer 5/6 SON projections; significance of post hoc tests of ANOVAs and mixed effect models are indicated with # in figures).

Next, we investigated whether OT axons release the neuropeptide in the ILC. To this end, we injected a rAAV expressing an opsin under the control of the OT promoter into the PVN and SON (rAAV-OTp-

ChrimsonR-mScarlet, Supplementary Fig. S1A, right: patch-clamp verification of opsin functionality, Supplementary Fig. S1B1: representative injection site). In parallel, we injected a rAAV expressing the GRAB OT 1.9 sensor (rAAV-hSyn-OT1.9-GRAB) [kindly provided by Prof. Yulong Li, GRAB OT 1.0 published in:[31] in the ILC, and performed imaging experiments in acute ILC slice preparations. We observed a significant increase of GRAB OT 1.9 fluorescence emission upon stimulation of OT fibers (625 nm light, 30 Hz, 10 ms light on, for 30 s) (Fig. 2A1), consistent with OT release and consecutive binding[32]. This effect was blocked by incubation of the tissue with the OT receptor antagonist (Atosiban) (Fig. 2A2–3, Supplementary Fig. S1B2–3) ($123.4 \pm 110.5$ vs. $-23.9 \pm 155.8$ AUC, $p = 0.0232$).

To functionally probe the hypothalamic-ILC OT pathway, we next expressed channelrhodopsin 2 (ChR2) in OT neurons of the PVN and SON (rAAV-OTp-ChR2-mCherry, Supplementary Fig. S1 A left for patch-clamp verification of opsin functionality) and implanted an optic fiber above the ILC (Fig. 2B1-2). Next, rats were exposed to an unknown, age-, strain- and sex-matched conspecific (Fig. 2B3) in an open field arena (OF) for 5 min, where we analyzed the social behavior throughout the session upon light stimulation of OT fibers in the ILC for the first 2 min of the session. In keeping with previous studies[4,5], we employed adult female rats to exclude the aggressive component of social behavior often observed between unfamiliar males[33]. Social interaction time was significantly increased (total interaction time $47.1 \pm 11.8$ s vs. $95.9 \pm 11.9$ s, $p < 0.0001$) during and immediately after optogenetic stimulation with blue light (BL, 470 nm, 30 Hz,10 ms pulses, laser power ~10 mW, 120 s) compared to controls expressing Venus fluorescent protein instead of ChR2. This effect was mainly due to significantly increased non-anogenital sniffing and following behaviors ($25.0 \pm 8.5$ s vs. $46.8 \pm 14.7$ s, $p = 0.0103$ and $10.7 \pm 5.9$ s vs. $27.2 \pm 8.6$ s, $p = 0.0019$, respectively) (Fig. 2 B4). Importantly, no difference was observed between the two groups when comparing the time they spent investigating a toy rat ($25.6 \pm 12.0$ s vs. $25.4 \pm 10.8$ s, $p = 0.9826$) (Fig. 2 B5), indicating that the behavior reflects a social component rather than general interest and novelty seeking. No changes in locomotor activity, exploratory, or anxiety-related behaviors were found (Supplementary Fig. S2). Altogether, these results indicate that optogenetic activation of the OTergic hypothalamic-ILC projections are sufficient to enhance social behavior in female rats.

## OTR+ neurons in the ILC are intrinsically active during social behavior

Next, to target and manipulate OTR+ neurons in the ILC we utilized newly generated OTR-Cre knock-in rats[32]. The rats were injected with a rAAV expressing GFP in a Cre-dependent (double-floxed-inverted-orientation: DIO) manner into the ILC (rAAV-Ef1a-DIO-GFP) (Fig. 3A1, see also Supplementary Fig. S3B for further characterization of the rat line and protein expression). Semi-automated quantification revealed that among the entire ILC neuronal population, visualized by NeuN (Fig. 3A2), 1.2% of neurons expressed OTRs with a higher occurrence in superficial layers 2/3 compared to the deeper layers 5/6 (1.6% and 0.9%, respectively, $p < 0.0001$) (Fig. 3A3, and Supplementary Fig. S3A). To monitor OTR+ neuronal intrinsic activity, we performed in vivo fiber photometry during social interaction utilizing a rAAV expressing a Cre-dependent Ca$^{2+}$ sensor (rAAV-Ef1a-DIO-GCaMP6s) and implanted optic fibers (Fig. 3B1–2). During repeated 5-min social interaction sessions, multiple behavioral events (non-anogenital and anogenital sniffing, and investigation of a toy) were scored and the aligned calcium activity of OTR+ neurons was compared (Fig. 3B3). Social sniffing behaviors performed by the test rat were accompanied by a significant increase of calcium signal in OTR+ neurons compared to baseline (z score for anogenital: $0.12 \pm 0.01$, non-anogenital: $0.18 \pm 0.01$), while toy investigation did not lead to a significant change in calcium signal ($0.04 \pm 0.01$) (mean ± sem, $p < 0.0001$ for anogenital and non-

anogenital sniffing compared to baseline, $p = 0.4707$ for toy) (Fig. 3 B4). Furthermore, we also compared the amplitude of calcium signals during social events that were initiated by the test rat (active social behavior) with the amplitude of calcium signals during social events performed by the stimulus rat (received social behavior) (Fig. 3B5). Only active social behavior increased calcium-dependent fluorescence in OTR+ neurons in the ILC (z score for active: $0.12 \pm 0.006$, $p < 0.0001$, received: $-0,01 \pm 0.008$, $p = 0.9035$, mean ± sem) compared to baseline (Fig. 3B6). Therefore, we conclude that OTR+ neurons in the ILC are highly and specifically active during active social behavior.

## OTR+ neurons in the ILC facilitate social behavior

After demonstrating that OTR+ neurons in the ILC are active during social interaction, we aimed to artificially activate them using optogenetic stimulation, in order to test whether activation of ILC OTR+ neurons affects social behavior. For that, we injected a rAAV expressing ChR2 in a Cre-dependent manner (rAAV-Ef1a-DIO-ChR2) into the ILC of OTR-Cre-rats and implanted optic fibers above the injection sites (Fig. 3C1–2). During the first 2 min of a 5-min social interaction session, OTR+ neurons were either activated by BL (30 Hz,10 ms pulses), or no light was applied for a baseline social behavior readout. Compared to the baseline, BL activation significantly increased social interaction times (total interaction time $45.6 \pm 15.8$ s vs. $87.4 \pm 30.6$ s, $p = 0.0124$), and more specifically increased anogenital and non-anogenital sniffing ($9.8 \pm 5.9$ s vs. $24.4 \pm 11.6$ s, $p = 0.0426$ and $26.4 \pm 9.7$ s vs. $49.4 \pm 21.2$ s, $p = 0.0036$, respectively) (Fig. 3 C3).

To verify that the endogenous OT/OTR+ signaling in the ILC modulates sociability, we introduced a local and discreet mutation in the OTR gene[34] (Supplementary Fig. S3D) by injecting a mixture of two rAAVs expressing Cas9 (rAAV-hSyn-spCas9) and a guide RNA specific for the OTR-sequence (rAAV-u6-gRNA(OTR)-hSyn-mCherry) or LacZ-sequence which served as a control (rAAV-u6-gRNA(lacZ)-hSyn-mCherry) (Fig. 3D1). Injection sites and viral expression were verified post mortem (Supplementary Fig. S3C). We first confirmed the depletion of OTRs in the ILC functionally by bath application of the selective OTR agonist, TGOT, and ex vivo patch-clamp electrophysiological recordings of OTR+ neurons in both groups (Supplementary Fig. S3E). TGOT-induced membrane depolarization was significantly abolished by the viral OTR depletion compared to the control group (Supplementary Fig. S3F). Notably, the ILC OTR depletion was characterized by a significantly decreased social interaction time (total social interaction time $71.5 \pm 31.1$ s vs. $41.3 \pm 22.2$ s, $p = 0.0259$) with an unfamiliar conspecific. This effect was driven by a significant decrease in anogenital sniffing ($36.2 \pm 26.6$ vs. $21.2 \pm 13.0$ s, $p = 0.0493$), while non-anogenital sniffing also trended towards a decrease in time (Fig. 3 D2). Importantly, no difference was observed regarding the investigation time of a toy rat ($37.7 \pm 18.0$ s vs. $41.7 \pm 22.5$ s, $p = 0.6656$) (Fig. 3D3).

## OTRs are predominantly expressed in GABAergic neurons of the ILC

A multiplex fluorescent in situ hybridization (RNAscope) approach was used to characterize the types of ILC OTR+ neurons (Fig. 4A1–3). Probes for both OTR as well as Cre were used to verify the specificity of our OTR-Cre rat line; whereas probes for vGlut1 (vesicular glutamate transporter), vGlut2 and vGAT1 (vesicular GABA transporter) were utilised to identify both glutamatergic and GABAergic neurons. We observed that $92.6 \pm 8.6\%$ of Cre-positive neurons were also OTR-positive, and vice versa $82.9 \pm 8.5\%$ of OTR-positive neurons were also Cre-positive, confirming high specificity of the transgenic OTR-Cre line (Fig. 4A4). Notably, the vast majority of OTR-positive neurons co-expressed vGAT1 (87.63%), while only a small fraction co-expressed vGlut1 (9.14%), and none co-expressed vGlut2 (Fig. 4A5). Moreover, we observed differences between the superficial and deeper layers of the ILC: OTR+ neurons in layer 2/3 were predominantly GABAergic (96.84% vGAT1-positive), while OTR+ neurons in layer 5/6 were prevalently

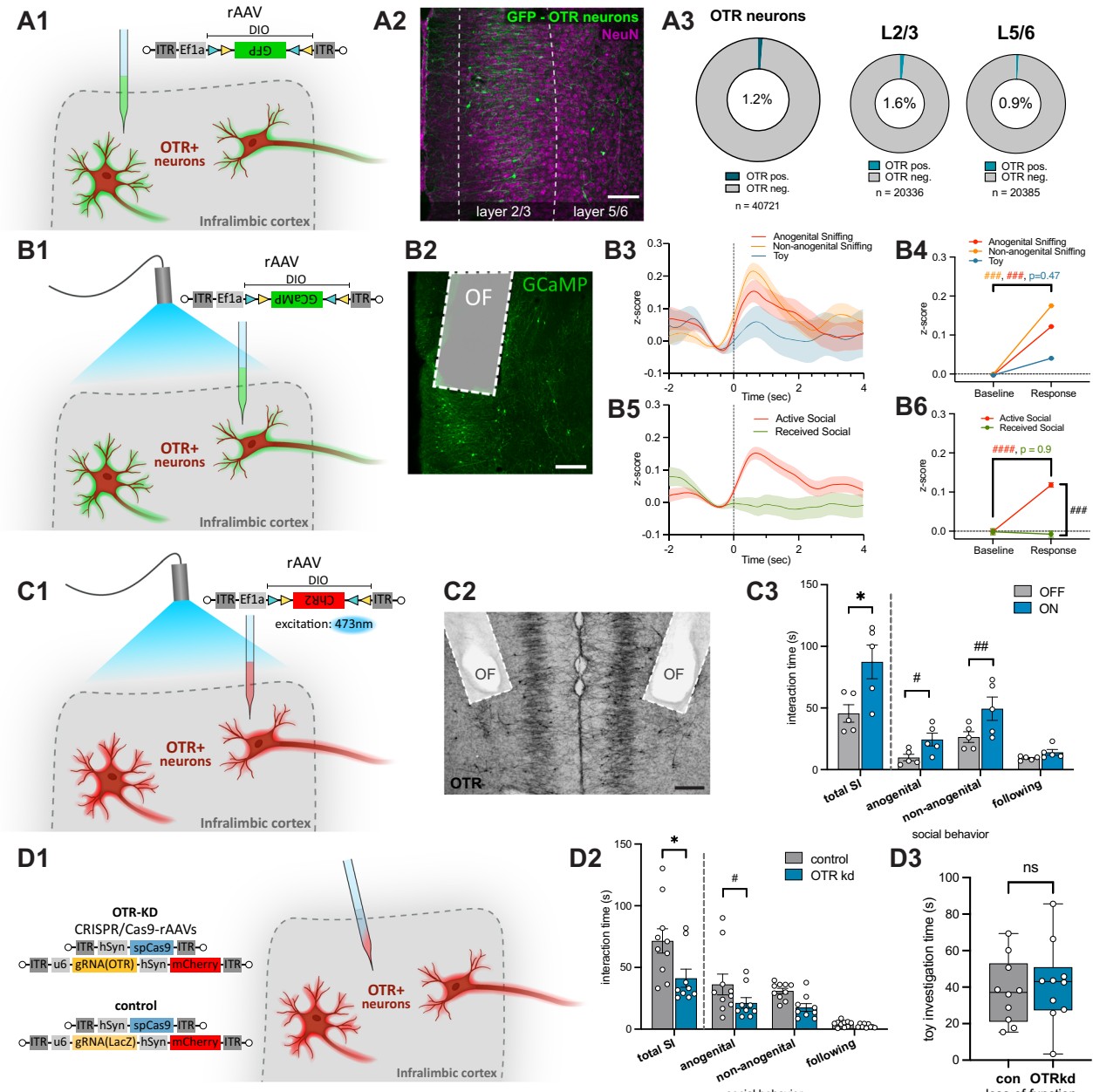

**Fig. 3 | OTR neurons in the ILC are active during social behavior and modulate sociability. A** ILC OTR⁺ neurons. **A1** Injection of a Cre-dependent rAAV expressing GFP **A2** counterstained for NeuN. Scalebar 150 μm. **A3** 1.2% of ILC neurons express OTRs, with similar distributions in superficial and deeper layers (*n* = 11 sections obtained from 4 animals; *n* in the figure indicates the number of tracked neurons). **B** OTR⁺ neuron activity during social interaction (SI). **B1** Injection of a Cre-dependent rAAV expressing GCaMP6s and optic fiber implantation into the ILC of OTR-Cre-rats. **B2** Injection site showing OTR⁺ neurons expressing GCaMP (green) and the optic fiber (OF) tract. Scalebar 200 μm. **B3** Calcium activities aligned to behavior onset (t = 0). 1321 observations from 4 animals averaged per behavior (anogenital sniffing *n* = 391, non-anogenital sniffing *n* = 706, toy investigation *n* = 224). **B4** Anogenital and non-anogenital sniffing behavior increased intracellular calcium in OTR⁺ neurons. **B5** Neuronal calcium activity during active (*n* = 1368) or received (*n* = 600) SI. **B6** Only active social behaviors increased OTR⁺ neuron calcium activity (*p* < 0.0001). Error bands show mean±sem. **C** Optogenetic

activation of ILC OTR⁺ neurons during SI. **C1** Injection of a Cre-dependent rAAV expressing ChR2 and optic fiber implantation into the ILC. **C2** Injection site showing OTR⁺ neurons infected with ChR2 (black) and the optic fiber tract. Scalebar 200 μm. **C3** After two minutes of BL stimulation animals spent significantly more time socially interacting (*n* = 5 animals). **D** CRISPR/Cas9-mediated OTR depletion decreased SI time. **D1** Specific and functional knockdown of ILC OTRs via injection of a CRISPR/Cas9-based virus system (*n* = 9 animals loss-of-function group, *n* = 10 animals control group). **D2** OTR depletion significantly (*p* = 0.026) decreased SI time. **D3** Investigation time of a toy rat remained unaltered. Statistical significance is indicated as * *p* < 0.05, ** *p* < 0.01, *** *p* < 0.001, **** *p* < 0.0001. Statistical significance of post-hoc tests is indicated as # *p* < 0.05, ## *p* < 0.01, ### p < 0.001, #### *p* < 0.0001. Error bars show mean±sd. Box plots show the median, 25th-75th percentiles, and whiskers from minimum to maximum. For details on statistical tests please refer to Supplementary Data 1.

GABAergic (78.02% vGAT1-positive), but also included a substantial number of glutamatergic neurons (17.58% vGlut1-positive) (Fig. 4A5). In line with the viral approach (Fig. 3A), OTR⁺ neurons constitute only a small population of 1.35% of all neurons). Amongst the entire

GABAergic and glutamatergic population, OTR⁺ neurons make up 6.82% and 0.16% respectively (Fig. 4A6).

To complementarily characterize the distinct glutamatergic and GABAergic OTR⁺ populations in the ILC, we injected both: a rAAV

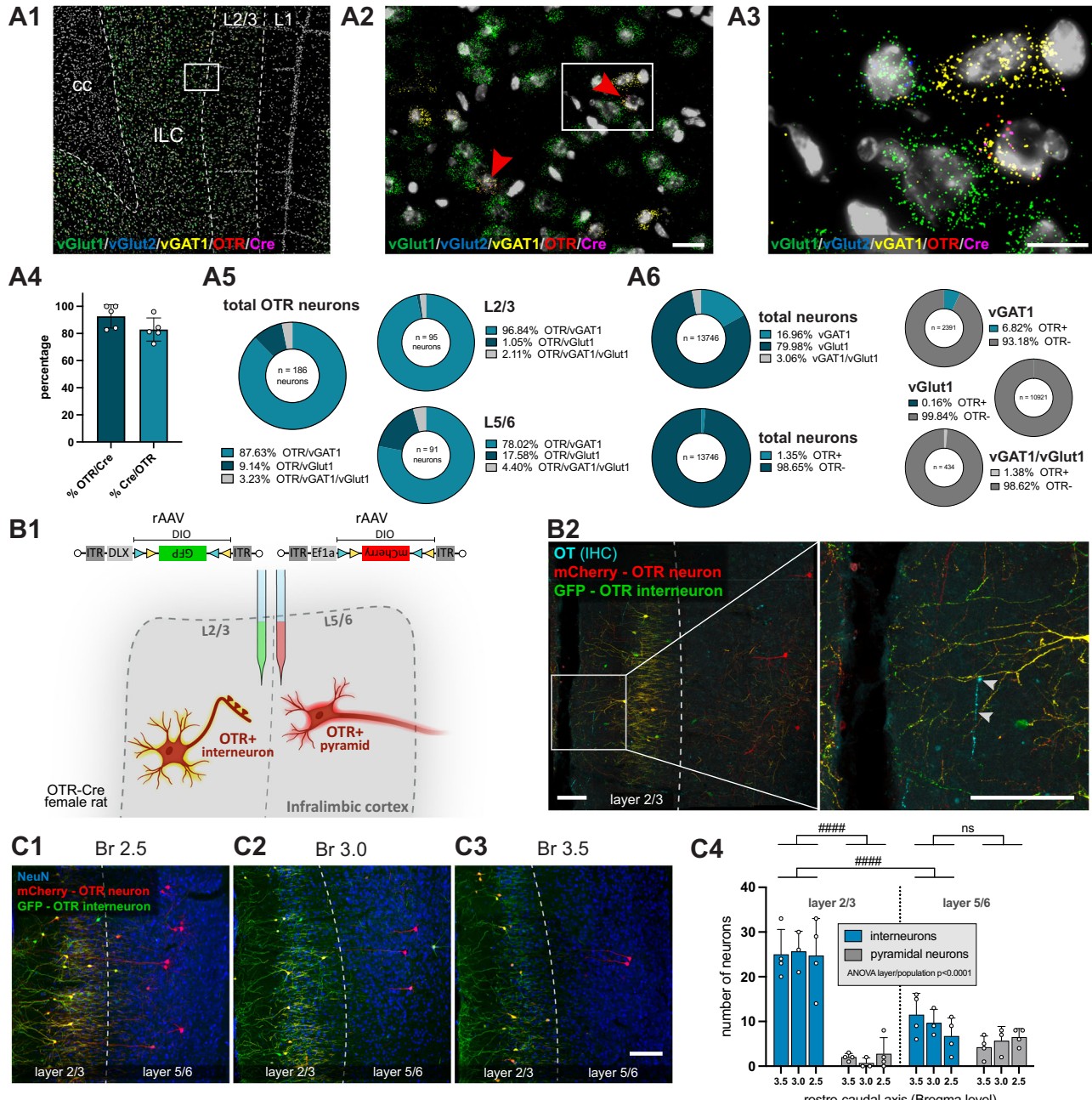

**Fig. 4 | OTR⁺ cell types and their apposition to OT axons. A** RNAscope based characterization of ILC OTR⁺ neurons ($n = 5$ animals, 2 sections each). **A1** Probes for vGlut1 (green), vGlut2 (blue), VGAT1 (yellow), OTR (red) and Cre (magenta) were used to visualize mRNA co-expression in the ILC of female OTR-Cre rats. **A2–A3** Magnifications show representative vGAT1-positive neurons co-expressing Cre and OTR. Scalebar 20 µm and 10 µm. **A4** Colocalization of OTR and Cre mRNA shows high specificity of the transgenic Cre rat line. **A5** The majority of ILC OTR⁺ neurons are GABAergic interneurons, with a small fraction of glutamatergic neurons predominantly found in the layers 5/6. **A6** In-situ-hybridization showed that 1.35% of all neurons in the ILC are OTR⁺ neurons, with 6.82% of all GABAergic neurons being OTR⁺. **B** Viral vector-based characterization of OTR⁺ neurons in the ILC. **B1** Injection of a mixture of two rAAVs: one expressing mCherry in a Cre-dependent manner to label the entire population of OTR⁺ neurons and a second rAAV expressing GFP in a

Cre-dependent manner only in OTR⁺ interneurons (utilizing the DLX-enhancer for specific expression in interneurons) into the ILC of OTR-Cre female rats. **B2** Representative confocal image depicting OTR⁺ interneurons dominating layers 2/3 (yellow), and individual OTR⁺ pyramidal neurons localized in layers 5/6 (red), with OT fibers (blue, white arrows) in both subdivisions of the ILC. Scalebar 100 µm. **C** Quantification of OTR⁺ neurons in layer 2/3 and layer 5/6 along the rostro-caudal axis at **C1** Bregma 2.5, **C2** Bregma 3.0 and **C3** Bregma 3.5, Scalebar 100 µm. **C4** Semi-automated, high-throughput analysis of the number of OTR⁺ neurons identified as interneurons (yellow) or pyramidal neurons (red) in the ILC ($n = 4$ animals, each with 3 sections at three different bregma levels), $p < 0.0001$ (layer x population). Statistical significance of post-hoc tests is indicated as # $p < 0.05$, ## $p < 0.01$, ### $p < 0.001$, #### $p < 0.0001$. Error bars show mean±sd. For details on statistical tests please refer to Supplementary Data 1.

labeling all OTR⁺ neurons in red (rAAV-Ef1a-DIO-mCherry) as well as a second rAAV labeling all OTR⁺ GABAergic interneurons (specificity achieved by DLX-enhancer[35]) in green (rAAV-DLX-DIO-GFP, as all OTR⁺ interneurons will also be OTR⁺ neurons, they will appear yellow) into

the ILC of female OTR-Cre rats (Fig. 4B1). Similar to our findings with the RNAscope approach, we could identify an OTR⁺ interneuron population in layer 2/3 (yellow) and a mixed OTR⁺ pyramidal (red) and interneuron population in layer 5/6 (Fig. 4B2, and Supplementary

Fig. S4 A). The results of semi-automated quantification of OTR[+] neurons in distinct cortical layers (2/3 and 5/6) and along the rostro-caudal axis (Bregma 2.5, 3.0, 3.5), resembled the findings of in situ hybridization: in layer 2/3, 92.93% of OTR[+] neurons are interneurons, while layer 5/6 also has a larger population of 37.04% pyramidal neurons (Fig. 4C). No significant differences of OTR[+] neuron distribution were found along the rostro-caudal axis. Together, these experiments demonstrate that the largest population of OTR[+] neurons in the ILC consists of GABAergic interneurons located in layer 2/3.

Histological quantification after viral labeling of OTR[+] cell types confirmed that the majority of OTR[+] neurons were GAD-expressing interneurons immunopositive for parvalbumin and/or calbindin (Supplementary Fig. S4B). Detailed anatomical imaging of cell bodies of OTR[+] neurons and OT fibers revealed the localization of OT-immunoreactive axons in close proximity to OTR[+] neurons in different layers, pointing to axonal release as a probable source of OT acting on ILC OTR[+] neurons and corroborating our aforementioned optogenetic experiments (Supplementary Fig. S4C). Notably, OT fibers were found to be significantly closer to OTR[+] neurons in layer 2/3 compared to layer 5/6 ($0.04 \pm 0.07\,\mu m$ vs. $0.91 \pm 0.55\,\mu m$, $p < 0.0001$) (Supplementary Fig. S4D).

To verify the presence of functional OTRs, we performed an ex vivo patch-clamp experiment, where OTR[+] neurons were virally labeled by rAAV-Ef1a-DIO-GFP (Supplementary Fig. S4E1), and classified based on their firing pattern, localization, and biocytin-based morphology visualization (Supplementary Fig. S4E2). All tested cells (interneurons $n = 9$; pyramids $n = 5$) were depolarized in response to bath application of selective OTR agonist (20-100 nM TGOT) (Supplementary Fig. S4E3).

## Activity of ILC OTR[+] interneurons promotes both social interaction and social preference

Given that the major population of ILC OTR-expressing neurons consists of inhibitory cells (primarily in layers 2/3), we aimed to investigate whether modulation of their activity induces similar behavioral effects as recruiting all ILC OTR[+] neurons (Fig. 3C). To achieve this, we employed a rAAV equipped with the interneuron-specific enhancer DLX[35] to express Cre-dependent DREADDS (designer receptor exclusively activated by designer drugs, either hm3Dq(Gq) for activation or hM4Di(Gi) for inhibition) in OTR[+] interneurons (rAAV-DLX-DIO-hm3Dq-GFP or rAAV-DLX-DIO-hm4Di-GFP) of the ILC (Fig. 5A1). Post-hoc IHC analysis verified the main population of OTR[+] interneurons being located in layer 2/3 (Fig. 5A2). The two groups of rats (Gq: activation or Gi: inhibition) received CNO (3 mg/kg) or saline i.p. injections 40 min before being introduced into a free social interaction session with an unknown conspecific. Chemogenetic activation of OTR[+] interneurons significantly increased social interaction time ($61.5 \pm 17.7$ s saline vs. $77.0 \pm 16.4$ s CNO, $p = 0.0346$), while chemogenetic inhibition significantly decreased social interaction time ($61.0 \pm 16.6$ s saline vs. $42.7 \pm 13.3$ s CNO, $p = 0.0197$) (Fig. 5B). Investigation of a toy rat was not altered by either activation or inhibition of OTR[+] interneurons (activation: $35.7 \pm 15.6$ s vs. $40.8 \pm 19.7$ s, $p = 0.4381$; inhibition: $39.2 \pm 16.0$ s vs. $49.0 \pm 28.5$ s, $p = 0.4372$). Given the previously indicated anxiolytic effect of OT action in OTR[+] interneurons of male mice[21], we performed OF and elevated zero maze (EZM) tests to investigate the putative effect of ILC OTR[+] interneuron modulation on anxiety levels in female rats. Notably, chemogenetic activation of these cells changed neither anxiety nor locomotion parameters in our experiments (Supplementary Fig. S5A, B). Moreover, we conducted a social novelty preference test and observed no difference induced by chemogenetic activation of ILC OTR[+] interneurons (corner: $p = 0.0100$, treatment: $p = 0.1474$, corner x treatment: $p = 0.1937$, post-hoc tests all ns) (Supplementary Fig. S5 C). Taken together, these experiments showed that recruitment of ILC OTR[+] interneurons is sufficient to promote social behavior without alterations of social novelty preference or anxiety.

To further elucidate the social motivational aspect of the observed behavior, we analyzed whether manipulation of the OTR[+] interneuron population has an effect on social preference over another rewarding stimuli. Therefore, we subjected animals to an adapted social vs. food preference paradigm[36] (Supplementary Fig. S5E1–2), where in a satiated state animals typically prefer social interaction, while food restriction for 24 h increases the interest in food. Rats were injected either with the chemogenetic excitatory rAAV system (rAAV-DLX-DIO-hm3Dq-GFP, 'Gq') previously used (Fig. 5A1), or with a rAAV lacking the Gq sequence as a control (rAAV-DLX-DIO-GFP, 'GFP'). In the food vs. social preference experiment we observed that animals spent significantly more time at the social compared to the food corner (main effect corner, $p < 0.0001$), and that food deprivation altered allocation across corners as expected (more time spent investigating the food corner after 24hrs of food restriction; satiety × corner, $p = 0.0004$). Most importantly, chemogenetic activation influenced the interaction time with the corners in a group specific manner (virus × corner, $p = 0.0431$), pointing to a significant increase in social corner interaction time independently of satiety state (no interaction of virus x corner x satiety state $p = 0.2387$). Additional post-hoc tests showed greater social corner time in Gq vs. GFP condition under deprivation ($p = 0.0155$), with no group difference in food corner time (adjusted $p = 0.9766$) (Fig. 5C1). Notably, this effect was driven by an increased average length of bouts (average length per visit of the social corner, virus × corner, $p = 0.0475$) after CNO injection (Fig. 5C2), while the number of bouts (number of visits to the social corner) (Supplementary Fig. S5 E3) remained unaltered. To further investigate the effect of ILC OTR[+] interneurons on non-social interest in a non-competing context we performed a similar test where only food was offered in one of the corners, with the second corner being empty. In this experiment, food deprivation had a significant effect on food investigation time (main effect food restriction, $p = 0.0448$), but chemogenetic activation of OTR[+] interneurons did not alter it (main effect virus, $p = 0.6628$). The number of bouts of food investigation, as well as distance traveled and speed during the test were also not affected (Supplementary Fig. S5D1–4). Together, these findings indicate that when a rat is presented with conflicting interests, ILC OTR[+] interneurons promote social engagement in the hungry state, overriding competing homeostatic demands while not exerting a direct anorexigenic influence.

## OTR[+] fast-spiking interneurons inhibit pyramidal neurons in the ILC

The aforementioned ex vivo patch-clamp experiments and consecutive biocytin-based morphological analysis revealed that most layer 2/3 OTR[+] cells in the ILC display fast-spiking firing patterns (83%, $n = 24$ patched cells, slices obtained from 5 rats) and exhibits axonal arborization characteristic for axo-axonic interneurons (Chandelier cells) (85.7%, $n = 14$)[37,38] (Fig. 5D). Bath application of TGOT at both low and high concentrations (20 nM and 100 nM, respectively) during ex vivo patch-clamp recordings reliably and significantly increased their firing activity. Note that although the percentage of responding cells did not depend on the TGOT concentration, the average firing rate was twice as high at higher agonist concentrations (Supplementary Fig. S5F). Furthermore, IHC staining for ankyrin (a marker for the axon initial segment) revealed that the terminals of axon arbors of the OTR[+] neurons form distinct arrays (so-called "cartridges") along the axon initial segments of OTR-negative (OTR[-]) pyramids (Fig. 5E). Based on these morphological and functional features we could identify the vast majority of OTR[+] interneurons in layer 2/3 of the ILC as Chandelier cells.

To functionally test whether OTR[+] interneurons in layer 2/3 provide an inhibitory synaptic input to the neighboring pyramidal neurons, we performed ex vivo paired patch-clamp recordings in brain slices. Three weeks following an injection of a rAAV expressing GFP in a

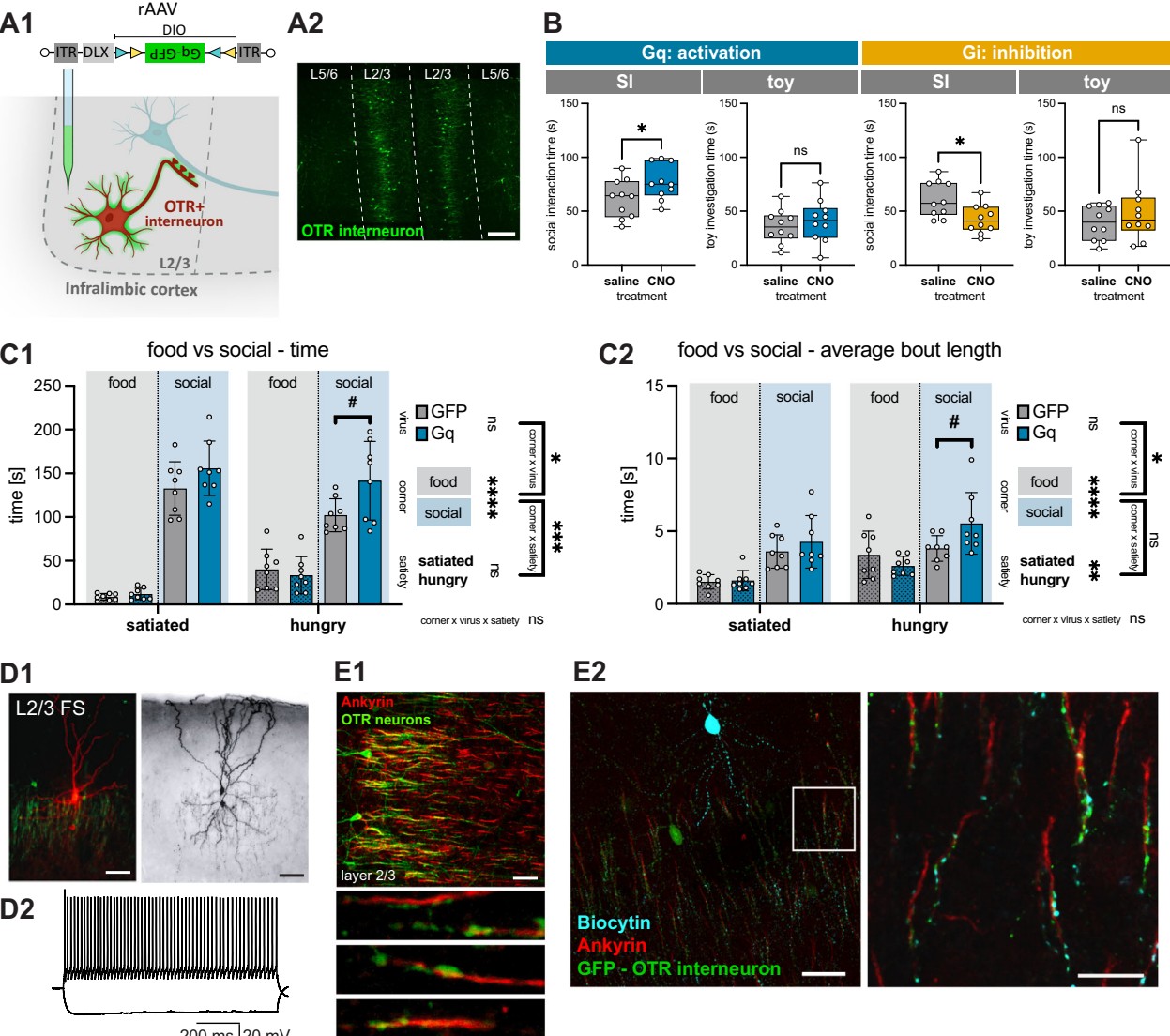

**Fig. 5 | OTR⁺ interneurons modulate social behavior. A** Chemogenetic manipulation of ILC OTR⁺ interneurons. **A1** Injection of a rAAV expressing the excitatory (Gq) or inhibitory (Gi) DREADD in a Cre-dependent manner in all OTR⁺ interneurons (via DLX enhancer) in the ILC of transgenic OTR-Cre female rats. **A2** Injection site showing localization of OTR⁺ interneurons (green) in layer 2/3. Scalebar 150 μm. **B** Chemogenetic manipulation of OTR⁺ interneurons via CNO injection before a 5 min social interaction (SI) session with an unknown conspecific in the open field. Activation ($n = 10$ animals) or inhibition ($n = 10$ animals) of OTR⁺ interneurons significantly increased or decreased social interaction time, respectively. No effect on investigation time of a toy rat. **C** Food vs. social paradigm. **C1** Chemogenetic activation of OTR⁺ interneurons increased time spent investigating the social corner ($n = 8$ animals per group), $p = 0.043$ (corner x virus). **C2** This effect was driven by an increase in the average length of bouts, $p = 0.048$ (corner x virus). **D** Morphological and functional characterization of OTR⁺ Chandelier neurons. **D1** Fluorescent (left)

and DAB (right) staining of a biocytin-filled OTR⁺ ILC neuron revealed the morphological features of Chandelier cells bearing vertically aligned axonal cartridges. Scalebar 50 μm. **D2** Representative ex vivo recording of OTR⁺ Chandelier interneuron showing its fast-spiking (FS) activity. **E1** Ankyrin staining (red, marker for axon-initial segments) revealed the close proximity of OTR⁺ boutons (green, viral labeling) to the initial segment of the axon of the local OTR-negative (OTR⁻) pyramidal neurons. Scalebar 50 μm. **E2** Biocytin filling (blue) of OTR⁺ neurons (green) and staining for ankyrin (red) verified typical Chandelier morphology in 12 of 14 OTR⁺ interneurons. Scalebar 50 μm overview, 25 μm zoom. Statistical significance is indicated as * $p < 0.05$, ** $p < 0.01$, *** $p < 0.001$, **** $p < 0.0001$. Statistical significance of post-hoc tests is indicated as # $p < 0.05$, ## $p < 0.01$, ### $p < 0.001$, #### $p < 0.0001$. Error bars show mean±sd. Box plots show the median, 25th–75th percentiles, and whiskers from minimum to maximum. For details on statistical tests please refer to Supplementary Data 1.

Cre-dependent manner into the ILC (rAAV-Ef1a-DIO-GFP), we searched for synaptic connections between fluorescently labeled OTR⁺ neurons and non-fluorescent OTR⁻ pyramidal cells in layer 2/3 (Fig. 6A1). We observed that stimulation of putatively presynaptic OTR⁺ interneurons led to generation of a post-synaptic potential in 43% of recorded pairs with latency indicative of monosynaptic connections (Fig. 6A2–3). Additionally, we applied a population-based ex vivo approach, utilizing optogenetic stimulation of ChR2-expressing OTR⁺ neurons (in rAAV-Ef1a-DIO-ChR2-mCherry ILC injected OTR-Cre rats) while simultaneously recording layer 2/3 OTR⁻ pyramidal neurons (Fig. 6B1).

Consistent with the paired-patch recordings, BL stimulation resulted in a significant ($p < 0.0001$) decrease of action potential frequency in all recorded local pyramidal neurons (Fig. 6B2–3). In a follow-up ex vivo patch-clamp experiment we recorded spontaneous excitatory (EPSC) and inhibitory (IPSC) post-synaptic currents in layer 2/3 OTR⁻ pyramidal neurons. The bath application of TGOT significantly increased the IPSC amplitude, with no effect on EPSC amplitude or EPSC and IPSC frequency (Supplementary Fig. S6), further corroborating the inhibitory nature of the synaptic connection of OTR⁺ neurons in layer 2/3 targeting OTR-negative pyramidal neurons. Together, these

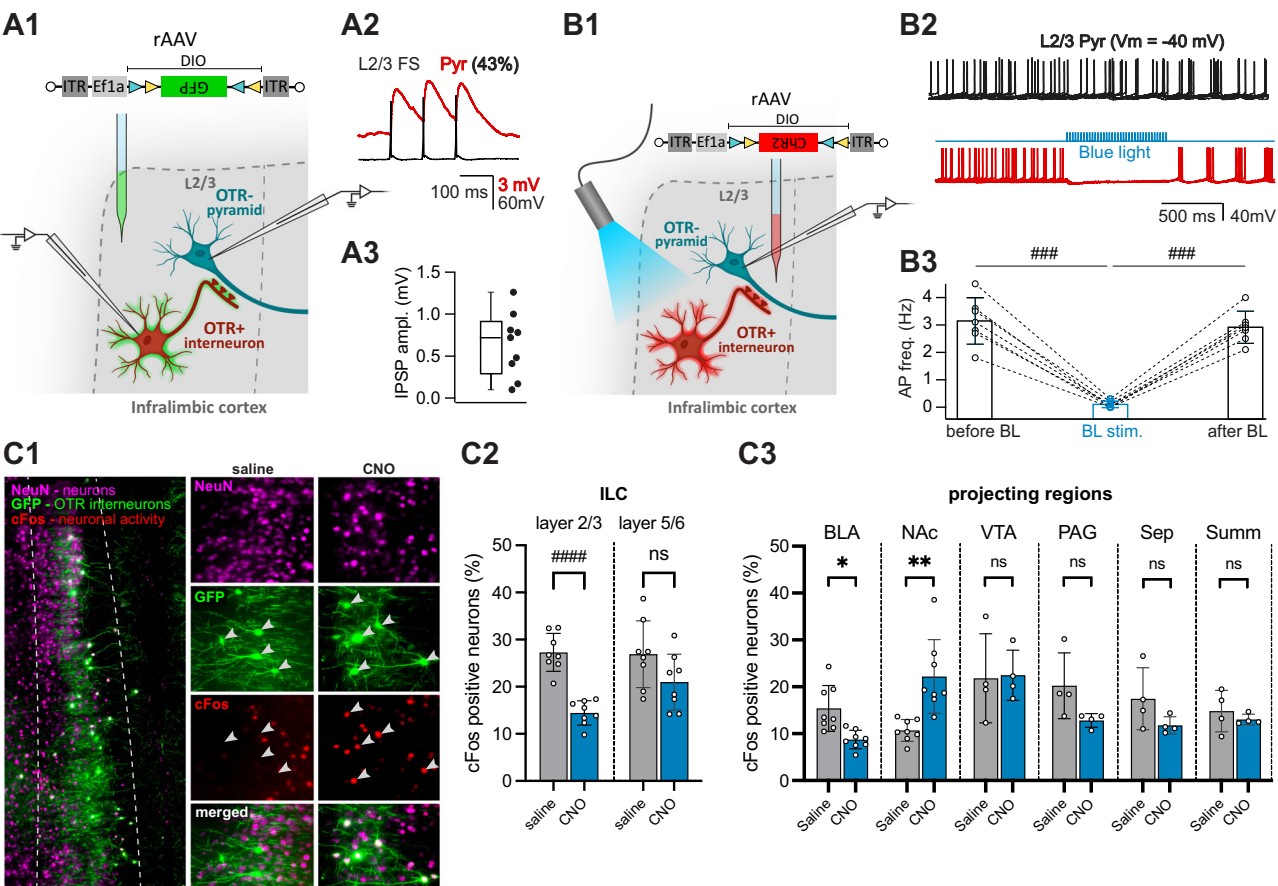

**Fig. 6 | OTR⁺ interneurons modulate neuronal activity of OTR⁻ pyramidal neurons. A** Paired patch-clamp recordings of OTR⁺ fast-spiking (FS) interneurons. **A1** Injection of a Cre-dependent GFP-expressing rAAV into the ILC of OTR-Cre rats; ex vivo paired patch-clamp experiment: stimulation of OTR⁺ interneurons (green) and recording of OTR⁻ pyramidal neurons. **A2** Representative paired recording: stimulation of L2/3 FS OTR⁺ interneuron (black) and the evoked postsynaptic potential of an OTR⁻ pyramidal neuron (red). **A3** Post-synaptic potential (PSP) amplitude of OTR⁻ pyramids after OTR⁺ interneuron stimulation (tested pairs: 21, connected pairs: $n = 9$, sections obtained from 3 rats). **B** Ex vivo optogenetic stimulation of OTR⁺ interneurons. **B1** Injection of a Cre-dependent ChR2-expressing rAAV into the ILC of OTR-Cre rats; ex vivo patch-clamp experiment: BL stimulation of OTR⁺ neurons (red) and recording of OTR⁻ pyramidal neurons. **B2** Representative traces of a L2/3 OTR⁻ pyramidal neuron at baseline (black) and during BL stimulation of OTR⁺ neurons (red). **B3** Action potential frequency (AP freq.) of inhibited OTR⁻ pyramids upon optogenetic stimulation of OTR⁺ neurons ($n = 7$ cells, sections obtained from 4 rats), $p < 0.0001$. **C** cFos expression after chemogenetic activation of ILC OTR⁺ interneurons. **C1** Injection of a Cre-dependent rAAV expressing Gq-GFP in all OTR⁺ interneurons (green) in the ILC; perfusion 100 min post-injection of CNO or saline; immunohistochemical staining for NeuN (magenta) and cFos (marker for neuronal activity, red). OTR⁺ interneurons were activated by CNO but not saline (white arrows). Scalebar 100 μm **C2** Chemogenetic activation of ILC OTR⁺ interneurons reduced cFos expression in L2/3, $p < 0.0001$ (post hoc, L2/3). **C3** Local reduction or increase of cFos expression in the BLA or NAc, respectively. No significant change of cFos expression in the VTA, PAG, septum, or Summ (sections obtained from $n = 8$ or 4 animals per group). Statistical significance is indicated as * $p < 0.05$, ** $p < 0.01$, *** $p < 0.001$, **** $p < 0.0001$. Statistical significance of post-hoc tests is indicated as # $p < 0.05$, ## $p < 0.01$, ### $p < 0.001$, #### $p < 0.0001$. Error bars show mean±sd. Box plots show the median, 25th–75th percentiles, and whiskers from minimum to maximum. For details on statistical tests please refer to Supplementary Data 1.

experiments suggest that endogenous OT release in the ILC activates layer 2/3 OTR⁺ interneurons, which inhibit neighboring OTR⁻ downstreaming principal cells.

## OTR⁺ interneurons inhibit the ILC−basolateral amygdala pathway

Having established the targeted inhibitory input from OTR⁺ interneurons to layer 2/3 OTR⁻ pyramidal neurons, our next objective was to investigate their potential downstream projection sites. Therefore, we analyzed the neuronal activity marker cFos expression in seven preselected brain regions downstream to the ILC[39], which were also reported to be functionally involved in modulation of social behaviors[40], upon the chemogenetic activation (rAAV-DLX-DIO-Gq-GFP) of ILC OTR⁺ interneurons. While control animals receiving saline injections showed no specific cFos expression in OTR⁺ interneurons ($0.69 \pm 0.37$ % cFos/GFP double-positive cells, $n = 375$ GFP⁺ neurons from 3 animals), animals receiving CNO showed cFos signal in these

GFP⁺ cells ($99.91 \pm 0.15$ % cFos/GFP double-positive cells, $n = 424$ GFP⁺ neurons from 3 animals) (Fig. 6C1). To allow for unbiased quantification of neuronal cFos expression, we employed a fully automated pipeline based on the Imaris software which analyzed cFos signals in more than 500,000 neurons across the seven different brain regions. Activating OTR⁺ inhibitory interneurons in the ILC resulted in an overall decrease of cFos expression in the ILC (main treatment effect, $p = 0.0004$), with a significant decrease in layer 2/3 (post-hoc layer 2/3, $p < 0.0001$, post-hoc layer 5/6 $p = 0.0604$) (Fig. 6C2), indicating extensive inhibition of OTR(GFP)-negative principal cells (Fig. 6B), particularly in layer 2/3 where the main population of OTR⁺ interneurons resides. As for the analyzed downstream regions, cFos expression was not significantly changed in the periaqueductal gray (PAG), lateral septum, supramammillary nuclei (Summ), or ventral tegmental area (VTA). However, it was significantly reduced in the basolateral amygdala (BLA) ($15.4 \pm 4.9$% vs. $8.7 \pm 2.0$%; $p = 0.0185$) (Fig. 6C3). This suggests that experimental OT release in the ILC

ultimately reduces the activity of BLA neurons. Notably, in the same paradigm, we observed a significant increase in cFos expression in the nucleus accumbens (NAc) ($10.7 \pm 2.3\%$ vs. $22.2 \pm 7.9\%$; $p = 0.0084$), hinting at differences between the possible functional dichotomy of BLA and NAc projections from the ILC. This may suggest that ILC output neurons projecting to the NAc are not directly (monosynaptically) inhibited by layer 2/3 OTR⁺ interneurons.

To test this hypothesis, we performed viral retrograde tracing of neurons projecting to the BLA or to the NAc by injection of a retrograde GFP-expressing rAAV (retro-rAAV-hSyn-GFP) into either BLA or NAc. The same animals were injected into the ILC with a Cre-dependent rAAV expressing mCherry in OTR⁺ interneurons (rAAV-DLX-DIO-mCherry) to visualize the overall topography of back-labeled projecting cells in the ILC in relation to the OTR⁺ cell population (Fig. 7A1, B1). Injection sites in the BLA or NAc were marked by co-injection of a small amount of soluble fluorescent beads in the tip of the injection pipette (Supplementary Fig. S7A). A substantial number of BLA-projecting neurons was found within the "inhibitory sleeve" formed by the vertical axonal cartridges of OTR⁺ Chandelier cells in layer 2/3 ($18.1 \pm 7.4$ neurons per hemisphere in 50 µm sections) as well as in the deeper layers of the ILC ($31.8 \pm 12.1$ neurons per hemisphere in 50 µm sections) (Fig. 7A2, and Supplementary Fig. S7C). In contrast, retrogradely labeled NAc-projecting neurons were found exclusively in the deeper layers of the ILC, but not within the "inhibitory sleeve" of layer 2/3 (Fig. 7B2). Furthermore, detailed morphological analysis of BLA-projecting neurons revealed that OTR⁺ interneurons form direct axo-axonic contacts with BLA-projecting pyramidal neurons in layer 2/3 (Fig. 7C, Supplementary Fig. S7B), but not with NAc-projecting pyramidal neurons in layer 5/6. Thus, we showed anatomical evidence of a direct and local OTR⁺-neuron-mediated inhibition of BLA-projecting neurons in layer 2/3 of the ILC, and focused on investigating this ILC-BLA downstream pathway.

Following this clear anatomical distinction, we next verified inhibitory action of ILC OTR⁺ interneurons on OTR⁻ pyramidal neurons projecting to the BLA. In the first set of experiments, we injected a retrograde rAAV expressing GFP into the BLA (retro-rAAV-hSyn-GFP) and, in parallel, a Cre-dependent rAAV expressing ChR2 in interneurons (rAAV-DLX-DIO-ChR2-mCherry) into the ILC of OTR-Cre-rats. After a 3-week expression period, OTR⁺ interneurons were optogenetically activated by BL stimulation in an ex vivo experimental setup, while BLA-projecting OTR⁻ pyramids were recorded in the whole cell mode (Fig. 7D1). Neurons were patched only if they were both GFP-positive (BLA-projecting) and mCherry-negative (OTR⁻ neurons) (Fig. 7D2). Recorded neurons were identified as principal cells based on their typical discharge pattern and morphology after post-recording staining procedure (Supplementary Fig. S7D and Fig. 7D2). Notably, all recorded neurons ($n = 11$) responded with inhibition to BL induced OTR⁺ interneuron activation (Fig. 7D3 for responses of BLA-projecting OTR⁻ pyramids, Supplementary Fig. S7E for representative trace verifying activation of OTR⁺ interneuron by BL).

Finally, we chemogenetically targeted the BLA-projecting population in the ILC, in order to verify its involvement in social behavior. We injected a rAAV expressing the excitatory DREADD Gq-mCherry in a Flp-dependent manner (rAAV-hSyn-FRT-Gq-mCherry) into the ILC of female rats, while simultaneously injecting a retro-rAAV expressing Flp (retro-rAAV-Ef1a-Flp) into the BLA (Fig. 7E1). With post-hoc verification of injection sites we could observe cell bodies in the ILC as well as dense fibers in the BLA (Fig. 7 E2). Animals that received CNO injection prior to a free social interaction session spent significantly less time investigating the social partner ($31.9 \pm 18.1$ s vs. $19.3 \pm 15.0$ s; $p = 0.042$) (Fig. 7 E3, Supplementary Fig. S7 F). Thus, we could show that activating BLA-projecting neurons had the same effect on social interaction as inhibiting OTR⁺ interneurons in the ILC (Fig. 5 B, right), supporting our hypothesis of OTR⁺ interneuron mediated inhibition of BLA-projecting neurons driving sociability in female rats.

## Discussion

Our study demonstrates that OT release within the mPFC directly facilitates social behavior in female rats. We found that the ventral part of the mPFC (ILC) is preferentially innervated by OT fibers and enriched in OTR-expressing neurons, rendering it a major site of OT action. Intriguingly, we observed OT neuron axons across all cortical layers with the highest density in layer 5/6, while the majority of OTR⁺ neurons resides in layer 2/3. This apparent mismatch can be explained by the anatomical positioning of the ILC adjacent to the forceps minor of the rat corpus callosum, and long-range axons coursing within callosal/white-matter tracts that frequently send collaterals towards deep cortical layers[29]. Consistent with this architecture, OT projections arising from hypothalamic nuclei travel through major forebrain fiber bundles and elaborate fine collaterals in deep layers of the mPFC, producing higher apparent OT-fiber density in layers 5/6 than in superficial layers. This difference is especially apparent when using virus-based reporters, which reveal thin-caliber OT axons and terminals that are often below the detection threshold of classical peptide immunohistochemistry. The OT innervation detected in layer 1 aligns with the presence of dendritic arbors of layer 2/3 OTR⁺ Chandelier cells extending into the superficial cortex, a site of long-range modulatory inputs and distal dendritic integration[41]. Given the capacity of OT to act via local diffusion at very low concentrations, OT released from sparse layer-1 afferents could effectively engage OTR⁺ interneurons in layers 2/3 despite the lower local fiber density[42]. Notably, utilizing the novel GRAB OT-sensor[31] ex vivo, we showed OT binding upon optogenetic stimulation of OT fibers in the ILC, demonstrating local axonal OT release in this brain region.

In vivo, a similar optogenetic stimulation of OT fibers in the ILC increased social interactions between female rats. These findings are consistent with the involvement of OT in the modulation of social behavior through its actions in various brain regions[7,8]. Moreover, our results extend the recently reported analgesic OT input to the mPFC[43], revealing its role in facilitating social interactions. Taken together with the prosocial effects of hypothalamic OT neuron stimulation[4,23], this work supports the role of OT in promoting social motivation and interaction.

We showed that OTR⁺ neurons in the ILC are intrinsically active during social behavior, with activity specific to interaction events initiated by the test rat, not the social partner (e.g., sniffing vs. being sniffed). This suggest their role in driving social interactions and may further indicate that ILC OTR⁺ neurons are crucial in the decision-making processes underlying social executive functions of the mPFC[26]. Population imaging in the mouse mPFC shows that ensembles preferentially encode social olfactory cues and refine with experience[25]. This is consistent with a role for ILC circuits in driving social actions rather than merely reflecting sensory processing. It is tempting to propose a pivotal role of OT signaling in this process, yet a direct link between acute OT release and the ILC OTR⁺ calcium activity requires further targeted testing. Nevertheless, we proved the necessity of ILC OTR signaling for social interaction by local depletion of OTRs. Our in vivo CRISPR/Cas9-mediated OTR targeting should be interpreted as a partial loss-of-function rather than a complete receptor knockout. First, viral transduction and CRISPR editing are inherently mosaic, so a fraction of OTR-expressing neurons likely retain intact receptors. Second, indel heterogeneity may generate alleles with partial activity. Third, residual receptor functionality unrelated to high-affinity ligand binding (for example heteromerization/dimerization) may persist and maintain some downstream signaling[34]. These features likely explain the residual responsiveness to TGOT we detected ($n = 1$ neuron, Supplementary Fig. S3F) and indicate that our behavioral effects probably underestimate the full impact of abolishing OTR signaling in the ILC. Our results demonstrate that OTR⁺ neuron activation in the ILC facilitates sociability via axonal OT release[5].

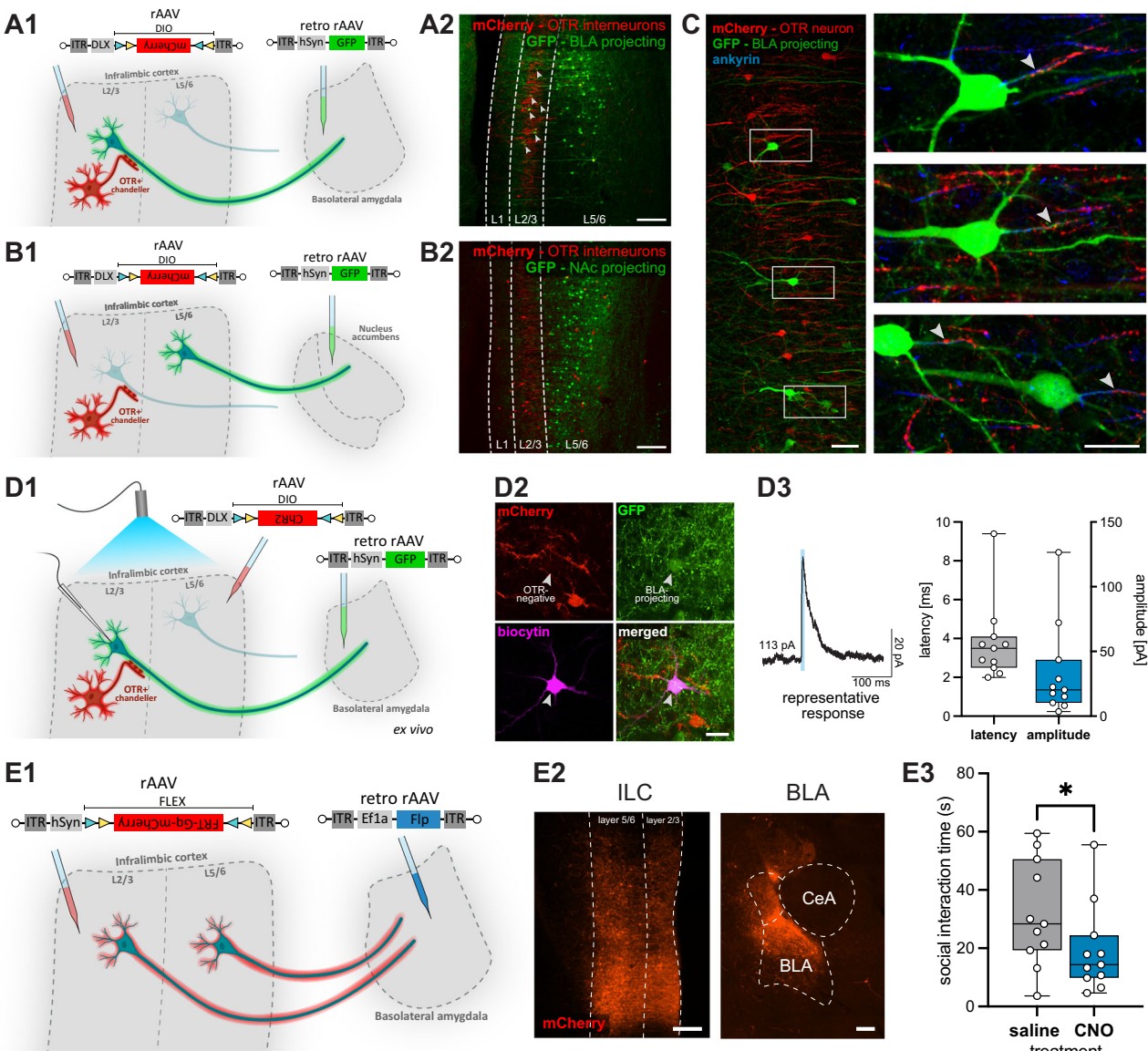

**Fig. 7 | OTR⁺ interneurons modulate neuronal activity in the BLA. A** Retrograde tracing of BLA-projecting neurons in the ILC. **A1** Injection of a retrogradely GFP-expressing rAAV into the BLA and a Cre-dependent mCherry-expressing interneuron-specific rAAV into the ILC. **A2** BLA-projecting neurons located in L5/6, as well as within the "inhibitory sleeve" of OTR⁺ interneurons in L2/3 (white arrows). Scalebar 200 μm. **B** Retrograde tracing of NAc-projecting neurons in the ILC. **B1** Injection of a retrogradely GFP-expressing rAAV into the NAc and a Cre-dependent mCherry-expressing interneuron-specific rAAV into the ILC. **B2** NAc-projecting neurons located only within L5/6. Scalebar 200 μm. **C** Representative axo-axonic contacts (white arrows) of OTR⁺ interneurons (red) on BLA-projecting pyramidal neurons (green). Immunohistochemical staining of the axon-initial segment for ankyrin (blue). Scalebar 50 μm overview, 25 μm zoom. **D** Electrophysiological patch-clamp recording of OTR⁻ neurons projecting to the BLA. **D1** Injection of a rAAV expressing ChR2 in a Cre-dependent manner in OTR⁺ interneurons of OTR-Cre animals, and injection of a retrograde rAAV into the BLA. Ex vivo patch-clamp recording of BLA-projecting neurons with BL activation of OTR⁺ interneurons. **D2** Representative image of a biocytin-filled OTR⁻ BLA-projecting pyramidal neuron. Scalebar 20 μm. **D3** Representative averaged trace of a whole-cell patch-clamp recording (voltage clamp, holding voltage −60mV) depicting an inhibitory response of a BLA-projecting neuron within L2/3. Latency and amplitude of recordings for all neurons (*n* = 11 recorded neurons). **E** Social interaction after chemogenetic activation of BLA-projecting neurons originating in the ILC. **E1** Injection of a retrogradely Flp-expressing rAAV into the BLA and a Flp-dependent (Frt) Gq-mCherry-expressing rAAV into the ILC to chemogenetically activate BLA-projecting ILC neurons. **E2** Representative injection site showing expression of rAAV-hSyn-FRT-Gq-mCherry in ILC (cell bodies) and BLA (projecting fibers). Scalebar 250 μm. **E3** Chemogenetic activation of BLA-projecting neurons decreases social interaction time (*n* = 11 animals). Statistical significance is indicated as * *p* < 0.05, ** *p* < 0.01, *** *p* < 0.001, **** *p* < 0.0001. Box plots show the median, 25th-75th percentiles, and whiskers from minimum to maximum. For details on statistical tests please refer to Supplementary Data 1.

We further explored the local ILC neuronal circuit receptive to OT using multiplex fluorescent in situ hybridization and neuroanatomical analyses. Our findings revealed that OTR expression in the ILC is restricted to a small subset of neurons, underscoring their significant role in social behavior. This finding is in line with our previous work in the PVN, where we showed that a small population of parvocellular OT neurons modulates inflammatory pain processing[44]. Taken together, this highlights the potent influence of OT in central circuits, even under conditions of low concentration or limited OTR expression. Additionally, we found that the vast majority of OTR⁺ neurons in the ILC are GABAergic interneurons, located predominantly in layers 2/3. This aligns with mouse mPFC reports emphasizing a predominantly

interneuronal OTR population[20,21]. By contrast, besides the inter-neuron population, Tan, Singhal and colleagues[22] identified a sub-stantial VGluT1[+] OTR[+] population in the mouse mPFC. We suspect these differences reflect a combination of species (rat vs. mouse), subregion (ILC vs. broader PFC/PL), sex (female rats), and methodo-logical factors (driver lines, reporter/probe sensitivity, and positivity thresholds). Notably, our OTR-Cre specificity was validated by RNA-scope (Cre↔OTR mRNA overlap ~83–93%) and pharmacology (all OTR[+] neurons marked with GFP responded to bath application of TGOT in our patch-clamp experiments). Based on morphological and electro-physiological features, we found that the majority of these OTR[+] neurons are Chandelier cells. These GABAergic interneurons are powerful inhibitors of output principal cells in cortical circuits[45]. They form unique axo-axonic synapses, quenching the generation of action potentials at the axon initial segment of pyramidal neurons. This unique inhibitory efficacy of Chandelier cells enables precise control over the activity of output pyramidal neurons and overall network excitability. The functional activation of these interneurons in our experiments led to significant behavioral changes, suggesting that GABAergic modulation within the ILC is crucial for social behavior regulation. This finding is consistent with the known role of GABAergic interneurons in shaping cortical output and influencing behavior[46]. Notably, we did not observe any effects on anxiety behaviors when activating ILC OTR[+] interneurons. This is in line with work from Li, Nakajima and colleagues[21], who showed that activating OTR-expressing interneurons in the mPFC reduces anxiety-like behavior in males but not females (with OTR loss producing the opposite, male-specific anxiogenic effect). In contrast, the same manipulation enhances social behavior in females but not males, revealing com-plementary sex-specific routing of OT's influence on anxiety vs. sociability, a sex difference yet unexplored in rats.

Considering that the activation of ILC OTR[+] interneurons increased investigation of the social, but not the food corner in the social vs. food paradigm, the observed change can be interpreted as facilitation of social approach under competing motivation rather than an anorexigenic effect. This choice-specific engagement supports a selective action of OTR[+] neuron modulation, rather than a global change in arousal or locomotion. The observed shift towards increased social investigation even in the presence of food after 24hrs of food deprivation is consistent with the hypothesis of mutually inhibitory ensembles in the mPFC postulated by Gangopadhyay and colleagues[47]. In line with their hypothesis and based on our results, we suggest that social decision-making is facilitated by specific neuronal ensembles within the ILC that are selectively activated by social stimuli and modulated by OT action. Thus, specific OTR[+] interneurons within the ILC promote social behavior and contribute to the encoding of socially relevant information and biasing decisions toward prosocial beha-viors. In contrast, certain non-social ILC ensembles are activated by non-social stimuli and are involved in processing individual-centric information. Independent of OT signaling, work in rats shows that inactivation of the ILC broadly impaired both active and inhibitory avoidance and suppressed inappropriate reward seeking[48]. Consistent with this, a comprehensive review concluded that ILC function is highly context-dependent, varying with task demands and the specific neuronal ensembles recruited[49]. In our social vs. food choice para-digm, activation of OTR[+] ILC interneurons selectively increased social investigation without generalized arousal, consistent with the pro-posed role of the ILC in resolving competing motivations by sup-pressing non-social drives when social cues are behaviorally salient. Intriguingly, the prosocial effect of ILC OTR[+] interneuron modulation emerges stronger under hunger. This suggests that OT signaling in the ILC acts not necessarily as a tonic enhancer of sociability, but as a context-dependent biasing mechanism that promotes social engage-ment specifically when motivational conflict is strong.

We also investigated subcortical regions downstream to OTR[+] interneurons focusing on the BLA and the NAc, two areas that are well-known to be involved in socially relevant functions including fear, reward, and social interaction[39,50,51]. Chemogenetic activation of OTR[+] ILC neurons in our experiment led to downregulation of cFos expres-sion in the BLA and increased cFos expression in the NAc. We propose that the observed reduction in the number of cFos-positive cells in the BLA is caused by the ILC OTR[+] interneurons-mediated inhibition of ILC principal cells providing excitatory input to the BLA. Among the mPFC subdivisions, the ILC in particular is known to target a dense population of GABAergic interneurons in the BLA[52] and the BLA in turn densely innervates the NAc[53]. We hypothesize that the observed cFos upregu-lation in the NAc might be caused by the suppression of BLA inter-neuron activity, thereby increasing the excitatory signaling from the BLA to the NAc. As mapping circuits via cFos activity patterns comes with the caveat of spatial and temporal ambiguity, we further investi-gated the OTR-ILC-BLA/NAc circuit both anatomically and electro-physiologically. Retrograde tracing following injections in the BLA and NAc revealed BLA-projecting principal cells in the ILC that were embedded within a dense "inhibitory sleeve" and NAc-projecting prin-cipal cells only in deeper cortical layers. Additionally, we showed OTR[+] interneurons forming axonal cartridges at the initial axonal segment of neurons projecting to the BLA which was not observed in the case of NAc-projecting principal cells. This was further confirmed by ex vivo studies which showed inhibitory monosynaptic coupling between OTR[+] interneurons and BLA-projecting principal cells. Notably, a study by Lu, Tucciarone and colleagues[54] demonstrated that a subset of layer 2 Chandelier cells in the PrL cortex selectively innervates BLA-projecting principal cells ($_{BLA}$PCs) over contralateral-cortex-projecting principal cells ($_{CC}$PCs), while receiving preferential input from $_{CC}$PCs, thereby establishing a directional inhibitory module that gates BLA-related output. Interestingly, they also found $_{BLA}$PCs in the same laminar layers as Chandelier cells, while $_{CC}$PCs were located in deeper layers, resem-bling our finding of BLA- vs. NAc-projecting neurons. This spatial coin-cidence suggests that OTR[+] interneurons are well-positioned to modulate $_{BLA}$PCs ensembles. While their study did not indicate a behavioral role of this neuronal subset, our work suggests its prosocial role. While unlikely, considering the absence of connections to NAc-projecting neurons we found in this study, we cannot exclude the possibility of OTR[+] Chandelier cells projecting onto neurons in deeper layers. It has been reported previously that Chandelier cells in layer 2/3 innervate pyramidal neurons in cortical layers 2, 3, and 5[55,56], and that while they have mostly vertically oriented axonal segments in layer 2/3, individual long descending axons reached and arborized in layer 6[57]. However, in our experiments, chemogenetic activation of ILC OTR[+] interneurons did not alter cFos expression in deep ILC layers, suggest-ing that their effects are largely confined to layer 2/3.

Importantly, several studies indicate the ILC as a key node for social approach via amygdala-directed outputs. In mice, BLA-projecting ILC neurons are preferentially recruited by social cues, and normal sociability requires intact ILC-BLA signaling[58]. In hamsters, chemogenetic activation of the ILC-BLA connection during social defeat confers resilience and reduces defeat-associated behavior while lowering BLA c-Fos[59]. Our finding that activating OTR[+] interneurons in the rat ILC increases sociability, coupled with reduced BLA cFos expression, aligns with a model in which ILC microcircuits gate amygdala-directed output to favor social approach under challenge. This is further strengthened by the antisocial effects of stimulation of ILC-BLA-projecting pyramidal neurons we observed.

We acknowledge the possibility that, beyond the ILC-BLA circuit described here, other OT-dependent pathways originating in the mPFC may underlie distinct aspects of behavior. There are reports that prin-cipal neurons in the mPFC are sensitive to bath application of OT[60] or express OTRs [Ref. 22 and this study]. In mice, numerous OT-sensitive

principal cells can be found throughout the entire mPFC, including the infralimbic area[61], and they are located in both superficial and deeper layers. On the contrary, our results in rats report considerably fewer (9.14% in situ hybridization data, Fig. 4A) principal cells that express OTRs and these cells are located almost exclusively in the deeper layers, particularly outside of the layer 2/3 "inhibitory sleeve". This may explain why we did not observe social novelty preference alterations following the activation of OTR⁺ interneurons (Supplementary Fig. S5 C) as previously found in male mice after manipulation of the direct mPFC-BLA pathway[22]. In addition to the differences in species and sex, Tan and colleagues showed that these principal OTR⁺ neurons specifically alter social memory, while having no particular effect on general sociability. This could indicate the existence of two independent, and potentially species- and sex-dependent OT-modulated pathways for sociability (interneurons) and social memory (pyramidal neurons). A distinctive organization of the OT systems in different mammals, including mice and rats, has been identified at the anatomical and behavioral level[62–64], reflecting species-dependent processing of social information and subsequent selection of optimal social strategy[65–68].

In conclusion, our study identifies a critical role for OT signaling within the ILC in facilitating social behaviors in female rats (Supplementary Fig. S7G). Activation of GABAergic OTR⁺ interneurons in particular appears to play a significant role in modulating social interaction. These findings contribute to our understanding of the neural mechanisms underlying social behavior and highlight the potential of targeting OT pathways for therapeutic interventions in social deficits[69], as the PVN/SON-to-mPFC OT signaling pathway is conserved throughout evolution and exists in several primate species including humans[70,71].

## Methods

### Animals

Female Sprague-Dawley rats (3–8 weeks, Janvier Labs) and transgenic OTR-Cre rats bred in-house with a Sprague-Dawley background (Central Institute for Mental Health, Mannheim) were used. Adult virgin females (8–16 weeks) were included in anatomical, functional, and behavioral experiments. Age is provided as a range, as experiments always range over the course of several weeks, starting with stereotaxic injection, followed by 3 weeks of expression time, then behavioral experiments, and closed with sacrificing the animal and anatomy and/or ex vivo experiments. The age for injection was always 8–10 weeks, while the age for all social interaction and food vs. social behavior experiments were 11-16 weeks. For single-day behavioral tests (free social interaction, between-subject design), only animals in diestrus were included, as described previously by Tang, Benusiglio and colleagues[4]. Rats were group-housed under a 12-h light/dark cycle (lights on 07:00, 22–24 °C, 50 ± 5% humidity) with food and water ad libitum unless stated otherwise. All experiments were conducted under animal license G-193/20 and G-55/23 authorized by the German Animal Ethics Committee of the Baden Württemberg (Regierungspräsidium Karlsruhe, Germany) and in accordance with the German law, under license APAFIS #46161- 2024011918045241_v2 from the French Ministry, and under EU regulations.

### Transgenic OTR -Cre Rats.
Transgenic OTR-Cre rats[32] express Cre recombinase in all OTR-expressing neurons, enabling cell type–specific expression of Cre-dependent (double-floxed inverted orientation, DIO) rAAVs. Animals were generated by breeding heterozygous Cre⁺ males with wild-type females, and offspring were genotyped using a PCR-based protocol to identify heterozygous Cre⁺ or homozygous Cre⁻ animals.

### Viral vectors

If not stated otherwise, plasmids for the production of rAAVs were generated as described previously[5]. Recombinant adeno-associated viruses (rAAVs) of serotype 1/2 were used following previously established methodologies[5]. HEK293T cells for viral production were obtained from Addgene (catalog number 240073). The OT-promoter controlled rAAV-OTp-ChR2-mCherry, rAAV-OTp-Venus, and rAAV-OTp-ChrimsonR-mScarlet, as well as the Cre-dependent rAAV-Ef1a-DIO-GFP, rAAV-Ef1a-DIO-mCherry, rAAV-Ef1a-DIO-GCaMP6s, and rAAV-Ef1a-DIO-ChR2-mCherry, were produced in-house (titers between $10^9$ and $10^{10}$ genomic copies/μl). The plasmid for rAAV-hSyn-FLEX-FRT-Gq-mCherry (plasmid #161575) was obtained from Addgene and the viruses were produced in-house (titer > $10^9$ genomic copies/μl). rAAVs for interneuron targeting (rAAV-DLX-DIO-GFP (number v277), rAAV-DLX-DIO-mCherry, rAAV-DLX-DIO-Gq-GFP (number v592), rAAV-DLX-DIO-Gi-GFP (number v703), rAAV-DLX-DIO-ChR2-mCherry (number v317)) were purchased from the ETH Zurich Viral Vector Facility (serotype DJ, titers $6.5 \times 10^{12}$ – $8 \times 10^{12}$ vg/ml). The serotype 9 rAAVs used for OTR knockdown (rAAV-hSyn-spCas9, rAAV-u6-gRNA(OTR)-hSyn-mCherry, rAAV-u6-gRNA(lacZ)-hSyn-mCherry) were obtained in collaboration with Larry Young and Arjen Boender (titers $3.0 \times 10^{10}$ and $1.5 \times 10^{10}$ genomic copies/μl for gRNA and Cas9 AAVs, respectively). The AAV-CRISPR/Cas9 plasmids to produce the viruses used in this study are available through Addgene: pAAV-hSyn-spCas9 (#231404), pAAV-u6-gRNA(OXTR.1)-CMV-eGFP, (#194015), pAAV-u6-gRNA(CTRL)-CMV-eGFP (#194017). Retrograde rAAVs (retro-rAAV-hSyn-GFP (plasmid #114213), retro-rAAV-Ef1a-Flp (plasmid #55637), titer ≥ $7 \times 10^{12}$ vg/mL) were purchased from Addgene. The plasmid for GRAB-OT1.9 was obtained from the Yulong Li laboratory.

### Stereotactic injections

Rats were anesthetized with either isoflurane (3-2% in 1 L/min $O_2$) or ketamine/xylazine (75 mg/kg and 5 mg/kg, respectively). Viral solutions (300 nl per hemisphere) were delivered at 150 nl/min using glass micropipettes. Injection coordinates (mm from bregma) were: PVN (AP − 1.8, ML ± 0.3, DV − 8.0), SON (AP − 1.8, ML ± 1.2, DV − 9.25), ILC (AP + 3.0, ML ± 0.5, DV − 4.6), BLA (AP − 3.0, ML ± 5.0, DV − 8.4 to −9.0), and NAc (AP + 1.2, ML ± 1.0, DV − 7.8 to −8.2). Following all experiments, injection sites and transgene expression were confirmed histologically.

### Behavioral tests

Behavioral tests were performed in an odor-resistant square arena (60 × 60 × 60 cm) under dim light ( < 20 lux, SO 200 K lux-meter, Sauter). Rats were habituated to the arena for 15 min the day before testing, and the arena was cleaned with 80% ethanol between trials. Experimental and stimulus rats were housed separately.

**Free social interaction.** Experimental rats were paired with novel, age- and weight-matched conspecifics for 5 min. Animals were placed in diagonally opposite corners at trial start. For optogenetic experiments, 470 nm light (10 mW/mm², 30 Hz, 10 ms pulses) was delivered during the first 2 min of the session. For chemogenetic activation, the DREADD agonist CNO (3 mg/kg) was injected 40 min prior to testing. Behaviors (anogenital/non-anogenital sniffing, following, approach, toy investigation) were manually scored using BORIS software (https://www.boris.unito.it) by two independent, blinded experimenters. Discrepancies were resolved by consensus. The order of animals tested was randomized. Additionally, the order of free social interaction or toy investigation was randomized. The assignment of interaction partners to the test animals was randomized, with the constraint that interaction partners had to be unknown.

**Food vs. social preference test.** The arena was equipped with triangular corner dividers containing mesh fronts, allowing indirect investigation of either an unfamiliar juvenile female rat or standard food pellets. Each 5-min test session followed a 15-min habituation period. The test was repeated after 24 h of food deprivation. Investigation time

was analyzed using Ethovision XT v11.5 (Noldus) by measuring nose position within predefined zones. The following parameters were quantified: total investigation time, number of bouts, and mean bout duration (a bout was defined as each re-entry after leaving the zone). Control experiments used the same setup, but only food was presented in one corner, with the other left empty. The positions of the corners were assigned randomly, the order of animals tested was randomized.

## Ex vivo electrophysiology and imaging

**Slice Preparation.** Brain slices were prepared 3–6 weeks after viral injections. Animals were anesthetized using isoflurane or injection of ketamine/xylazine, and decapitated. For experiments 1–4 (1: TGOT application in current clamp, 2: TGOT application in voltage clamp, 3: paired-patch recordings, 4: optogenetic stimulation of OTR[+] neurons with recordings from OTR[−] pyramidal cells), slice preparation and visualization followed established methods[72,73]. For experiment 5 (GRAB-OT1.9 sensor imaging combined with optogenetic stimulation of OT axons in the ILC), slice preparation followed Iwasaki, Lefevre[32]. For experiment 6 (optogenetic stimulation of OTR[+] interneurons with recordings from BLA-projecting pyramidal neurons), rats were anesthetized with isoflurane, decapitated, and brains rapidly removed into chilled, carbogenated ACSF containing (in mM): 92 choline chloride, 2.1 KCl, 1.2 NaH$_2$PO$_4$, 30 NaHCO$_3$, 20 HEPES, 25 glucose, 5 Na-ascorbate, 2 thiourea, 3 Na-pyruvate, 5 N-acetyl-L-cysteine, 10 MgSO$_4$, and 0.5 CaCl$_2$, pH 7.3–7.4, 290–310 mOsm. Coronal vmPFC sections (400 µm) were cut using a Leica VT1200 vibratome, incubated for 10 min at 32 °C in the same solution, and then transferred to recovery ACSF containing (in mM): 92 NaCl, 2.1 KCl, 1.2 NaH$_2$PO$_4$, 30 NaHCO$_3$, 20 HEPES, 10 glucose, 5 Na-ascorbate, 2 thiourea, 3 Na-pyruvate, 5 N-acetyl-L-cysteine, 1.3 MgSO$_4$, and 2.4 CaCl$_2$, pH 7.3–7.4, 290–310 mOsm, where they were kept for at least one h before recording. After recordings, selected slices were fixed overnight in 4% PFA at 4 °C and immunostained to examine recorded neurons. Sections were incubated with chicken anti-GFP (1:10,000), rabbit anti-RFP (1:2,000), and Streptavidin-iFluor750 (1:4000) in 0.5% Triton X-100/PBS for 72 h at 4 °C, washed in PBS, and then incubated with Alexa Fluor–conjugated secondary antibodies (1:2000) for 24 h. Sections were mounted with Mowiol and imaged on an Olympus VS200 slide scanner and Leica Stellaris 5 confocal microscope.

**Patch-clamp recordings.** Slices were transferred to a recording chamber on a Nikon Eclipse FN1 microscope and perfused at 2 ml/min with carbogenated ACSF (32 °C). For experiments 1–4 the ACSF contained (in mM): 125 NaCl, 25 NaHCO$_3$, 2.5 KCl, 1.25 NaH$_2$PO$_4$, 2 CaCl$_2$, 1 MgCl$_2$, and 25 glucose, pH 7.2, 290–300 mOsm. For experiment 5 the ACSF contained (in mM): 118 NaCl, 25 NaHCO$_3$, 3 KCl, 1.2 NaH$_2$PO$_4$, 2 CaCl$_2$, 1.3 MgSO$_4$, and 10 glucose, pH 7.4, 290–300 mOsm.

Patch pipettes (5–7 MΩ) were filled with experiment-specific internal solutions. For experiments 1 and 4 the solution contained (in mM): 144 K-gluconate, 4 KCl, 4 Mg-ATP, 10 phosphocreatine, 0.3 GTP, and 10 HEPES, pH 7.3 with KOH, 293 mOsm. For experiment 2 it contained 144 Cs-gluconate, 4 CsCl, 4 Mg-ATP, 10 phosphocreatine, 0.3 GTP, and 10 HEPES, pH 7.3 with CsOH, 293 mOsm. For experiment 3 it contained 110 K-gluconate, 30 KCl, 8 NaCl, 4 Mg-ATP, 10 phosphocreatine, 0.3 GTP, and 10 HEPES, pH 7.3 with KOH, 293 mOsm. For experiment 6 it contained 145 K-gluconate, 2 MgCl$_2$, 4 Na$_2$ATP, 0.4 Na$_3$GTP, 5 EGTA, and 10 HEPES, pH 7.3 with KOH, 290–300 mOsm, and 0.05% biocytin for post-hoc neuron identification.

Whole-cell configuration was achieved by gentle negative pressure. Signals were recorded with an EPC 10 USB amplifier using PATCHMASTER software (HEKA Elektronik), low-pass filtered at 2.9 kHz, digitized at 50 kHz.

**Experimental protocols and analysis.** For experiment 3 presynaptic neurons were stimulated with a 10 Hz train of three suprathreshold current pulses every 7 s, and postsynaptic responses were averaged over 50–100 sweeps. The success rate of synaptic coupling was calculated as the inverse (in %) of the number of neurons tested before a connection was found. In experiment 2 spontaneous EPSCs and IPSCs were recorded at −90 mV (E$_{GABA}$) and 0 mV (E$_{AMPA}$), respectively, and detected using a Wiener deconvolution algorithm in MATLAB (deconvwnr function) with τ_onset/τ_offset set to 4/15 ms for IPSCs and 2.5/10 ms for EPSCs. Peaks were identified in a 50 ms sliding window using dynamic and absolute thresholds, and PSC features were extracted from traces around each peak. In experiment 6 only layer 2/3 ILC pyramidal neurons projecting to the BLA were analyzed. Cells were voltage-clamped at −60 mV and stimulated with ten 470 nm light pulses (10 ms, -10 mW) at 30 s intervals, which were later averaged.

Data were analyzed with IGOR PRO (Wavemetrics), Signal (CED), and MATLAB. Electrophysiological results are reported as mean ± SD. Cumulative distributions were compared with the Kolmogorov–Smirnov test.

**GRAB OT1.9 oxytocin sensor imaging.** Animals were euthanized with an overdose of ketamine (100 mg/kg) and xylazine (20 mg/kg), and decapitated, followed by brain extraction. Coronal brain slices containing the ILC were allowed to recover for at least one h in oxygenated ACSF at room temperature after preparation. Expression of rAAV-OTp-ChrimsonR-mScarlet in the PVN and SON was confirmed using an X-Cite 110LED illumination system (XT-640-W). GRAB-OT1.9 imaging was performed on a Zeiss Axio Examiner microscope equipped with a CREST X-Light spinning disk confocal unit and a 20× water-immersion objective (NA 1.0). Images were acquired at 2 Hz using an optiMOS sCMOS camera (QImaging) controlled by MetaFluor software (v7.8.8.0). The OT1.9 sensor was excited at 475 nm (30 ms illumination), and ILC-projecting OT fibers were stimulated with 635 nm light at 30 Hz, 10 ms pulse width for 30 s after a 5 min baseline. Slices were then incubated in oxygenated ACSF with 1 µM Atosiban for ≥1 h before the second recording.

Fluorescence intensity was quantified using Fiji (ImageJ) and processed with a custom Python script. Signals were corrected for photobleaching by fitting a mono-exponential or polynomial decay model, and light-evoked OT release was quantified as the area under the curve (AUC) in a 50–350 s window post-stimulation, as previously described[32].

## In Vivo Fiber-Optic Experiments

**Optic Fiber Implantation.** For in vivo optogenetic experiments, rats were anesthetized with either isoflurane (3-2% in 1 L/min O$_2$) or ketamine/xylazine (75 mg/kg and 5 mg/kg, respectively), and bilaterally injected with rAAV-OTp-Venus or rAAV-OTp-ChR2-mCherry into the hypothalamic nuclei (SON, PVN) or with rAAV-Ef1a-DIO-ChR2-mCherry into the ILC, followed by bilateral optic fiber implantation above the ILC. Fibers (Ø 200 µm, NA 0.39, Thorlabs CFMC12L10) were cut to 5.5 mm length.

For in vivo fiber photometry, rats received bilateral injections of rAAV-Ef1a-DIO-GCaMP6s into the ILC and bilateral optic fiber implants (Ø 400 µm, NA 0.50, 5.5 mm, Doric Lenses) above the ILC.

Implantation coordinates for the ILC were AP + 3.1 mm, ML ± 1.52 mm, DV − 4.75 mm, with a 10° angle to allow connection of patch cables.

**Optogenetic Stimulation of OT Fibers.** Two 473 nm blue lasers (DreamLasers, Shanghai) were coupled to patch cables (Thorlabs RJPFF2-FC/PC) and controlled by a Master-8 pulse generator. Light was delivered at 10 mW/mm², 30 Hz, 10 ms pulses for 120 s, as described previously[5].

**Fiber photometry of ILC OTR neurons.** Rats were recorded for 10 min in a square open field arena. After 5 min habituation, an unfamiliar female rat or toy was introduced. Each rat underwent multiple sessions (44 total recordings across four animals). Data were acquired using a Doric Lenses photometry system and Doric Neuroscience Studio v6.1 software with 470 nm ($Ca^{2+}$ signal) and 405 nm (isosbestic) excitation at 20 Hz and 0.4 mW output power. Videos were recorded in parallel for behavioral annotation.

Raw signals were exported and preprocessed using the pMat toolbox[74]. High-frequency noise was removed by smoothing, and ΔF/F was calculated as (calcium signal − scaled isosbestic signal) / scaled isosbestic signal. The ΔF/F traces were then Z-scored relative to the mean and standard deviation of the baseline. For each behavioral event (anogenital sniffing, non-anogenital sniffing, toy sniffing), we compared the normalized signal averaged over 1 s before event onset to the signal averaged over the first 1 s after onset.

Statistical analysis was performed in R (Version 4.0.4; R Core Team, 2021) using RStudio (Version 1.4.1106, PBC; Team R, 2019). Linear mixed-effects models were fitted using the *lme4*[75] and *lmerTest*[76] packages, with type III ANOVA performed using Satterthwaite's method. Post hoc Tukey's multiple comparisons were conducted with the emmeans package[77], and the Kenward–Roger approximation for denominator degrees of freedom. Model assumptions were verified by inspection of residual plots.

The models compared Z-scored ΔF/F values before vs. during each behavioral event in a paired manner, with trial included as a random effect nested within session and sessions nested within animal. When recordings were performed bilaterally, hemisphere was added as a cross-nested random effect. Fixed factors included time in relation to trial onset (before vs. during the behavior), and in the case of comparison of active and received behaviors also behavior type. The model for the behavior towards the toy was fitted separately, as these were recorded from separate sessions.

## Histology

Rats were deeply anesthetized and transcardially perfused with PBS followed by 4% paraformaldehyde (PFA). Brains were post-fixed overnight in 4% PFA at 4 °C, sectioned coronally at 50 μm on a vibratome (Leica VT1000S), and collected into four series. Free-floating sections were processed using established protocols. Sections were incubated with primary antibodies in PBS with 0.5% Triton X-100 for 48–72 h at 4 °C, rinsed, and incubated with secondary antibodies for 24 h. Sections were mounted in Mowiol for imaging. Whole-section scans were acquired on an Olympus SLIDEVIEW VS200 slide scanner for overview images and verification of viral injection sites. Images of individual neurons, co-labeled markers, and OT fiber proximity were captured on a Leica Stellaris 5 confocal microscope in Z-stack mode.

Primary antibodies included anti-OT (mouse, 1:2000, gift of H. Gainer, clone: PS38), anti-RFP (Rockland 600-401-379, rabbit, 1:2000, polyclonal), anti-GFP (Abcam ab13970, chicken, 1:1000, polyclonal), anti-cFos (Cell Signaling 2250, rabbit, 1:500, clone: 9F6), anti-NeuN (Merck MAB377, mouse, 1:1000, clone: A60), anti-NeuN (Abcam ab177487, rabbit, 1:2000, clone: EPR12763), anti-Ankyrin (Millipore MABN466, mouse, 1:1000, clone: N106/36), anti-Parvalbumin (Abcam ab11427, rabbit, 1:5000, polyclonal), anti-Somatostatin (BMA T-4103, rabbit, 1:3000, polyclonal), anti-Calbindin (SWANT 300, mouse, 1:3000, clone: CB300), and anti-Calretinin (SWANT 6B3, mouse, 1:5000 clone: 6B3).

Secondary F(ab')₂ fragment donkey antibodies (Jackson ImmunoResearch, 1:1000, polyclonal) included anti-mouse Alexa 488-conjugated (715-546-150), anti-chicken Alexa 488-conjugated (703-546-155), anti-mouse Cy3-conjugated (715-166-150), anti-rabbit Cy3-conjugated (711-166-152), anti-mouse Alexa 647-conjugated (715-606-150), and anti-rabbit Alexa 647-conjugated (711-606-152).

**Fiber quantification analysis.** Densities of OT-positive fibers were quantified in serial sections collected between Bregma +3.72 and +2.52, corresponding to the rostral and caudal borders of the ILC ($n = 11$ sections from 5 animals). Anatomical borders between mPFC subdivisions were defined based on the Paxinos Rat Brain Atlas and previous cytoarchitectonic studies[78]. Whole-brain section scans containing Cg1, PrL, and ILC were captured with an Olympus VS200 slide scanner at 20× magnification and stitched into VSI image files. Fiber densities were quantified by counting crossing points of OT fibers in digitized images overlaid with a non-destructive grid of lines[79] in ImageJ. Crossing points were averaged per animal, normalized to region area, and presented as fibers/mm².

**Neuron quantification analysis.** For each animal, one of four series of 50 μm sections through the ILC was analyzed to quantify OTR-expressing neurons. Two analyses were performed: (1) the proportion of inhibitory vs. excitatory OTR-expressing neurons across the ILC, and (2) co-expression of cytochemical markers. Neurons were manually counted in ImageJ using touch-count mode, averaged, and presented as percent of total OTR-expressing neurons.

**Automated image analysis.** Proximity of OT fibers to OTR neurons was quantified in Imaris (Bitplane, *Oxford Instruments*). Green (OTR⁺ interneurons) and red (OTR⁺ neurons) channels were surface-reconstructed using a grain size of 0.6 μm, sphere diameter 0.5 μm, with threshold and volume parameters adjusted per image (volumes >1.7–5.25 μm³ included for green, >10 μm³ for red). Blue-channel OT fibers were reconstructed using a grain size of 0.44 μm and included if >3.85–10 μm³. Fiber segments were manually identified and assigned ID numbers for proximity mapping. GFP-, cFos-, and NeuN-positive cells were quantified in Imaris (Spots function, 10 μm diameter, quality threshold 1.21–1.35). The ILC was segmented manually into layers 1, 2/3, and 5/6 based on anatomical landmarks, and neurons were assigned accordingly. OT fiber distribution across ILC layers was calculated from surface reconstructions with a surface detail of 0.89 μm, absolute intensity threshold 2.2–6, and minimum volume 70 μm³.

## Western blot

Protein was extracted from OTR-Cre and WT rat brains using RIPA Lysis buffer, quantified with a BCA kit, separated on 4–12% NuPage Bis-Tris gels (Invitrogen), and transferred to nitrocellulose membranes. Membranes were blocked in 5% milk/TBS-T, incubated overnight at 4 °C with rabbit anti-oxytocin receptor (Thermo Fisher, 1:1000) and mouse anti-GAPDH (Abcam, 1:5000), followed by HRP-conjugated secondary antibodies. Bands were visualized by chemiluminescence and quantified with ImageJ.

## Multiplex fluorescent in situ hybridization

Expression profiles of OTR neurons in the ILC were characterized using RNAscope™ HiPlex (Advanced Cell Diagnostics, Hayward, CA, USA) on fresh-frozen 16 μm sections as previously described[80]. The following probes were used: vGlut1 (Rn-Slc17a7-T1, cat. no. 317001-T1), Cre (CRE-T2, cat. no. 312281-T2), vGAT1 (Rn-Slc32a1-T3, cat. no. 424541-T3), vGlut2 (Rn-Slc17a6-T5, cat. no. 31701-T5), and OTR (Rn-Oxtr-T6, cat. no. 483671-T6). Imaging was performed on a Zeiss Axio Imager M2 equipped with an automated z-stage and Axiocam 503 mono camera. Images were processed with Zen (3.1 blue edition and 3.0 SR black edition, Zeiss), CorelDraw 2020 (Corel), ImageJ[81], and HiPlex Image Registration Software v1.0 (ACD). OTR mRNA-positive cells were identified by DAPI-stained nuclei or cell-like mRNA distribution and manually counted in ImageJ (Cell Counter), following Iwasaki, Lefevre[32]. Total neuronal populations were quantified in QuPath[82] with the following parameters: detection[Channel 2]-vGlut1 75, detection[Channel 6]-vGAT 20, doSmoothing TRUE, splitByIntensity TRUE,

splitByShape TRUE, spotSizeMicrons 2, minSpotSizeMicrons 0.15, maxSpotSizeMicrons 6, includeClusters TRUE.

## Software

Statistical analysis was performed using SigmaPlot 11 (Systat, USA), GraphPad Prism 7.05 (GraphPad Software, San Diego, California, USA), and custom scripts written in MATLAB R2015a (MathWorks, USA). Experimental schemes and figures were created in Procreate (5.4) and Adobe Illustrator (v27.9).

## Statistics and reproducibility

An Excel file containing all statistical tests and results is provided as Supplementary Data 1. For each dataset, normality was assessed before selecting the appropriate test. Parametric tests (paired or unpaired two-tailed t-tests, two- or three-way ANOVA) were used when assumptions were met; otherwise, non-parametric tests (Wilcoxon signed-rank, Mann–Whitney U) were applied. For repeated-measures or unbalanced designs, linear mixed-effects models were used. Unless stated otherwise, α was set at 0.05 and post hoc comparisons were corrected for multiple testing. In figures, asterisks (*) indicate significant main effects and hash symbols (#) indicate significant post hoc effects. Fiber-photometry data were analyzed using linear mixed-effects models as described in the section Fiber Photometry of ILC OTR Neurons.

For panels showing representative microscopy images, the following list details the number of repeated experiments with similar results: Fig. 4 B2: $n = 4$ animals, each with 3 sections, a second representative image is shown in Supplementary Fig. S4 A; Figure S4 C: these representative images are part of the dataset utilized for the analysis in Figure S4 D, therefore $n = 4$ animals with 3 sections each; Fig. 5 D1: neurons were biocytin-filled during patching, and one representative cell is shown here, therefore $n = 24$ patched cells, slices obtained from 5 rats; Fig. 5E: $n = 5$ animals, each with 3 sections; Fig. 5 E2: comparable morphological features were found in 12 of 14 OTR$^+$ interneurons filled with biocytin, sections were obtained from 4 rats; Fig. 7 A2: $n = 5$ animals, each with 3 sections; Fig. 7B2: $n = 4$ animals, each with 3 sections; Fig. 7C: same animals as in Fig. 7A2 but with additional ankyrin staining, therefore $n = 5$ animals, each with 3 sections, an additional representative image is shown in Supplementary Fig. S7B.

## Reporting summary

Further information on research design is available in the Nature Portfolio Reporting Summary linked to this article.

## Data availability

We provide two Supplementary Documents with this publication: Supplementary Data 1 includes statistical test results and details for all performed tests, divided on individual Excel sheets per Figure. The statistical tests performed with all plotted data in this study have been deposited in the zenodo database under accession code https://doi.org/10.5281/zenodo.17903734. **Source Data** includes all plotted data in individual Excel sheets. The data generated in this study have been deposited in the figshare database under accession code https://doi.org/10.6084/m9.figshare.30186424. Source data are provided with this paper.

## Code availability

The python script used for analysis of OT1.9-GRAB-sensor fluorescent signal is made available with this publication. It has been deposited in the Zenodo database under accession code https://doi.org/10.5281/zenodo.17878732. Reference: Charlet & Wang (2025). Additionally, the R code for the statistical analysis of fiber photometry data is made available with this publication. It has been deposited in the Zenodo database under accession code https://doi.org/10.5281/zenodo.

17878540. Reference: Sanetra (2025). Any readily available software and packages we utilized are detailed in the method section.

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

## Acknowledgements

This study was inspired by Professor Peter H. Seeburg (1944–2016), who encouraged us to investigate oxytocin signaling in cortical areas. Although its completion was delayed, we hope that our work fulfills his vision. We thank Andreas Draguhn for his input on electrophysiological experiments, Laura Dötsch for performing the Western blot experiment, Selina Wunsch for assisting in the cFos quantification, and Quirin Krabichler for technical assistance with virus injections. This work was supported by a PhD scholarship of the Studienstiftung des Deutschen Volkes to S.S., Humboldt Research Fellowship to A.K., Marie Sklodowska-Curie fellowship (101018877) to A.L., I.K.Y. Scholarship, "Maria Zaousi Memorial" bequest, for postdoctoral research abroad in psychiatry to ARa, DFG Emmy Noether starting grant AL 2466/2-1 to F.A., the National Science Centre, Poland grant UMO-2023/49/B/NZ4/01885 to A.Bla., the FRM Equipe grant EQU202403018071 to A.C., the CNRS International Research Project grant ICOT2023 to AC and VG, the Graduate School of Pain EURIDOL, ANR-17-EURE-0022 to A.C. and C.D., the National Science and Technology Council Postdoctoral Research Abroad Program to K.-Y.W., the Synergy European Research Council (ERC) grant "OxytocINspace" 101071777 to V.G., S.F.B. Consortium 1158-3 to V.G., D.F.G. grant 533293533 to V.G., and German-Israeli Project cooperation (DIP) GR3619-1 to R.H., V.G. and S.W., and ERANET-Neuron grant GR 3619/25-1 to V.G. and S.W.

## Author contributions

Project conception: M.E., V.Grin. Anatomical experiments: M.E., S.S., F.A., A.K., K.A., T.S., F.S., Y.P. Behavioral experiments (optogenetic/chemogenetic): S.S., A.K., A.L., A.Ra, K.A., A.Z., H.S. Ex vivo patch-clamp electrophysiology: J.L., A.Ro, A.K., K.A., A.C., D.H., S.M., H.M. Ex vivo oxytocin sensor: C.D., H.P., V.Grel, K.-Y.W., A.C. In vivo fiber photometry: A.L., R.P., A.S. RNAscope: A.K., A.G., A.T., K.A., A.Bla. Methodology: S.S., A.K., J.S., A.Boe, A.L., R.H., R.T., S.N., S.W., L.G., Y.L., A.C. Manuscript preparation: S.S., A.K., VGrin. Project administration and supervision: M.E. and V.Grin. Funding acquisition: V.Grin.

## Funding

## Competing interests

The authors declare no competing interests.

## Additional information

[1]Department of Neuropeptide Research in Psychiatry, Central Institute of Mental Health, Medical Faculty Mannheim, University of Heidelberg, Mannheim, Germany. [2]German Center for Mental Health (DZPG), partner site Mannheim-Heidelberg-Ulm, Mannheim, Germany. [3]Institut des Sciences Cognitives Marc Jeannerod, CNRS, University of Lyon, Lyon, France. [4]Centre National de la Recherche Scientifique and University of Strasbourg, Institute of Cellular and Integrative Neuroscience, Strasbourg, France. [5]Department of Psychiatry and Psychotherapy, University Medical Center of the Johannes Gutenberg University, Mainz, Germany. [6]Department of Psychiatry and Psychotherapy, Central Institute of Mental Health, Mannheim, Germany. [7]Institute of Physiology and Pathophysiology, University of Heidelberg, Heidelberg, Germany. [8]Laboratory of Neurobiology, Kazan Federal University, Kazan, Russia. [9]Federal Center of Brain Research and Neurotechnologies, Moscow, Russia. [10]Department of Psychology, Laboratory of Behavioral Neuroscience, Faculty of Social Sciences, University of Crete, Crete, Greece. [11]Sagol Department of Neurobiology, the Integrated Brain and Behavior Research Center (IBBRC),

University of Haifa, Haifa, Israel. [12]State Key Laboratory of Membrane Biology, New Cornerstone Science Laboratory, School of Life Sciences, Peking University, Beijing, China. [13]Department of Animal Models in Psychiatry, Central Institute of Mental Health, Clinic Psychiatry and Psychotherapy, Mannheim, Germany. [14]Department of Psychiatry and Psychotherapy, University of Münster, Münster, Germany. [15]Center for Translational Social Neuroscience, Division of Behavioral Neuroscience and Psychiatric Disorders, Emory National Primate Research Center, Emory University, Atlanta, GA, USA. [16]Achucarro Basque Center for Neuroscience, Leioa, Spain. [17]Ikerbasque, Basque Foundation for Science, Bilbao, Spain. [18]Institute of Human Genetics, University Hospital Heidelberg, Heidelberg, Germany. [19]Department of Neurophysiology and Chronobiology, Institute of Zoology and Biomedical Research, Faculty of Biology, Jagiellonian University in Krakow, Krakow, Poland. [20]Department of Psychiatry, School of Medicine & Health Sciences, Carl von Ossietzky University of Oldenburg, Oldenburg, Germany. [21]Department of Neuronal Cell Biology (Center for Brain Research), Medical University of Vienna, Vienna, Austria. [22]Department of Clinical Neurobiology, Medical Faculty of Heidelberg University and DKFZ, Heidelberg, Germany. [23]International Joint Laboratory for Translational Research on Neuromodulation, Shenzhen Institutes of Advanced Technology, Chinese Academy of Sciences, Shenzhen, China. [24]These authors contributed equally: Stephanie Schimmer, Alan Kania, Arthur Lefevre. ✉e-mail: marina.eliava@zi-mannheim.de; valery.grinevich@zi-mannheim.de

