## [Transparent Peer Review file · Nature Communications]

Oxytocin facilitates social behavior of female rats via selective modulation of interneurons in the medial prefrontal cortex

Corresponding Author: Professor Valery Grinevich

Version 0:

Reviewer comments:

Reviewer #1

(Remarks to the Author)

Schimmer et al. present a connectivity diagram in female rats that details the neural circuits involved in social interaction, facilitated by oxytocin (OXT) release from PVN-OXT neurons to OTR-positive interneurons in layer 2/3 of the infralimbic cortex (ILC). The authors further elucidate that these OTR-positive interneurons are predominantly Chandelier cells. These Chandelier cells inhibit OTR-negative pyramidal neurons in layer 2/3 that project to the basolateral amygdala (BLA). The study is particularly exciting because it is the first to demonstrate the involvement of OTR-positive Chandelier cells in social behaviors. A series of in vivo and in slice experiments provide compelling data to support the main claim of the paper. cFos based downstream circuit mapping is a plus. The use of a toy rat as a control is a great idea. Despite these strengths, the impact of this manuscript is diminished by several major concerns outlined below:

Major comments:

- The ILC has been previously implicated in social behaviors (PMID: 32668253, PMID: 32327679, social odor: PMID: 31768051) as well as reward-based motivational behaviors (PMID: 32393535, PMID: 34624366). Additionally, OTR-positive inhibitory neurons in the ILC are already known to play a role in social behaviors (PMID: 25303526). Furthermore, there are existing reports linking ILC to BLA connectivity in social behaviors—ILC neurons projecting to the BLA are preferentially activated by social cues, and inhibition of this circuit impairs social behavior (PMID: 32668253). It is important to discuss these previous findings in detail and compare them with the observations in this manuscript to contextualize the significance of the current results.
- Figure 2C3: The plot is unclear as it currently compares trials from the same mouse during the entire 5-minute lights OFF phase (grey dots) and trials where the light (BL) was ON for 2 minutes followed by an OFF phase. Instead of comparing the entire 5 minutes, it would be more effective to compare the light ON phase (first 2 minutes) with the light OFF phase (last 3 minutes) to allow for clearer comparisons from the same animals. Alternatively, if there was a light OFF phase followed by a light ON phase within the same animal, one could observe a significant increase in social interaction from the timepoint the BL is turned ON.
- Fig 3A1-A6: The RNA in situ quantifications (Fig 3A5) were performed using data from 2 animals, with 2 sections per brain. This sample size seems underpowered for reliable quantification and statistics. I would recommend using at least 3 animals (preferably 4 or more) for more robust comparisons.
- Fig 3C: This is a weak statement, what's the definition of the close proximity here? Any other cell type to compare against? There should be some quantitative analysis to back the statement.
- Figure 3B1-3: The data in Fig 3B1-3 do not agree with the RNA in situ data presented in Fig 3A5. According to the RNA in situ results, 93% of OTR neurons in layer 2/3 are inhibitory, while in layer 5/6, 85% are inhibitory. However, in the OTR-Cre rats, this ratio is not consistent after virus injections. In the OTR-Cre rats, layer 2/3 remains approximately 100% inhibitory, but the images depicted show nearly 100% excitatory OTR neurons in layer 5/6. The methods section mentions that a mixture of two viruses was injected. Please clarify the reason for this inconsistency.
- Figure 4B: The activation of L2/3 ILC OTR interneurons leads to increased social interaction. Additionally, in Figure 5A2, chemo-genetic activation of ILC interneurons shows reduced c-Fos activation in both the ILC and BLA. This suggests that the activation of L2/3 OTR-positive interneurons in the ILC inhibits pyramidal neurons projecting to the BLA, resulting in increased social interaction. Does this imply that the overall inhibition of the ILC-BLA circuit is responsible for driving the increase in social interaction? If so, how does this finding align with the results reported in PMID: 32663438, where an

overall increase in ILC activity (19% of neurons showed increased GCaMP activity, with the majority of cells being non-responsive) was observed?

- Figure 4C, Line 321-328: By looking at the data, CNO stimulation after 24h or food deprivation remove (or nullify) preference toward food rather than shifting its preference toward the social. I would suggest to be more conservative in interpreting the result. Moreover, in C3, how is the bout defined and quantified? It's not clearly stated in the method section.
- Figure 5D: Chandelier cells in layer 2/3 are known for making axo-axonic inhibitory connections. However, the image in Figure 5D appears to show axo-dendritic connections (explained as axo-axonic in line 431-432). It would be valuable to quantify the colocalizations between OTR-positive interneurons and Chandelier cells in layer 2/3 to confirm majority of OTR interneurons are Chandelier cells.
- Figure 5F: The LFP recordings using MEA appear more confusing than informative. How is the LFP result aligned with the rest of the results was not well described. An alternative approach to convincingly demonstrate Oxt-Chandler to ILC excitatory to BLA neurons for social behavioral regulation would be to activate ILC-BLA projection neurons by injecting retro-Chr2 into the BLA, then activate these neurons and compare social interaction levels to a BL OFF control. Additionally, LFP oscillations between the ILC and BLA in the context of social behaviors have been described in PMID: 35580019. How does the activation of OTR-positive Chandelier cells compare with those previously recorded oscillations?
- Figure 1C4, 2D2: Is the social interaction scoring done blindly by at least two different experts? Rigorous quantification methods should be included in the method section.
- Fig5B and C needs quantification; "A substantial number of BLA-projecting neurons was found within the "inhibitory sleeve" in layer 2/3 (Figure 5 B2)". Moreover, actual majority of BLA projecting ILC neurons are in L5/6 rather than L2/3 based on 5B2 and C2. Is it possible that L2/3 Ex neurons communicated with L5/6, then control BLA circuit?

Minor comments:

- Figure 3B2: The OT (IHC) image in blue is difficult to see, particularly due to the poor visibility of fibers and excessive background noise in layer 5/6. Enhancing the clarity of the image with different color and reducing background noise would improve its readability.
- Figure 3C1: If the OT fibers shown are identified through immunohistochemistry (IHC), please indicate this by labeling them as OT (IHC), not mcherry, in the figure.
- Extended data S4 A1: Extended Data S4 A1: Social memory tests performed with saline (control) injections did not show significant differences between novel and familiar rats, although a trend is present. The discussion regarding social memory appears to be based on this non-significant observation. It would be important to address this limitation and consider whether the trend warrants further investigation or if additional controls are needed.

Reviewer #2

(Remarks to the Author)

Schimmer et al is presenting a manuscript describing how oxytocin release in the infra-limbic area of the prefrontal cortex (ILC) facilitates social interactions between female rats. First, they characterize OT projections to the PFC and demonstrate that OT release promote sociability. Then, they leverage fiberphotometry of calcium signaling to show OTR+ neurons are activated during social interactions initiated by the test rat. They also show that chemogenetic activation of OTR+ neurons promote social interaction while CRISPR/Cas9-mediated knockout of OTR has the opposite effect. Then, they demonstrate the OTRs are predominantly expressed by GABAergic neurons in L2/3 of the ILC using in situ hybridization and viral injections. They also show that some L2/3 interneurons and L5 pyramidal neurons are functionally modulated by OTR agonists using patch-clamp recordings. The authors perform chemogenetic manipulation of OTR interneurons and repeat their previous effect on sociability while showing social novelty preference is not affected. Manipulation of OTR+ INs can however facilitate the preference for social interaction over food. Then, they perform paired recordings in slices and show PN receive direct inhibition from OTR+ INs. Finally, they use chemogenetic activation of OTR+ INs coupled with c-fos labelling to test which downstream area is modulated and found that BLA activity is decreased while NAc activity is increased. Viral expression of fluorescent protein, optogenetic and chemogenetic coupled to MEA recording achieve to demonstrate that OTR INs silence L5 PN projecting to the BLA.

The article is well-written, the experiments are thorough and use state-of-the-art techniques to probe the function and mechanism of OT release in the ILC (promoter-specific expression coupled to Cre-dependent viruses, fiberphotometry, paired patch recordings, chemogenetic an optogenetic coupled with slice recording or behaviors). The authors blend these techniques in a very elegant fashion to successfully and convincingly demonstrate their claims. The results are novel and timely as mechanistic studies of the cortical neuromodulatory circuits are still lacking despite knowing for a long-time that neuropeptide and their receptor are widely expressed throughout the cortex.

In only have minor concerns:

- Statistics are sometimes a bit light with the name or p values of the test missing (Fig. S4A2 for example). Systematically provide F values and degrees of freedom for ANOVAs. Perform nested t tests when multiple observations are done in the same mice. Consider putting all the statistic in a separate table rather than in the text. Try to be consistent with the number of significant digits when reporting p values (same degree of precision).
- Line 159: "9.5% of neurons expressed OTRs with a higher occurrence in superficial layers 2/3 compared to the deeper layers 5/6 (12.7% and 8.9%, respectively)". Provide statistical comparison.
- Line 169: " $p < 0.0001$ for anogenital and non-anogenital". What test is used?

- Fig. 2B4. What are the tests used. Give exact p values.

- Line 252: "Similar to our findings with the RNAscope approach, we could identify an OTR+ interneuron population in layer 2/3 (yellow) and an OTR+ pyramidal population in layer 5/6 (red)." Give percentages so one can compare with the ISH results.

- Line 254: "Notably, we could also observe OT fibers throughout the ILC, with denser innervation in layer 2/3". Quantify the density of fibers per ILC layers.

- Line 304 (social memory test). I would prefer that the authors avoid calling this test the social memory test but rather call it social novelty preference test. There's no indication the PFC is needed to store or consolidate memory. Rather, the PFC is likely to utilize social cues from vCA1 to guide the expression of a social memory during recall. In the data presented in Fig. S4A2, comment on the fact that rats did not show a significant preference for the novel rat. Calculate the preference indexes as well and analyze both graphs according to the indications given in the next comment.

- Fig. 4C and Fig. S4B. Chemogenetic activation during food vs. social preference. Please show the interaction times with both corners followed with the preference index (4C2). Data in 4C2 should be analyzed with a two-way ANOVA (injection x feeding status). Also, perform one-sample t tests to compare each group against 0 and determine whether there is a preference for social or feeding. Graphs C3 and C4 should be plotted with social and food next to each other (do the same for the social and food interaction time graph, as you did in Fig. S4A2). Given that interaction with one compartment is not independent from the interaction with the other compartment I would limit the analysis of this graph to comparing social and food within each group, not across group. Calculating the index allows across group comparison by reducing the number of variables to one variable per group.

- Line 314: "where in a satiated state animals are expected to highly prefer social interaction, while food restriction for 24 hours leads to a shift of preference towards food (Extended Data Figure S4 B1-2)." Figure reference is incorrect, it should be Fig. 4C2

- Given the clear effect of OT release on facilitating social interaction with another female, the effect on social vs. food preference appears to be a consequence of facilitating the social interaction rather than choice-specific. Please comment on this briefly in the discussion.

- Fig. 5E1. It should be ChR2 instead of mCherry in the virus construct.

- Fig. 4S2. Check the scale for the layer 2/3 FS OTR+ interneuron recording.

- Fig. S1. Can you show the ratio of time spent in the surround divide by the time spent in the center?

- Fig. S5E. Can you orient the coronal slices with dorsal axis upward? I find the current orientation confusing.

- Line 401. Can you first discard any long-range projections from OTR+ interneurons in the ILC? This way performing the complex c-fos analysis following manipulation of the OTR+ interneurons would make more sense. In addition, explicit what kind of behavior perform the rats prior to perfusing them? A nested t test should be used for Fig. 4A2, give the p value.

- Fig. 5: The treatment to the NAc projections differs quite a lot from the BLA projection. Can you investigate a bit further or comment in the discussion about the putative microcircuit?

Reviewer #3

(Remarks to the Author)

This manuscript by Schimmer et al aims to elucidate the circuit mechanism through which oxytocin modulates PFC circuits regulating social behavior. The authors do a very thorough job tracing the axons of hypothalamic OT neurons into the PFC, genetically labeling the neurons expressing the OT receptor (OTR), and detailing how activation of this receptor can alter the activity of PFC excitatory neuron populations that are thought to control pro-social behaviors. The experiments are conducted to a high standard, and the manuscript is clear and well-written. While the authors do refer to some relevant previous work, they do miss a few important citations that should be mentioned, some of which containing results that the authors' findings should be reconciled with. Much work has been done on PFC circuits, and specifically on the reciprocal interactions between the PFC and BLA. Referring to these important studies can help the reader interpret the results of the current one. This is an important study that will be of great interest to the broader neuroscience audience as it provides a potential mechanism for how OT modulates the cortical aspect of social behavior regulation. I would be happy to hear the authors' response to my specific comments below - all are relatively minor, but can improve the validity of the findings and their interpretation.

Introduction

"With the exception of the auditory cortex in nursing mice, which has been extensively studied by Froemke and colleagues (Marlin, Mitre et al. 2015, Valtcheva and Froemke 2018, Schiavo, Valtcheva et al. 2020), the vast majority of studies demonstrate that the behavioral OT effects are mediated via subcortical brain regions, whereas OT signaling in the cortex is much less explored" - the authors mention specifically the important work of Robert Froemke while overlooking significant

contribution by Nathaniel Heinz (2014, 2016), Oettl et al 2016 and Tan et al 2019, who have also studied OT-mediated cortical modulation of social behavior.

“Despite the inspiring demonstrations that OT action and/or OTR-expressing neurons (OTR+ neurons) in the PFC modulate paternal (He, Young et al. 2019) and sexual behavior (Nakajima, Görlich and Heintz 2014) in voles and mice, respectively, it remains unknown how OT signaling in the PFC affects basic social behavior.” - the term “basic” social behavior is not clearly defined, which may confuse the reader about its distinction from “sociosexual” behavior (Nakajima et al 2014 & Li et al 2016). In this regard, the authors should also relate to Li et al (2016, Cell), which demonstrates how mPFC-projecting OT neurons differentially facilitate social behavior in females compared to males.

The authors might also want to refer to work by Josh Z. Huang detailing the molecular identity of chandelier interneurons in L2 of the PFC and how they innervate amygdala-projecting neurons in mice (see Lu et al., Nat Neurosci 2017, DOI: 10.1038/nn.4624). While the Lu paper failed to demonstrate a behavioral role for these neurons, the current study sheds important light on the topic.

Figure 1:

Panels B1-2 : The demonstration of PVN- and SON-originating OT neurons in the mPFC is compelling. What is the unique contribution of each nucleus to the density presented in B3? This could be achieved by performing the viral-based tracing, which is mentioned in the main text but not presented in Fig1, with two fluorophores within the same animal

Panel B3: Fiber density should be normalized to the area/volume. Although the area/volume used for the current analysis is missing, the results of PrL+Cgt seem lower than those reported in Knobloch et al 2012 (ILC is not reported in that paper). How do you explain these differences?

Number of sections, animals and statistics mentioned in legends are different than what is reported in the Methods section.

Comparing different areas within the same animal can be achieved by averaging each area per rat and use paired t-test (and not unpaired) to compare between areas or by using a linear model with random effect to control for pseudoreplication. Panels C

General comments: To support the main claim of this figure, the authors should provide supplemental schematics or images of the optic fiber targeting the ILC for all animals.

Some explanation is needed for the choice of blue light stimulation protocol (30 Hz for 120 s) for ChR2 activation, given the potential opsin desensitization (Nagel 2003; Lin 2011), the dynamics of large dense core vesicle release and potential depletion from the terminals under such strong stimulation. Have the authors tried recording OT release under this protocol? C4 - for the comparison of total interaction time the statistics should be unpaired t-test, rather than a two-way ANOVA.

Figure 2:

Panels B: The fiber photometry result, while consistent with later results in the paper, might not be specific to OTR+ neurons. I assume that other populations in the PFC would show the same pattern of activity as these neurons. Is this activity pattern consistent with the activity of PVN OT neurons? Does it change when OT release is suppressed in the PFC or more broadly? This should be discussed in the Discussion section if not experimentally addressed.

Panels C1-2: The rationale for using a stimulation protocol in OTR+ neurons in the ILC, is not fully clear until Fig3 D3.

Consider moving this experiment to Fig 3 or reorganize these data to better support the rationale for this experiment.

Panel D: The authors use an interesting approach to study the specific role of OTR by functionally disrupting OTR signalling in the ILC using gene editing. It's important to note that while most of the mutations induced to the OTR by this method result in complete loss of its ligand binding, some residual functionality can not be excluded (e.g. receptor dimerization; Boender et al 2023), as seen in the TGOT experiment (Extended Data Fig. 2). Therefore, this approach does not fall under the definition of “knockout”, and this term should be avoided to prevent misinterpretation.

Figure 3:

General comment: Given the small sample size in this figure (n=2), the conclusions drawn in this part should be taken with a grain of salt. Especially, regarding the distribution and expression of OTR which is considered sex-, age- and state-dependent. Specifically, the identity of OTR+ neurons in the mPFC and their characteristics appear inconsistent with previous literature (Nakajima et al., 2014; Li et al., 2016; Tan et al. 2019). This inconsistency should be addressed at least in the Discussion section, if not with further experiments.

In this regard, the authors don't mention the age of the animals, virginity status in all experiments and the estrous cycle for their non-behavioral experiments, which might be relevant for OTR expression (Bale et al 1995).

B2 - The OT channel (blue) seems much noisier in layer 5/6 compared to layer 2/3, which made it hard to appreciate the heavier OT innervation in layer 2/3. This result should be quantified by comparing OT fiber density in layer 2/3 and layer 5/6. “Histological quantification after viral labelling of OTR+ cell types qualitatively confirmed...” - It was either quantified or qualitatively confirmed, but not both.

Figure 4:

The authors show that OTR+ neurons target L2/3 PYR cells over L5 PYR cells. This is almost expected from the anatomical configuration and the projection patterns of L2/3 OTR+ neurons. Lu et al (Nat Neurosci 2017) demonstrated that within L2, chandelier interneurons preferentially target BLA-projecting over CC-projecting PYR cells. It would be interesting to see whether similar specificity exists for OTR+ interneurons.

C1-4: If I understand correctly, this experiment equally suggests that activation of ILC OTR interneurons in layer 2/3 promotes satiety, which corresponds with the anorexogenic effect of OT. It would be beneficial to include a control experiment to verify that such activation does not alter feeding behavior, supporting a specific effect on sociability. Furthermore, inclusion of a control group in this experiment seems to be important, given the known retroconversion of CNO to clozapine, particularly in rats. There is not much information on how hunger states interact with clozapine or CNO responses, and it would be good to make sure that the effects are indeed specific to the OT manipulation.

The term "cornered social interaction" is not a commonly-used one. Please rephrase.

Figure 5:

General comment: While the suppression of the ILC-BLA pathway by activation of ILC OTR+ interneurons is well demonstrated in-vitro and in-vivo, the lack of its effect on anxiety-related behavior in females is puzzling (Extended Data Figure S1). This finding may require further discussion, especially considering the results from Li et al. (2016), which highlighted sex-specific effects on anxiety-related behavior following activation of OTR+ interneurons in the PFC.

F: Since the authors claim a specific involvement of the ILC-BLA pathway in social interactions, it would be crucial to include an analysis that specifically compares changes in local field potentials during social versus non-social interactions (i.e. interaction with food), rather than presenting the change during the entire "encounter" period. Also, were these changes correlated with the social preference index described in Fig. 4 C2?

General note: when providing the construct names for Cre-dependent viral vectors (e.g. "rAAV-DLX-hm3Dq-GFP or rAAV-DLX-hm4Di-GFPC" as mentioned in Fig. 4) the Cre/lox cassette should be included.

Reviewer #4

(Remarks to the Author)

Version 1:

Reviewer comments:

Reviewer #1

(Remarks to the Author)

Thank you for thoroughly responding all comments. I'm very pleased with the response and updated manuscript. No further comments. I applaud all the effort by authors and congrat on this manuscript with high quality work.

(Remarks on code availability)

Reviewer #2

(Remarks to the Author)

The authors have thoroughly and adequately answered to all my concerns. I fully support the publication of this manuscript.

(Remarks on code availability)

Reviewer #3

(Remarks to the Author)

The authors have addressed all of my comments with additional experiments, data analysis and refinement of statistical tests. I have no further comments on the revised manuscript. Congratulations to the authors on this very important study.

(Remarks on code availability)

Reviewer #4

(Remarks to the Author)

(Remarks on code availability)

ANSWERS TO REVIEWERS' COMMENTS

We thank the reviewers for dedicating their time and effort to evaluate our manuscript. We are grateful for their thoughtful and constructive feedback, which we have addressed point by point below. We hope that our responses will resolve their concerns and support the publication of this study. For ease of reference, all responses and newly added data in the revised manuscript are typed in blue.

Reviewer #1 (Remarks to the Author):

Schimmer et al. present a connectivity diagram in female rats that details the neural circuits involved in social interaction, facilitated by oxytocin (OXT) release from PVN-OXT neurons to OTR-positive interneurons in layer 2/3 of the infralimbic cortex (ILC). The authors further elucidate that these OTR-positive interneurons are predominantly Chandelier cells. These Chandelier cells inhibit OTR-negative pyramidal neurons in layer 2/3 that project to the basolateral amygdala (BLA). The study is particularly exciting because it is the first to demonstrate the involvement of OTR-positive Chandelier cells in social behaviors. A series of in vivo and in slice experiments provide compelling data to support the main claim of the paper. cFos based downstream circuit mapping is a plus. The use of a toy rat as a control is a great idea. Despite these strengths, the impact of this manuscript is diminished by several major concerns outlined below:

Major comments:

- The ILC has been previously implicated in social behaviors (PMID: 32668253, PMID: 32327679, social odor: PMID: 31768051) as well as reward-based motivational behaviors (PMID: 32393535, PMID: 34624366). Additionally, OTR-positive inhibitory neurons in the ILC are already known to play a role in social behaviors (PMID: 25303526). Furthermore, there are existing reports linking ILC to BLA connectivity in social behaviors—ILC neurons projecting to the BLA are preferentially activated by social cues, and inhibition of this circuit impairs social behavior (PMID: 32668253). It is important to discuss these previous findings in detail and compare them with the observations in this manuscript to contextualize the significance of the current results.

We thank the reviewer for this very detailed comment, and have now added new paragraphs about the role of the ILC in social behaviors and reward-based motivational behaviors:

“Population imaging in the mouse mPFC shows that ensembles preferentially encode social olfactory cues and refine with experience (Levy, Tamir et al. 2019). This is consistent with a role for ILC circuits in driving social actions rather than merely reflecting sensory processing.”

“Independent of OT signaling, work in rats shows that inactivation of the ILC broadly impaired both active and inhibitory avoidance and suppressed inappropriate reward seeking (Capuzzo and Floresco 2020). Consistent with this, a comprehensive review concluded that ILC function is highly context-dependent, varying with task demands and the specific neuronal ensembles recruited (Nett and LaLumiere 2021).”

Also, we added a paragraph about the specific ILC-BLA pathway in social behaviors into our discussion, where we reference these papers and discuss our findings in context:

“Notably, a study by Lu, Tucciarone et al. (2017) demonstrated that a subset of layer 2 Chandelier cells in the PrL cortex selectively innervates BLA-projecting principal cells (BLAPCs) over contralateral-cortex-projecting principal cells (CCPCs), while receiving preferential input from CCPCs, thereby establishing a directional inhibitory module that gates BLA-related output. Interestingly, they also found BLAPCs in the same laminar layers as Chandelier cells, while CCPCs were located in deeper layers, resembling our finding of BLA- vs. NAc-projecting neurons. This spatial coincidence suggests that OTR⁺ interneurons are well-positioned to modulate BLAPCs ensembles. While their study did not indicate a behavioral role of this neuronal subset, our work suggests its prosocial role.”

“Importantly, several studies indicate the ILC as a key node for social approach via amygdala-directed outputs. In mice, BLA-projecting ILC neurons are preferentially recruited by social cues, and normal sociability requires intact ILC-BLA signaling (Huang, Zucca et al. 2020). In hamsters, chemogenetic

activation of the ILC-BLA connection during social defeat confers resilience and reduces defeat-associated behavior while lowering BLA c-Fos (Dulka, Bagatelas et al. 2020). Our finding that activating OTR+ interneurons in the rat ILC increases sociability, coupled with reduced BLA cFos expression, aligns with a model in which ILC microcircuits gate amygdala-directed output to favor social approach under challenge. This is further strengthened by the antisocial effects of stimulation of ILC-BLA-projecting pyramidal neurons we observed.”

- Figure 2C3: The plot is unclear as it currently compares trials from the same mouse during the entire 5-minute lights OFF phase (grey dots) and trials where the light (BL) was ON for 2 minutes followed by an OFF phase. Instead of comparing the entire 5 minutes, it would be more effective to compare the light ON phase (first 2 minutes) with the light OFF phase (last 3 minutes) to allow for clearer comparisons from the same animals. Alternatively, if there was a light OFF phase followed by a light ON phase within the same animal, one could observe a significant increase in social interaction from the timepoint the BL is turned ON.

We thank the reviewer for this valid point and agree that the initial representation of the data may have been unclear. To improve clarity and consistency with other parts of our study, we have reanalyzed the data and now present both the overall 5-minute session and the separated sub-behaviors, analogous to our other behavioral experiments (Fig. 2 C3, see on the right), including the optogenetic stimulation of OT fibers shown in Figure 1, which used the same stimulation protocol (Fig. 1 E4, Fig. 2 D2).

We also agree that comparing light ON vs. light OFF phases within the same animal might seem intuitive; however, this comparison would be confounded by the well-documented time-dependent decline in social interest, as observed in our control conditions (gray line in the original graph, see the graph on the right showing the significant decrease in interaction time over a 5min session, this data is from control groups of several optogenetic experiments). This would likely produce an artificial difference unrelated to the stimulation effect. The design we've chosen (within subject light ON and light OFF sessions) might seem suboptimal and we can see its weaknesses, however we made sure to control for habituation effects (part of the rats started with light ON, the rest with light OFF sessions). Moreover, we performed and presented several follow up chemogenetic experiments which yielded congruent results, serving as additional controls.

The idea of reversing the stimulation order (i.e., light OFF followed by light ON) is indeed interesting. However, implementing such a design would not directly address the central question of our study and would require repeating an entire cohort of animals. Given that this additional experimental setup is unlikely to provide critical new insights, and in light of the 3R principles (Replacement, Reduction, and Refinement), we consider such an approach not justified at this stage.

- Fig 3A1-A6: The RNA in situ quantifications (Fig 3A5) were performed using data from 2 animals, with 2 sections per brain. This sample size seems underpowered for reliable quantification and statistics. I would recommend using at least 3 animals (preferably 4 or more) for more robust comparisons.

We fully agree with the reviewer and have extended this experiment with another 3 animals (so 5 animals in total). While the overall picture does not change by this added data, we can now confidently perform reliable quantifications and comparisons. For results see Fig. 3 A4-6 below.

Fig. 2 C3 Animals spend significantly more time socially interacting if the BL is ON in the first two minutes of the session, compared to when it is OFF (n=5 animals). This effect is significantly driven by anogenital and non-anogenital sniffing.

Fig. 3 A4 Colocalization of OTR and Cre mRNA shows high specificity of the transgenic Cre rat line. **A5** The majority of OTR+ neurons in the ILC are GABAergic interneurons, with a small fraction of glutamatergic neurons, which are predominantly found in the layers 5/6. **A6** In-situ-hybridization shows that 1.35% of all neurons in the ILC are OTR+ neurons, with 6.82% of all GABAergic neurons being OTR+.

- Fig 3C: This is a weak statement, what's the definition of the close proximity here? Any other cell type to compare against? There should be some quantitative analysis to back the statement.

We thank the reviewer for this concern, and have accordingly performed a new experiment utilizing semi-automated, high-throughput analysis in IMARIS to quantify the proximity of OT fibers to OTR+ neurons in layer 2/3 and layer 5/6. The results are now depicted in Fig S3 D, and described in the text:

“Notably, OT fibers were found to be significantly closer to OTR+ neurons in layer 2/3 compared to layer 5/6 ($0.04 \pm 0.07 \mu\text{m}$ vs. $0.91 \pm 0.55 \mu\text{m}$, $p < 0.0001$) (Extended Data Figure S3 D).”

Fig. S3 D Automated, high-throughput analysis of the distance of OT fibers to OTR neurons in the ILC. Fibers are closer to neurons in layer 2/3 compared to layer 5/6 (n=24 neurons and n=20 neurons, sections obtained from 4 animals).

- Figure 3B1-3: The data in Fig 3B1-3 do not agree with the RNA in situ data presented in Fig 3A5. According to the RNA in situ results, 93% of OTR neurons in layer 2/3 are inhibitory, while in layer 5/6, 85% are inhibitory. However, in the OTR-Cre rats, this ratio is not consistent after virus injections. In the OTR- Cre rats, layer 2/3 remains approximately 100% inhibitory, but the images depicted show nearly 100% excitatory OTR neurons in layer 5/6. The methods section mentions that a mixture of two viruses was injected. Please clarify the reason for this inconsistency.

We thank the reviewer for pointing out this important inconsistency. To address this concern, we performed a new, semi-automated high-throughput Imaris-based quantitative analysis of rAAV-labeled interneurons and pyramidal neurons in the different layers of the ILC, now presented in Fig. 3 C4, please see below. This updated quantification yields results that more closely match the RNA in situ data: approximately 93% of OTR+ neurons in layer 2/3 and 63% in layer 5/6 are classified as interneurons. We also agree with the reviewer that the microscope image originally shown in Fig. 3B2 was not representative in light of the RNA in situ findings. We have therefore replaced it with a more representative image (see below). Additional images throughout the manuscript (Fig. 3C, Fig. 4A2, Fig.

Fig. 3 C4 Semi-automated, high-throughput analysis of the number of OTR+ neurons identified as interneurons (yellow) or pyramidal neurons (red) in the ILC (n=4 animals, each with 3 sections at three different bregma levels).

Fig. 3 B2 Representative confocal image depicting OTR+ interneurons dominating layers 2/3 (yellow), and individual OTR+ pyramidal neurons localized in layers 5/6 (red), with OT fibers (blue, white arrows) in both subdivisions of the ILC. Scalebar 100µm.

S4C3) further demonstrate the presence of interneurons in layer 5/6.

As noted in the Methods, our labeling strategy used a mixture of two viruses: rAAV-DIO-mCherry, which labels all OTR+ neurons in our OTR-Cre line red, and rAAV-DLX-DIO-GFP, which specifically labels OTR+ interneurons green. Since all GFP+ cells also express mCherry, these interneurons appear yellow in merged images. A likely explanation for the underrepresentation of interneurons in layer 5/6 in the viral labeling data is the reliance on GFP expression to identify interneurons. This approach assumes near-complete infection and expression in all interneurons, which may not be the case. Weak expression or incomplete infection, particularly among certain subtypes (Yazdan-Shahmorad et al., 2024), can lead to underestimation of interneuron numbers and overestimation of excitatory neurons in layer 5/6. Moreover, our earlier RNAscope analysis revealed that vGAT mRNA puncta are more abundant in layer 2/3 (average 9 dots) compared to layer 5/6 (average 4 dots). While both would be counted equally in RNAscope, these differences in mRNA levels may be reflected in lower rAAV-DLX-DIO-GFP expression in layer 5/6, again biasing classification.

In summary, while technical limitations in viral labeling likely account for some discrepancies, our new quantitative analysis using Imaris provides a more accurate and consistent picture of OTR neuron identity across layers, and aligns well with our RNA in situ data.

- Figure 4B: The activation of L2/3 ILC OTR interneurons leads to increased social interaction. Additionally, in Figure 5A2, chemo-genetic activation of ILC interneurons shows reduced c-Fos activation in both the ILC and BLA. This suggests that the activation of L2/3 OTR-positive interneurons in the ILC inhibits pyramidal neurons projecting to the BLA, resulting in increased social interaction. Does this imply that the overall inhibition of the ILC-BLA circuit is responsible for driving the increase in social interaction? If so, how does this finding align with the results reported in PMID: 32663438, where an overall increase in ILC activity (19% of neurons showed increased GCaMP activity, with the majority of cells being non-responsive) was observed?

We agree with the conclusion of the reviewer that the overall inhibition of the ILC-BLA circuit (mediated by OTR+ interneurons) is responsible for driving the increase in social interaction. The paper they refer to however has several differences to our study, and does not contradict our findings: Kingsbury and colleagues found that in mice (not rats), who are interacting with both females and males (not just females), 22% of neurons in the dmPFC (not vmPFC/ILC) respond to social interaction. This response however is both an increase as well as a decrease in activity, with slightly more neurons decreasing in their activity. So overall, while this study focuses on a different subdivision of the PFC and looks at the entire neuron population (not specific to OTR+ neurons), they also find a heterogenous population of neurons modulated by social interaction.

Additionally, we performed chemogenetic stimulation of ILC–BLA projecting pyramidal neurons (Fig. 5F, see below), which led to a decrease in social interaction time. This provides additional evidence supporting our claim that inhibition of the ILC–BLA pathway likely promotes social interaction in female rats.

Fig. 5 F Social interaction after chemogenetic activation of BLA-projecting neurons originating in the ILC. **F1** Injection scheme depicting the injection of a retrogradely Flp-expressing rAAV into the BLA with simultaneous injection of a Flp-dependent (Frt) Gq-mCherry-expressing rAAV into the ILC to chemogenetically activate BLA-projecting ILC neurons after i.p. application of CNO. **F2** Representative injection site showing expression of the rAAV-hSyn-FRT-Gq-mCherry in cell bodies and fibers in the ILC and projecting fibers in the BLA. Scalebar 250µm. **F3** Chemogenetic manipulation of BLA-projecting neurons via CNO injection before a 5 min social interaction (SI) session with an unknown conspecific in the open field. Activation of BLA-projecting neurons decreases social interaction time (n=11 animals).

- Figure 4C, Line 321-328: By looking at the data, CNO stimulation after 24h or food deprivation remove (or nullify) preference toward food rather than shifting its preference toward the social. I would suggest to be more conservative in interpreting the result. Moreover, in C3, how is the bout defined and quantified? It's not clearly stated in the method section.

We thank the reviewer for this comment. Based on another reviewers' comment we have now also repeated the food vs social experiment with more appropriate control groups (to control for CNO off-targets), and are now depicting the results as 'time investigating food/social corner' (Fig. 4 C1, please see below). This direct readout (instead of using the calculated SPI) allows for more intuitive interpretation. Moreover, we performed a dedicated experiment to verify the influence of ILC OTR interneurons chemogenetic activation solely on food interest (Fig S4 D1, see below). In this improved design, both the food vs social experiment as well as food interest experiment clearly show no effect of ILC OTR interneurons activation on interaction with food corner, with clear effects on social corner. Taken together, these results strengthen our interpretation, while we also acknowledge and point out the crucial limitations of the design.

Fig. 4 C1 Chemogenetic activation of OTR+ interneurons increases time spent investigating the social corner (n=8 animals).

Fig. S4 D1 The time spent investigating the food was only influenced by hunger, but not chemogenetic activation of OTR+ interneurons.

We have also added a definition of the number of bouts and general quantification in the method section.

„Corner zones were determined uniformly prior to the experiment, and the position of the nose of the rat within this zone was counted. The parameters quantified were a) the total duration of the nose being within the zone, b) the number of bouts and c) the average length of bouts, with one bout being defined as every new re-entry into the zone after previously exiting it.“

- Figure 5D: Chandelier cells in layer 2/3 are known for making axo-axonic inhibitory connections. However, the image in Figure 5D appears to show axo-dendritic connections (explained as axo-axonic in line 431-432). It would be valuable to quantify the colocalizations between OTR-positive interneurons and Chandelier cells in layer 2/3 to confirm majority of OTR interneurons are Chandelier cells.

We thank the reviewer for this valid concern, and have addressed it with an additional experiment that has now been added to the manuscript: we have performed biocytin fillings of OTR+ interneurons in layer 2/3 with subsequent immunohistochemical staining for ankyrin (an axon-initial segment marker) and therefore could quantify that 86% of OTR+ interneurons showed the morphological features typical for Chandelier cells (Fig. 4 E2, see below).

Even though the original image in our expertise depicted an axo-axonic connection, we have also exchanged the image in Fig. 5D (see below) for one which should not raise further concerns, depicting pyramids projecting to the BLA with axons of OTR+ Chandelier cells.

Fig. 4 E2 Biocytin filling (blue) of OTR+ neurons (green) and staining for ankyrin (red) verifies typical chandelier morphology in 12 of 14 OTR+ interneurons. Scalebar 50µm overview, 25µm zoom.

Fig. 5 D Representative axo-axonic contacts (white arrows) of OTR+ interneurons (red) on BLA-projecting pyramidal neurons (green). The axon-initial segment is stained immunohistochemically for ankyrin (blue). Scalebar 50µm overview, 25µm zoom.

- Figure 5F: The LFP recordings using MEA appear more confusing than informative. How is the LFP result aligned with the rest of the results was not well described. An alternative approach to convincingly demonstrate Otr-Chandler to ILC excitatory to BLA neurons for social behavioral regulation would be to activate ILC-BLA projection neurons by injecting retro-Chr2 into the BLA, then activate these neurons and compare social interaction levels to a BL OFF control. Additionally, LFP oscillations between the ILC and BLA in the context of social behaviors have been described in PMID: 35580019. How does the activation of OTR-positive Chandelier cells compare with those previously recorded oscillations?

We sincerely appreciate this comment by the reviewer, and have decided to exclude this MEA experiment from the manuscript, as it does in fact lead to more confusion than information at its current state. Thorough and prolonged studies will be required to complete this subset of experiments. We feel like it was not crucial for the conclusion and elucidation of the other findings, and will be working on it for a future publication.

However, we agree with the idea of the reviewer that another behavioral experiment can be utilized to show the behavioral effect of manipulating ILC-BLA projecting neurons, and, as mentioned above, have performed it similarly to what the reviewer suggested: we chemogenetically activated BLA-projecting ILC neurons (utilizing a Flp-FRT rAAV system), and could show that this in fact decreased social interaction (Fig 5F, please see below).

Fig. 5 F Social interaction after chemogenetic activation of BLA-projecting neurons originating in the ILC. **F1** Injection scheme depicting the injection of a retrogradely Flp-expressing rAAV into the BLA with simultaneous injection of a Flp-dependent (Frt) Gq-mCherry-expressing rAAV into the ILC to chemogenetically activate BLA-projecting ILC neurons after i.p. application of CNO. **F2** Representative injection site showing expression of the rAAV-hSyn-FRT-Gq-mCherry in cell bodies and fibers in the ILC and projecting fibers in the BLA. Scalebar 250µm. **F3** Chemogenetic manipulation of BLA-projecting neurons via CNO injection before a 5 min social interaction (SI) session with an unknown conspecific in the open field. Activation of BLA-projecting neurons decreases social interaction time (n=11 animals).

- Figure 1C4, 2D2: Is the social interaction scoring done blindly by at least two different experts? Rigorous quantification methods should be included in the method section.

The different behavioral experiments were scored by ARa, SS, and AL. All videos were blinded before analysis. We have added the appropriate information in the method section as:

“Behavioral videos were manually scored by two independent experimenters blinded to experimental conditions. Scorers initially annotated videos separately; discrepancies were reviewed jointly and resolved by consensus to produce a final dataset”

- Fig5B and C needs quantification; “A substantial number of BLA-projecting neurons was found within the “inhibitory sleeve” in layer 2/3 (Figure 5 B2)”. Moreover, actual majority of BLA projecting ILC neurons are in L5/6 rather than L2/3 based on 5B2 and C2. Is it possible that L2/3 Ex neurons communicated with L5/6, then control BLA circuit?

We fully agree with the reviewer and have conducted this quantification now (see Fig S5 C) as below:

Fig. S5 C Quantification of BLA-projecting neurons in layer 2/3 and layer 5/6 of the ILC. **C1** Overview of the ILC with BLA-projecting neurons in red (rAAV-FRT-Gq-mCherry in ILC, retro-rAAV-Flp in BLA) and NeuN staining (blue). Scalebar 250µm. **C2** BLA-projecting neurons in layer 2/3 (white arrows) and layer 5/6 (yellow arrows). Scalebar 100µm. **C3** Number of BLA-projecting neurons in layer 2/3 and layer 5/6 of the ILC.

We could show that 36% of BLA projecting neurons are located in layer 2/3, with the remaining 64% located in layer 5/6. Our main point was the general existence of BLA-projecting neurons in layer 2/3, because the same is not true for NAc-projecting neurons (Fig5 C2). We are convinced that the (smaller) population of BLA-projecting neurons in layer 2/3 would be sufficient to control the ILC-BLA circuit, but it has also been reported previously that Chandelier cells in layer 2/3 innervate pyramidal neurons in cortical layers 2, 3, and 5 (Jiang et al., 2013; Lee et al., 2015) and that while they have mostly vertically oriented axonal segments in layer 2/3, individual long descending axons reached and arborized in layer 6 (Woodruff et al., 2010). These new references have now been included in the discussion.

“It has been reported previously that Chandelier cells in layer 2/3 innervate pyramidal neurons in cortical layers 2, 3, and 5 (Jiang, Wang et al. 2013, Lee, Wang et al. 2015), and that while they have mostly vertically oriented axonal segments in layer 2/3, individual long descending axons reached and arborized in layer 6 (Woodruff, Anderson and Yuste 2010). However, in our experiments, chemogenetic activation of ILC OTR+ interneurons did not alter cFos expression in deep ILC layers, suggesting that their effects are largely confined to layer 2/3.”

Additionally, the reviewer raised an interesting and valid question about L2/3 and L5/6 communication. As we cannot fully exclude such interactions in our case (as they have been reported: Collins et al., 2019; PMID: 29628187) we reanalyzed our cFos experiment (Fig. 5 A1-3) and quantified the cFos expression separately for L2/3 and L5/6 to be able to infer any significant activity changes in deep layers upon OTR interneurons chemogenetic activation. As depicted the number of cFos positive nuclei was significantly altered only in L2/3 suggesting lack of robust activity changes in L5/6 and therefore lack of putative layer interactions in this case.

Fig. 5 A2 Local reduction of cFos expression in the ILC after chemogenetic activation of OTR+ interneurons in the ILC, specific to layer 2/3.

Minor comments:

- Figure 3B2: The OT (IHC) image in blue is difficult to see, particularly due to the poor visibility of fibers and excessive background noise in layer 5/6. Enhancing the clarity of the image with different color and reducing background noise would improve its readability.

We thank the reviewer for this comment and have exchanged the image for one of better quality (Fig3 B2), where OT fibers are clearly visible with minimal background (please see below).

- Figure 3C1: If the OT fibers shown are identified through immunohistochemistry (IHC), please indicate this by labeling them as OT (IHC), not mCherry, in the figure.

We apologize for not stating this clearer, but while (old) Fig 3 B2 does show OT fibers that were identified through immunohistochemistry, the fibers in (old) Fig 3 C1 (now Fig S3 C) were in fact labelled after viral injection with rAAV-OTp-mCherry in the PVN and SON and therefore stained for mCherry (IHC).

In line, we have adapted the figure legend accordingly as:

“OT fibers in close proximity to OTR+ neurons. OT fibers labelled by viral injection of rAAV-OTp-mCherry in the PVN and SON are found...”

- Extended data S4 A1: Extended Data S4 A1: Social memory tests performed with saline (control) injections did not show significant differences between novel and familiar rats, although a trend is present. The discussion regarding social memory appears to be based on this non-significant observation. It would be important to address this limitation and consider whether the trend warrants further investigation or if additional controls are needed.

We thank the reviewer for this valid comment, and have adapted the text accordingly to tone down the interpretation of the novel preference test as:

“Moreover, we conducted a social novelty preference test and observed no difference induced by chemogenetic activation of ILC OTR+ interneurons (corner: $p=0.0100$, treatment: $p=0.1474$, corner x treatment: $p=0.1937$, post-hoc tests all ns) (Extended Data Figure S4C). Taken together, these experiments showed that recruitment of ILC OTR+ interneurons is sufficient to promote social behavior without alterations of social novelty preference or anxiety.”

Reviewer #2 (Remarks to the Author):

Schimmer et al is presenting a manuscript describing how oxytocin release in the infra-limbic area of the prefrontal cortex (ILC) facilitates social interactions between female rats. First, they characterize OT projections to the PFC and demonstrate that OT release promote sociability. Then, they leverage fiberphotometry of calcium signaling to show OTR+ neurons are activated during social interactions initiated by the test rat. They also show that chemogenetic activation of OTR+ neurons promote social interaction while CRISPR/Cas9-mediated knockout of OTR has the opposite effect. Then, they demonstrate the OTRs are predominantly expressed by GABAergic neurons in L2/3 of the ILC using in situ hybridization and viral injections. They also show that some L2/3 interneurons and L5 pyramidal neurons are functionally modulated by OTR agonists using patch-clamp recordings. The authors perform chemogenetic manipulation of OTR interneurons and repeat their previous effect on sociability while showing social novelty preference is not affected. Manipulation of OTR+ INs can however facilitate the preference for social interaction over food. Then, they perform paired recordings in slices and show PN receive direct inhibition from OTR+ INs. Finally, they use chemogenetic activation of OTR+ INs coupled with c-fos labelling to test which

downstream area is modulated and found that BLA activity is decreased while NAc activity is increased. Viral expression of fluorescent protein, optogenetic and chemogenetic coupled to MEA recording achieve to demonstrate that OTR INs silence L5 PN projecting to the BLA.

The article is well-written, the experiments are thorough and use state-of-the-art techniques to probe the function and mechanism of OT release in the ILC (promoter-specific expression coupled to Cre-dependent viruses, fiberphotometry, paired patch recordings, chemogenetic an optogenetic coupled with slice recording or behaviors). The authors blend these techniques in a very elegant fashion to successfully and convincingly demonstrate their claims. The results are novel and timely as mechanistic studies of the cortical neuromodulatory circuits are still lacking despite knowing for a long-time that neuropeptide and their receptor are widely expressed throughout the cortex.

We thank the reviewer for their positive view of our study. We appreciate particularly the comments aimed at improving the quantitative analysis of our anatomical data, which we thoroughly followed and provided requested analyses (details below).

I only have minor concerns:

- Statistics are sometimes a bit light with the name or p values of the test missing (Fig. S4A2 for example). Systematically provide F values and degrees of freedom for ANOVAs. Perform nested t tests when multiple observations are done in the same mice. Consider putting all the statistic in a separate table rather than in the text. Try to be consistent with the number of significant digits when reporting p values (same degree of precision).

We want to thank the reviewer for this important comment, and have now created a Supplemental Document with a table including all statistical tests and corresponding values as suggested. In the text we have also corrected all p-values to have a uniform number of significant digits.

- Line 159: “9.5% of neurons expressed OTRs with a higher occurrence in superficial layers 2/3 compared to the deeper layers 5/6 (12.7% and 8.9%, respectively)”. Provide statistical comparison.

The absolute values of this analysis changed as we performed an additional semi-automated, high-throughput tracking of neurons and increased the n substantially as suggested by reviewer #1, but we have also now performed the suggested statistical comparison and included it into the manuscript and the data table (Fig. S2 A, see on the right).

Fig. S2 A More OTR+ neurons are found in layer 2/3 of the ILC, compared to layer 5/6.

- Line 169: “p<0.0001 for anogenital and non-anogenital”. What test is used?

The fiber photometry data was tested using a linear mixed effect model, all additional details (nesting, fixed factors) can be found in the methods section as:

“Statistical analysis was performed in R (Version 4.0.4; R Core Team, 2021) using RStudio (Version 1.4.1106, PBC; Team R, 2019). Linear mixed-effects models were fitted using the lme4 (Douglas Bates, Bolker and Walker 2015) and lmerTest (Kuznetsova, Brockhoff and Christensen 2017) packages, with type III ANOVA performed using Satterthwaite’s method. Post hoc Tukey’s multiple comparisons were conducted with the emmeans package (Lenith 2023), and the Kenward–Roger approximation for denominator degrees of freedom. Model assumptions were verified by inspection of residual plots. The models compared Z-scored $\Delta F/F$ values before vs. during each behavioral event in a paired manner, with trial included as a random effect nested within session and sessions nested within animal. When recordings were performed bilaterally, hemisphere was added as a cross-nested random effect. Fixed factors included time in relation to trial onset (before vs. during the behavior), and in the case of comparison of active and received behaviors also behavior type. The model for the behavior towards the toy was fitted separately, as these were recorded from separate sessions.”

We have also now added test details in the Supplemental table.

- Fig. 2B4. What are the tests used. Give exact p values.

Please see previous answer, and the Supplemental table.

- Line 252: "Similar to our findings with the RNAscope approach, we could identify an OTR+ interneuron population in layer 2/3 (yellow) and an OTR+ pyramidal population in layer 5/6 (red)." Give percentages so one can compare with the ISH results.

We thank the reviewer for this important comment! The previously mentioned populations were identified only qualitatively, we have now performed a semi-automated, high-throughput analysis of OTR+ neurons in the ILC via Imaris. In the manuscript we have plotted the data further subdivided along the rostro-caudal axis (Fig3 C4), and are mentioning the percentages in the text. On the right we have plotted the same data in analogy to the ISH results for direct and easy comparison. The RNAscope and viral results are comparable, for reasons that might explain the small discrepancies please refer to our answer to Reviewer #1 regarding the same concern (Reviewer #1, Comment 4).

- Line 254: "Notably, we could also observe OT fibers throughout the ILC, with denser innervation in layer 2/3". Quantify the density of fibers per ILC layers.

We appreciate this important comment by the reviewer, and have now performed additional viral injections aimed at addressing this question and others (Reviewer #3, first comment to Figure 1) and injected rAAV-OTp-Venus into the PVN and SON of separate animals to study the anatomical features of the OT input to the ILC. We observed that combined OT innervation of the superficial cortical layers (L1 + 2/3) is actually comparable with the deeper L5/6 (Fig. 1 C3, please see on the right). In our original experiment, we visualized OT fibers with immunohistochemical staining for OT and observed the fibers mostly in the superficial layers. One of the contributing factors might have been the high background staining we observed in the deep cortical layers, and which was pointed out by Reviewer #3. Moreover, layer 5/6 of the ILC has very blurry boundaries with the forceps of corpus callosum. All OT fibers ipsi- and contralaterally projecting to the insula, AON, aPir etc. travel along the white matter and branch in the deep ILC layers. All those thin OT branches became visible in virus-injected brain tissue, though they are not detectable in antibody-stained tissue. Notably, we later describe the population of OTR+ chandelier cells in layer 2/3 driving our observed behavior and downstream circuit, and their apical dendrites are located in layer 1, where we now accurately analyzed this layer and found substantial innervation.

Fig. 1 C3 Localized analysis of fiber innervation originating from the PVN or SON depending on the layer in the ILC (layer 1, layer 2/3 or layer 5/6) reveals most fibers in layer 5/6 (n=6 sections obtained from 3 animals per group).

- Line 304 (social memory test). I would prefer that the authors avoid calling this test the social memory test but rather call it preference test. There's no indication the PFC is needed to store or consolidate memory. Rather, the PFC is likely to utilize social cues from vCA1 to guide the expression of a social memory during recall. In the data presented in Fig. S4A2, comment on the fact that rats did not show a significant preference for the novel rat. Calculate the preference indexes as well and analyze both graphs according to the indications given in the next comment.

We appreciate this comment and have rephrased the 'social memory test' to 'social novelty preference test' throughout the manuscript. We apologize for the confusing description and depiction of statistical tests for this paradigm, as the ANOVA does in fact produce a significant effect on the corner term (novel vs familiar, p=0.0100), however post-hoc tests for the saline and CNO condition only show a trend (and this was reported misleadingly as non-

significant). Most importantly, our main finding of this experiment, which is the lack of effect after chemogenetic manipulation (treatment: $p=0.1474$, corner x treatment: $p=0.1937$) persist. We now present the statistical results clearly in the results section. We also have performed SPI (social preference index) analysis (see on the right). The SPI test also shows no significant difference between the saline and CNO group, our hypothesis that ILC OTR-neuron activation does not modulate social novelty preference remains unaltered.

- Fig. 4C and Fig. S4B. Chemogenetic activation during food vs. social preference. Please show the interaction times with both corners followed with the preference index (4C2). Data in 4C2 should be analyzed with a two-way ANOVA (injection x feeding status). Also, perform one-sample t tests to compare each group against 0 and determine whether there is a preference for social or feeding. Graphs C3 and C4 should be plotted with social and food next to each other (do the same for the social and food interaction time graph, as you did in Fig. S4A2). Given that interaction with one compartment is not independent from the interaction with the other compartment I would limit the analysis of this graph to comparing social and food within each group, not across group. Calculating the index allows across group comparison by reducing the number of variables to one variable per group.

We thank the reviewer for this detailed and constructive comment. This experiment has been scrutinized by all reviewers, who commented on the potential modulation of food interest (reviewer 1 comment to figure 4C), lack of essential controls for food interest and CNO effects (reviewer 3 comment to figure 4C) as well as the detailed analysis pipeline (this comment).

To comprehensively address these points as well as our own concerns, we repeated the experiment with a between-subject design. Instead of administering saline and CNO in the same rats as before, we used two separate animal groups, one injected with the rAAV-DLX-DIO-Gq, and the new group with a control virus rAAV-DLX-DIO-GFP, and administered CNO to both groups. This allowed us to correctly control for any unspecific CNO effects, and investigate properly the potential influence of ILC OTR interneuron modulation on food interest, clearly showing a lack of anorexigenic effects (see figure S4 D1-4 below).

Fig. S4 D Food interest. Animals were placed in an OF with food pellets behind a mesh in one corner. **D1** The time spent investigating the food was only influenced by hunger, but not chemogenetic activation of OTR⁺ interneurons. **D2** Number of bouts while investigating food in the OF. **D3** Average speed in the OF as well as **D4** distance travelled in the OF (n=8 animals per group).

Additionally, the cohort of rats used in this experiment showed much stronger preference towards the social corner, even when hungry, compared to the cohort used in the original experiment. This rendered the SPI to be a less effective parameter to visualize the observed effects of ILC OTR interneuron modulation on the social vs food preference.

Therefore, for clarity and transparency we decided to remove the original experiment from the manuscript, replacing it with the new between subject design. Moreover, we swapped the SPI for an unprocessed data approach: analyzing and depicting raw interaction time with the corners. Because time in the two corners is complementary within a fixed session, corner-wise raw times plus a model with a 'corner' factor provides more information than a single index. The original SPI panels are therefore removed from the main text to avoid redundancy. Rather than a two-way ANOVA on an index, we analyzed raw times using a linear mixed-effects model (fixed effects: *food restriction* [satiated vs 24 h], *virus* [GFP vs Gq], *corner* [social vs food]). This approach (i) respects the repeated-measures structure across corners, (ii) avoids distributional distortions that can arise with ratios/indices, and (iii) allows the specific virus x corner contrast that directly tests whether

hM3Dq activation selectively increases social vs food interaction time. All results and detailed data can be found in the Supplemental Table. In summary the test revealed that:

- corner: $p < 0.0001$ (****) animals spend more time at the social than food corner overall;
- food restriction \times corner: $p = 0.0004$ (***) deprivation changes allocation between corners (expected modulation of preference);
- virus \times corner: $p = 0.0431$ (*) hM3Dq activation increases social relative to food time (selective effect on social engagement). This result is of special importance, as it shows that CNO increases social interest regardless of hunger. The post hoc test specifically points to a significant effect only in the Gq-hungry group. Main effects of food restriction ($p = 0.7023$) and virus ($p = 0.0654$) were not significant, consistent with a selective, corner-specific modulation rather than a global change in exploration.

Regarding the suggestion to plot social and food corner directly next to each other: in the new experiment the social interest visibly and significantly dominates the food interest regardless of hunger. Therefore, the most interesting and crucial comparison is the effect of CNO on social interaction. Hence, we decided to leave the social bar next to each other to emphasize the difference.

- Line 314: "where in a satiated state animals are expected to highly prefer social interaction, while food restriction for 24 hours leads to a shift of preference towards food (Extended Data Figure S4 B1-2)." Figure reference is incorrect; it should be Fig. 4C2

We apologize for the confusing placement of this Figure reference; we have moved it to another point in the manuscript.

- Given the clear effect of OT release on facilitating social interaction with another female, the effect on social vs. food preference appears to be a consequence of facilitating the social interaction rather than choice-specific. Please comment on this briefly in the discussion.

We thank the reviewer for this valid comment and have added this into the discussion as:

"In our social vs. food choice paradigm, activation of OTR⁺ ILC interneurons selectively increased social investigation without generalized arousal, consistent with the proposed role of the ILC in resolving competing motivations by suppressing non-social drives when social cues are behaviorally salient. Intriguingly, the prosocial effect of ILC OTR⁺ interneuron modulation emerges stronger under hunger. This suggests that OT signaling in the ILC acts not necessarily as a tonic enhancer of sociability, but as a context-dependent biasing mechanism that promotes social engagement specifically when motivational conflict is strong."

- Fig. 5E1. It should be Chr2 instead of mCherry in the virus construct.

We thank the reviewer for pointing out this error, which we have corrected.

- Fig. 4S2. Check the scale for the layer 2/3 FS OTR+ interneuron recording.

We want to thank the reviewer for spotting this error in Fig. 4F2. We checked the original recording data and have indeed adjusted the scale to correct the mistake that was previously made during formatting of the figure.

- Fig. S1. Can you show the ratio of time spent in the surround divide by the time spent in the center?

We have added this calculation and added it in Fig. S1 C6 (see on the right).

Fig. S1 C6 The percentage of time spent in the center of the arena remains unchanged

- Fig. S5E. Can you orient the coronal slices with dorsal axis upward? I find the current orientation confusing.

We have omitted this experiment from the publication (for reasons refer to answers point 9 of reviewer #1), therefore the image has been removed.

- Line 401. Can you first discard any long-range projections from OTR+ interneurons in the ILC? This way performing the complex c-fos analysis following manipulation of the OTR+ interneurons would make more sense. In addition, explicit what kind of behavior perform the rats prior to perfusing them? A nested t test should be used for Fig. 4A2, give the p value.

We have never observed long-range projections of OTR+ neurons originating from the ILC (rAAV-DLX-DIO-GFP) in any of the investigated regions (BLA, NAc, VTA, PAG, Sep, Summ), therefore we can confidently assume that any cFos changes in these regions are caused by indirect inhibition of projecting non-OTR neurons. Moreover, we cannot exclude multisynaptic effects.

For the cFos analysis, rats were not subjected to specific behavioral tests, they received i.p. injections of CNO or saline and were placed back into the home-cage into their familiar social environment.

For Fig 5A2, we have further split the analysis in the ILC into the layers and separately analyzed the cFos expression in projecting regions. Therefore, we have performed t-tests and corrected for multiple testing (Sidak). The p values can be found in the Supplemental Table.

- Fig. 5: The treatment to the NAc projections differs quite a lot from the BLA projection. Can you investigate a bit further or comment in the discussion about the putative microcircuit?

We thank the reviewer for pointing out this lack in the manuscript, and have now added more explanation about the projections to the NAc and why we did not investigate them further (see below). To quickly summarize: anatomically, we did not find NAc-projecting neurons in the ILC layer 2/3 (as opposed to the BLA-projecting neurons, we now even quantified this population in Fig S5 C), and initial preliminary patch-clamp experiments (not shown) did show a clear picture of inhibition of BLA-projecting neurons in layer 2/3, which we later verified with more precise optogenetic patch experiments. Therefore, we chose to focus on the BLA-projecting neurons, as we anticipated that they would exert a more direct modulation of social behavior.

“Chemogenetic activation of OTR+ ILC neurons in our experiment led to downregulation of cFos expression in the BLA and increased cFos expression in the NAc. We propose that the observed reduction in the number of cFos-positive cells in the BLA is caused by the ILC OTR+ interneurons-mediated inhibition of ILC principal cells providing excitatory input to the BLA. Among the mPFC subdivisions, the ILC in particular is known to target a dense population of GABAergic interneurons in the BLA (Davis, Zaki et al. 2017) and the BLA in turn densely innervates the NAc (Correia, McGrath et al. 2016). We hypothesize that the observed cFos upregulation in the NAc might be caused by the suppression of BLA interneuron activity, thereby increasing the excitatory signaling from the BLA to the NAc. As mapping circuits via cFos activity patterns comes with the caveat of spatial and temporal ambiguity, we further investigated the OTR-ILC-BLA/NAc circuit both anatomically and electrophysiologically. Retrograde tracing following injections in the BLA and NAc revealed BLA-projecting principal cells in the ILC that were embedded within a dense “inhibitory sleeve” and NAc-projecting principal cells only in deeper cortical layers. Additionally, we showed OTR+ interneurons forming axonal cartridges at the initial axonal segment of neurons projecting to the BLA which was not observed in the case of NAc-projecting principal cells. This was further confirmed by ex vivo studies which showed direct monosynaptic coupling between OTR+ interneurons and BLA-projecting principal cells.”

Reviewer #3 (Remarks to the Author):

This manuscript by Schimmer et al aims to elucidate the circuit mechanism through which oxytocin modulates PFC circuits regulating social behavior. The authors do a very thorough job tracing the axons of hypothalamic OT neurons into the PFC, genetically labeling the neurons expressing the OT receptor (OTR), and detailing how activation of this receptor can alter the activity of PFC excitatory

neuron populations that are thought to control pro-social behaviors. The experiments are conducted to a high standard, and the manuscript is clear and well-written. While the authors do refer to some relevant previous work, they do miss a few important citations that should be mentioned, some of which containing results that the authors' findings should be reconciled with. Much work has been done on PFC circuits, and specifically on the reciprocal interactions between the PFC and BLA. Referring to these important studies can help the reader interpret the results of the current one. This is an important study that will be of great interest to the broader neuroscience audience as it provides a potential mechanism for how OT modulates the cortical aspect of social behavior regulation. I would be happy to hear the authors' response to my specific comments below - all are relatively minor, but can improve the validity of the findings and their interpretation.

Introduction

"With the exception of the auditory cortex in nursing mice, which has been extensively studied by Froemke and colleagues (Marlin, Mitre et al. 2015, Valtcheva and Froemke 2018, Schiavo, Valtcheva et al. 2020), the vast majority of studies demonstrate that the behavioral OT effects are mediated via subcortical brain regions, whereas OT signaling in the cortex is much less explored" - the authors mention specifically the important work of Robert Froemke while overlooking significant contribution by Nathaniel Heinz (2014, 2016), Oettl et al 2016 and Tan et al 2019, who have also studied OT-mediated cortical modulation of social behavior.

We thank the reviewer for pointing out these important studies in the field of cortical OT research. While we did not mention these works at this point in the introduction, we referenced all mentioned studies (except for Oettl et al. 2016) either towards the end of the introduction referring to specific PFC OT action, or in the discussion. We have now added a short paragraph in the introduction to further acknowledge these studies:

"While most studies show that OT's behavioral effects arise from subcortical circuits, emerging work has started to uncover its role in cortical signaling. OT signaling in the murine auditory cortex and its role in nursing, has been extensively studied by Froemke and colleagues (Marlin, Mitre et al. 2015, Valtcheva and Froemke 2018, Schiavo, Valtcheva et al. 2020). Moreover, OT has been shown to drive top-down modulation of early sensory processing in the olfactory bulb to enhance social recognition (Oettl, Ravi et al. 2016). Additionally, several studies have directly implicated cortical OT signaling in the regulation of socio-sexual behaviors: in the medial prefrontal cortex (mPFC), Heintz and colleagues identified OTR-expressing (OTR+) interneurons that gate female sociosexual behavior and revealed a sex-dimorphic OTR-CRH interaction shaping social and anxiety-related responses (Nakajima, Görlich and Heintz 2014, Li, Nakajima et al. 2016). Moreover, OTR-expressing glutamatergic neurons in the mPFC have been shown to selectively regulate social recognition memory (Tan, Singhal et al. 2019)."

Despite the inspiring demonstrations that OT action and/or OTR-expressing neurons (OTR+ neurons) in the PFC modulate paternal (He, Young et al. 2019) and sexual behavior (Nakajima, Görlich and Heintz 2014) in voles and mice, respectively, it remains unknown how OT signaling in the PFC affects basic social behavior." - the term "basic" social behavior is not clearly defined, which may confuse the reader about its distinction from "sociosexual" behavior (Nakajima et al 2014 & Li et al 2016). In this regard, the authors should also relate to Li et al (2016, Cell), which demonstrates how mPFC-projecting OT neurons differentially facilitate social behavior in females compared to males.

We thank the reviewer for pointing out the suboptimal terminology and have changed the wording in the manuscript to affiliative (non-sexual) social behavior / same-sex social interactions / conspecific sociability. We also included the work by Li et al., 2016 in the introduction (see previous answer), and refer to it in more detail in the discussion, as:

"Notably, we did not observe any effects on anxiety behaviors when activating ILC OTR+ interneurons. This is in line with work from Li, Nakajima et al. (2016), who showed that activating OTR-expressing interneurons in the mPFC reduces anxiety-like behavior in males but not females (with OTR loss producing the opposite, male-specific anxiogenic effect). In contrast, the same manipulation enhances social behavior in females but not males, revealing complementary sex-specific routing of OT's influence on anxiety vs. sociability, a sex difference yet unexplored in rats."

The authors might also want to refer to work by Josh Z. Huang detailing the molecular identity of chandelier interneurons in L2 of the PFC and how they innervate amygdala-projecting neurons in mice (see Lu et al., Nat Neurosci 2017, DOI: 10.1038/nn.4624). While the Lu paper failed to demonstrate a behavioral role for these neurons, the current study sheds important light on the topic.

We thank the reviewer for mentioning this study and included it in the manuscript. Considering we only point to the role Chandelier cells play in the ILC over the course of the result section, mentioning this study in the introduction might be confusing for potential readers, but we are dedicating a new paragraph in our discussion to this study as:

“Notably, a study by Lu, Tucciarone et al. (2017) demonstrated that a subset of layer 2 Chandelier cells in the PrL cortex selectively innervates BLA-projecting principal cells (BLAPCs) over contralateral-cortex-projecting principal cells (CCPCs), while receiving preferential input from CCPCs, thereby establishing a directional inhibitory module that gates BLA-related output. Interestingly, they also found BLAPCs in the same laminar layers as Chandelier cells, while CCPCs were located in deeper layers, resembling our finding of BLA- vs. NAc-projecting neurons. This spatial coincidence suggests that OTR⁺ interneurons are well-positioned to modulate BLAPCs ensembles. While their study did not indicate a behavioral role of this neuronal subset, our work suggests its prosocial role.”

Figure 1:

Panels B1-2: The demonstration of PVN- and SON-originating OT neurons in the mPFC is compelling. What is the unique contribution of each nucleus to the density presented in B3? This could be achieved by performing the viral-based tracing, which is mentioned in the main text but not presented in Fig1, with two fluorophores within the same animal.

We thank the reviewer for this excellent comment, and have indeed performed an additional experiment followed by semi-automated quantification of OT fibers originating from the PVN and SON. While we initially performed this experiment exactly as suggested by the reviewer (within subject comparison, rAAV-OTp-Venus in the PVN, rAAV-OTp-mCherry in the SON, and vice versa), we found that expression and detection levels of markers in the two viruses were significantly different. Therefore, we switched to an approach with between subject comparison, where animals received rAAV-OTp-Venus injections either in the PVN or SON. For results see Fig 1C below.

Fig. 1 C Innervation from the PVN and SON. **C1** Injection scheme depicting the injection of a rAAV expressing Venus under the control of the OT promoter (rAAV-OTp-Venus) into either the PVN or SON in separate animals, with respective injection sites. Scalebar 250µm. **C2** Automated analysis of fibers in the ILC quantifying their density by the sum of their volume originating from the PVN or SON reveals stronger innervation from the SON. **C3** Localized analysis of fiber innervation originating from the PVN or SON depending on the layer in the ILC (layer 1, layer 2/3 or layer 5/6) reveals most fibers in layer 5/6 (n=6 sections obtained from 3 animals per group).

Panel B3: Fiber density should be normalized to the area/volume. Although the area/volume used for the current analysis is missing, the results of PrL+Cgt seem lower than those reported in Knobloch et al 2012 (ILC is not reported in that paper). How do you explain these differences?

In line with the reviewer’s suggestion, we have now normalized the fiber density to ‘fibers per mm²’, exactly how we did it in Knobloch et al., 2013. While the data in Knobloch et al., 2013 is conducted in lactating female rats (where we reported that a 3-fold increase in fluorescence intensity is observed in late pregnancy/lactating rats compared to virgin rats), here we used virgin female rats and therefore lower expression and potentially detection rates would be possible. Additionally,

methodological differences might explain slight discrepancies, as in Knobloch's study 15 years ago we used DAB staining of thick brain slices and manual analysis with a *camera lucida*, which is clearly less advanced than the automated Imaris analysis we used now. Nevertheless, we found on average 50 fibers per mm² in the ILC, and while the ILC was not specifically analyzed in the Knobloch's paper, other cortical forebrain structures reported mostly 'up to 10' or 'up to 50' fibers per mm², providing comparable results.

Number of sections, animals and statistics mentioned in legends are different than what is reported in the Methods section.

We thank the reviewer for realizing this error, we have now corrected it. Additionally, we've added a supplementary spreadsheet summarizing all performed statistical comparisons to improve clarity.

Comparing different areas within the same animal can be achieved by averaging each area per rat and use paired t-test (and not unpaired) to compare between areas or by using a linear model with random effect to control for pseudoreplication.

We agree with the reviewer and have now performed the statistics on averaged counting per animal, using a paired t-test (please also see the new Supplemental Table).

Panels C

General comments: To support the main claim of this figure, the authors should provide supplemental schematics or images of the optic fiber targeting the ILC for all animals.

We routinely perform fiber implantations in rats over years (Knobloch et al., *Neuron* (2012), Hasan et al., *Neuron* (2019), Tang et al., *Nature Neuroscience* (2020), Tang et al., *STAR Protocols* (2021), Iwasaki et al., *Nature Communications* (2023) and others) and always verify implantation site post hoc by inspecting brain slices at the light microscope, excluding any animals that were mistargeted. This was also the case for this experiment, and based on the experimental protocols all the included animals were correctly implanted. To be fully transparent, this experiment was performed more than a decade ago, and we failed to retrieve requested specimen as it has inevitably degraded over the years.

To strengthen the claim of this figure we decided to perform a new experiment – utilizing a novel GRAB OT sensor to visualize ex vivo OT release in the ILC upon similar optogenetic stimulation of OT fibers, which was additionally pharmacologically verified by administration of atosiban – a potent OTR (and therefore GRAB OT sensor) antagonist (Fig. 1 D, see below).

Fig. 1 D Axonal OT release in the ILC ex vivo. **D1** Injection scheme depicting the injection of a rAAV expressing the excitatory ChrimsonR under the control of the OT promoter (rAAV-OTp-ChrimsonR-mScarlet) into the hypothalamic nuclei of female rats and injection of a rAAV expressing an OT-sensor (rAAV-OT1.9-GRAB-sensor) in the ILC. Stimulation of the OT fibers in the ILC (625nm light, 30 Hz, 10 ms light on, for 30s). **D2** Peri-stimulus GRAB-OT1.9 fluorescence changes in the ILC in ACSF-only solution (blue) or in the presence of Atosiban (OTR-antagonist, gray). Orange square marks the stimulation period. Bracket indicates the timeframe for area under the curve (AUC) analysis in panel D3. Data presented as mean±sem. **D3** Recordings in ACSF condition showed higher AUC, calculated between 50 and 350 seconds after light stimulation, compared to recordings in the Atosiban condition (n=10 sections).

We hope that, although we have failed to provide requested data, we have satisfactorily addressed the reviewers' concerns with this new, sophisticated experiment.

Some explanation is needed for the choice of blue light stimulation protocol (30 Hz for 120 s) for ChR2 activation, given the potential opsin desensitization (Nagel 2003; Lin 2011), the dynamics of large dense core vesicle release and potential depletion from the terminals under such strong stimulation. Have the authors tried recording OT release under this protocol?

We thank the reviewer for raising this important point. Our intention with the 30 Hz, 10 ms, 120 s train was to robustly engage dense-core vesicle exocytosis from long-range oxytocin (OT) axons. In our study Knobloch et al., 2012 we showed that 30 Hz as well as continuous stimulation with blue light reliably induce action potentials in PVN and SON neurons with our opsin constructs. Consistent with this rationale, Qian et al., 2022 (our co-authored study) used an OT sensor in PFC together with optogenetic activation of PVN OT neurons (20 Hz, 10 ms pulses; up to 10 s) and detected reliable OT release (Fig. 5f,g). Although that protocol is milder than ours, it demonstrates that sustained high-frequency optical stimulation of PVN OT neurons evokes measurable cortical OT release under closely related conditions.

Moreover, as mentioned in our response to the previous comment, we addressed this issue in a new experiment, proving that this stimulation leads to OT release in ILC, detectable with the GRAB OT sensor.

To additionally address potential opsin rundown and the ability of OT neurons to follow high-rate trains, we performed new whole-cell recordings from identified PVN OT neurons expressing ChR2-EYFP or ChrimsonR-mScarlet. 30Hz, 10ms light trains drove robust, time-locked responses in current clamp and sustained photocurrents in voltage clamp at room temperature (Fig S1 A; representative traces attached on the right). These data indicate that our opsins provide stable drive to OT neurons at the frequencies used.

Fig. S1 A Functional verification of OTp-ChR2 and OTp-ChrimsonR through patch-clamp recordings.

When it comes to the concern of potential overstimulation, it is important to note that we use a very strong protocol as established in previous studies (Knobloch et al., 2012) to circumvent potential reduction of stimulation due to tissue light absorption. While causing the depletion of OT vesicles is a valid concern of the reviewer, we want to point out that this is not only unproblematic for this experiment (as we only stimulate the system once), but rather wanted (as we want to release all available OT). Importantly, OT action lasts for several minutes, so following our stimulation, the ILC is plausibly flooded with OT throughout the entire length of the behavioral experiment (5 mins).

C4 - for the comparison of total interaction time the statistics should be unpaired t-test, rather than a two-way ANOVA.

We fully agree with the reviewer and have performed separate statistics for the total interaction time (t-test) and sub-behaviors (2-way-ANOVA). For details, please refer to the new Supplemental Table.

Figure 2:

Panels B: The fiber photometry result, while consistent with later results in the paper, might not be specific to OTR+ neurons. I assume that other populations in the PFC would show the same pattern of activity as these neurons. Is this activity pattern consistent with the activity of PVN OT neurons? Does it change when OT release is suppressed in the PFC or more broadly? This should be discussed in the Discussion section if not experimentally addressed.

To address the reviewer's concern that "other PFC populations might show a similar pattern," we emphasize that our claim is restricted to the genetically defined OTR⁺ population; while other populations could show correlated dynamics at the network level, the signal we measure arises from OTR-Cre-positive neurons only. Unfortunately, at this point we don't have reliably established tools

(e.g. dual color photometry) to properly address this interesting issue, which we believe is beyond the scope of the current study. Regarding consistency with PVN OT neuron activity, our prior work shows that PVN OT neurons increase activity during social touch/interaction (e.g., Tang et al., 2020, Nat Neurosci), and that optogenetic activation of PVN OT neurons produces measurable OT release in the PFC (Qian et al., 2022, Fig. 5f, g). Thus, the direction and timescale of our OTR⁺ PFC population signal are consistent with known PVN OT neuron dynamics and with evoked OT release in the cortex.

Regarding dependence on OT signaling: while we did not manipulate OT signaling during our photometry experiments, we performed a viral OTR knockdown in the ILC (Fig. 2D) and found that it reversed the behavioral effect in our social-interaction assay. Thus, ILC OTR signaling is necessary for the behavioral phenotype measured in the same paradigm, supporting the physiological relevance of the OTR⁺ population activity we report.

Panels C1-2: The rationale for using a stimulation protocol in OTR⁺ neurons in the ILC, is not fully clear until Fig3 D3. Consider moving this experiment to Fig 3 or reorganize these data to better support the rationale for this experiment.

We thank the reviewer for this comment. The rationale of performing this experiment at this point is that in the previous experiment we showed how social interaction activates OTR⁺ neurons (fiber photometry), and we now want to artificially mimic the observed effect of endogenous OT action by optogenetic activation of OTR⁺ neurons, in order to show the direct effect on behavior.

Panel D: The authors use an interesting approach to study the specific role of OTR by functionally disrupting OTR signaling in the ILC using gene editing. It's important to note that while most of the mutations induced to the OTR by this method result in complete loss of its ligand binding, some residual functionality cannot be excluded (e.g. receptor dimerization; Boender et al 2023), as seen in the TGOT experiment (Extended Data Fig. 2). Therefore, this approach does not fall under the definition of "knockout", and this term should be avoided to prevent misinterpretation.

We fully agree with the reviewer and now refer to this manipulation as a CRISPR/Cas9-mediated OTR depletion / functional knockdown, not a knockout. As the reviewer notes, CRISPR editing *in vivo* can yield mosaic indels and incomplete transduction, so residual OTR signaling cannot be excluded. Moreover, ligand-independent/heteromeric effects of OTR (e.g., potential dimerization reported in Boender et al., 2023) could preserve some downstream function even when ligand binding is reduced, consistent with the residual TGOT response we observe (Fig. S2 F). We added a paragraph to the Discussion to state these caveats, and emphasize that our conclusions concern necessity of OTR signaling in ILC rather than complete receptor ablation, as:

"Our in vivo CRISPR/Cas9-mediated OTR targeting should be interpreted as a partial loss-of-function rather than a complete receptor knockout. First, viral transduction and CRISPR editing are inherently mosaic, so a fraction of OTR-expressing neurons likely retain intact receptors. Second, indel heterogeneity may generate alleles with partial activity. Third, residual receptor functionality unrelated to high-affinity ligand binding (for example heteromerization/dimerization) may persist and maintain some downstream signaling (Boender, Boon et al. 2023). These features likely explain the residual responsiveness to TGOT we detected (n=1 neuron, Extended Data Figure S2 F) and indicate that our behavioral effects probably underestimate the full impact of abolishing OTR signaling in the ILC."

Figure 3:

General comment: Given the small sample size in this figure (n=2), the conclusions drawn in this part should be taken with a grain of salt. Especially, regarding the distribution and expression of OTR which is considered sex-, age- and state-dependent. Specifically, the identity of OTR⁺ neurons in the mPFC and their characteristics appear inconsistent with previous literature (Nakajima et al., 2014; Li et al., 2016; Tan et al. 2019). This inconsistency should be addressed at least in the Discussion section, if not with further experiments.

We fully agree with the reviewer and have extended the ISH analysis with additional 3 animals (so total n=5). We are happy to report no major changes in the overall results, but can draw much more reliable conclusions from this data (please see below):

Fig. 3 A4 Colocalization of OTR and Cre mRNA shows high specificity of the transgenic Cre rat line. **A5** The majority of OTR+ neurons in the ILC are GABAergic interneurons, with a small fraction of glutamatergic neurons, which are predominantly found in the layers 5/6. **A6** In-situ-hybridization shows that 1.35% of all neurons in the ILC are OTR+ neurons, with 6.82% of all GABAergic neurons being OTR+.

The previous literature is already quite broad: early work by Nakajima et al., 2014 and Li et al., 2016 reports that OTR+ cells in mPFC are predominantly Sst+ interneurons with strong outputs onto L2/3 and L5 pyramidal neurons; Li also shows CRHBP expression and sex-dependent effects, while Tan (2019) demonstrated a glutamatergic OTR+ subset with distinct circuit/behavioral consequences. These studies also vary in various details from our study:

- Species & line differences: mice in Nakajima 2014; Li 2016, still the overall conclusion of predominantly interneurons stays consistent. Also mice in Tan 2019, it's plausible that rat ILC has fewer glutamatergic OTR+ neurons than mouse PL/mPFC, or that driver-line and reporter sensitivity differ (BAC transgenics/knock-ins can label partly non-overlapping sets)
- Different anatomical compartment (ILC vs "mPFC" as a whole): several mouse papers pooled PL + IL + cingulate or emphasized PL, whereas we targeted the ILC specifically. Laminar and areal composition of OTR cells varies across PFC subfields; a PL-skewed sample could overrepresent a glutamatergic OTR subset that is sparse in ILC.
- Sex & hormonal state: many mouse studies were males or mixed without cycle control. We use specifically females in the diestrus cycle.

We have added an additional section to the discussion to address these differences, as:

"This aligns with mouse mPFC reports emphasizing a predominantly interneuronal OTR population (Nakajima, Görlich and Heintz 2014, Li, Nakajima et al. 2016). By contrast, besides the interneuron population, Tan, Singhal et al. (2019) identified a substantial VGlut1+ OTR+ population in the mouse mPFC. We suspect these differences reflect a combination of species (rat vs. mouse), subregion (ILC vs. broader PFC/PL), sex (female rats), and methodological factors (driver lines, reporter/probe sensitivity, and positivity thresholds). Notably, our OTR-Cre specificity was validated by RNAscope (Cre↔OTR mRNA overlap ~83-93%) and pharmacology (all OTR+ neurons marked with GFP responded to bath application of TGOT in our patch clamp experiments)."

In this regard, the authors don't mention the age of the animals, virginity status in all experiments and the estrous cycle for their non-behavioral experiments, which might be relevant for OTR expression (Bale et al 1995).

We thank the reviewer for noticing that we only mentioned age, virginity status and estrous cycle for behavioral animals. We did in fact use the same age (8-12 weeks) and virginity status (virgin). We have added the respective sentence in the method section.

B2 - The OT channel (blue) seems much noisier in layer 5/6 compared to layer 2/3, which makes it hard to appreciate the heavier OT innervation in layer 2/3. This result should be quantified by comparing OT fiber density in layer 2/3 and layer 5/6.

We thank the reviewer for this comment and have exchanged the image for one of better quality (Fig3 B2), where OT fibers are clearly visible with minimal background as below.

We also have now performed the suggested quantification of fiber density in layer 2/3 and layer 5/6. It can be found in Fig1 C3 (on the right). Please refer to our previous answer to Reviewer #2 (Comment 6) concerning further explanations.

Fig. 3 B2 Representative confocal image depicting OTR+ interneurons dominating layers 2/3 (yellow), and individual OTR+ pyramidal neurons localized in layers 5/6 (red), with OT fibers (blue, white arrows) in both subdivisions of the ILC. Scalebar 100µm.

Fig. 1 C3 Localized analysis of fiber innervation originating from the PVN or SON depending on the layer in the ILC (layer 1, layer 2/3 or layer 5/6) reveals most fibers in layer 5/6 (n=6 sections obtained from 3 animals per group).

“Histological quantification after viral labelling of OTR+ cell types qualitatively confirmed...” - It was either quantified or qualitatively confirmed, but not both.

We thank the reviewer for this point and have corrected the sentence in the manuscript as:

“Histological quantification after viral labelling of OTR+ cell types confirmed that the majority of OTR+ neurons were GAD-expressing interneurons immunopositive for parvalbumin and/or calbindin (Extended Data Figure S3 B).”

Figure 4:

The authors show that OTR+ neurons target L2/3 PYR cells over L5 PYR cells. This is almost expected from the anatomical configuration and the projection patterns of L2/3 OTR+ neurons. Lu et al (Nat Neurosci 2017) demonstrated that within L2, chandelier interneurons preferentially target BLA-projecting over CC-projecting PYR cells. It would be interesting to see whether similar specificity exists for OTR+ interneurons.

We thank the reviewer for the insightful suggestion! Lu et al. (2017) showed that in the prelimbic cortex, layer-2 Chandelier cells (ChCs) preferentially innervate BLA-projecting pyramidal neurons (BLAPCs) over contralateral cortex-projecting neurons (CCPCs) within L2, thereby exerting directional inhibitory control over BLAPC ensembles.

Our study in the ILC was not designed to resolve the projection identity of our L2/3 recordings, as we did not combine OTR+ cell activation with dual retrograde labeling. Consequently, we cannot directly test whether OTR+ interneurons display the same BLAPC-over-CCPC specificity within L2/3. We agree that this is clearly an interesting question, but does not lie within the scope of our study. Nevertheless, we have added the work of Lu and colleagues into our discussion:

“Notably, a study by Lu, Tucciarone et al. (2017) demonstrated that a subset of layer 2 Chandelier cells in the PrL cortex selectively innervates BLA-projecting principal cells (BLAPCs) over contralateral-cortex-projecting principal cells (CCPCs), while receiving preferential input from CCPCs, thereby establishing a directional inhibitory module that gates BLA-related output. Interestingly, they also found BLAPCs in the same laminar layers as Chandelier cells, while CCPCs were located in deeper layers, resembling our finding of BLA- vs. NAc-projecting neurons. This spatial coincidence suggests that OTR+ interneurons are well-positioned to modulate BLAPCs ensembles. While their study did not indicate a behavioral role of this neuronal subset, our work suggests its prosocial role.”

C1-4: If I understand correctly, this experiment equally suggests that activation of ILC OTR interneurons in layer 2/3 promotes satiety, which corresponds with the anorexigenic effect of OT. It would be beneficial to include a control experiment to verify that such activation does not alter feeding behavior, supporting a specific effect on sociability. Furthermore, inclusion of a control group in this experiment seems to be important, given the known retroconversion of CNO to clozapine, particularly in rats. There is not much information on how hunger states interact with clozapine or CNO responses, and it would be good to make sure that the effects are indeed specific to the OT manipulation.

We fully agree with these comments and have decided to repeat this experiment employing a between subject design. Instead of administering saline and CNO in the same rats as before, we used two separate animal groups, one injected with the rAAV-DLX-DIO-Gq, and the new group with a control virus rAAV-DLX-DIO-GFP, and administered CNO to both groups. This allowed us to correctly control for any unspecific CNO effects, and investigate properly the potential influence of ILC OTR interneuron modulation on food interest, clearly showing a lack of anorexigenic effects (see figure S4 D1-4). For detailed discussion on statistical tests used please refer to our answer to reviewer #2 and the Supplemental Table containing all statistics.

Fig. S4 D Food interest. Animals were placed in an OF with food pellets behind a mesh in one corner. **D1** The time spent investigating the food was only influenced by hunger, but not chemogenetic activation of OTR⁺ interneurons. **D2** Number of bouts while investigating food in the OF. **D3** Average speed in the OF as well as **D4** distance travelled in the OF (n=8 animals per group).

The term "cornered social interaction" is not a commonly-used one. Please rephrase.

We thank the reviewer for this comment and refrain from using the term in the manuscript.

Figure 5:

General comment: While the suppression of the ILC-BLA pathway by activation of ILC OTR⁺ interneurons is well demonstrated in-vitro and in-vivo, the lack of its effect on anxiety-related behavior in females is puzzling (Extended Data Figure S1). This finding may require further discussion, especially considering the results from Li et al. (2016), which highlighted sex-specific effects on anxiety-related behavior following activation of OTR⁺ interneurons in the PFC.

We thank the reviewer for this interesting point. Li et al. (2016) actually found a very interesting sexual dimorphism: when activating OxtrINs in the mPFC (prelimbic) it reduced anxiety-like behavior (↑ open-arm time in EPM; ↑ center time in OF) without changing sociability in males; in females it actually increased sociability without affecting anxiety. Even though the species and subdivision of the PFC differs from our study, we have included this very interesting finding in our discussion:

“Notably, we did not observe any effects on anxiety behaviors when activating ILC OTR⁺ interneurons. This is in line with work from Li, Nakajima et al. (2016), who showed that activating OTR-expressing interneurons in the mPFC reduces anxiety-like behavior in males but not females (with OTR loss producing the opposite, male-specific anxiogenic effect). In contrast, the same manipulation enhances social behavior in females but not males, revealing complementary sex-specific routing of OT’s influence on anxiety vs. sociability, a sex difference yet unexplored in rats.”

Nevertheless, we wanted to further investigate any anxiety-related effects of OTR+ interneuron activation in the ILC, and performed additional chemogenetic experiments (activation of OTR+ interneurons) in the open field and elevated zero maze. In line with our previous observations, we did not find any significant effects on center time in the open field or time or entries to the open arm in the EZM. For the full results see Fig. S4 A and B (as below).

Fig. S4 A Exploratory behavior in the open field (OF) after chemogenetic activation of OTR+ interneurons in the ILC. **A1** Distance travelled in the OF, **A2** Time in the center of the OF, **A3** number of entries into the OF, and **A4** average speed in the OF (n=8 animals). **B** Exploratory behavior in the elevated zero maze (EZM). **B1** Distance travelled in the EZM, **B2** Time spent in the open parts of the EZM, **B3** number of entries into the open parts of the EZM, and **B4** average speed in the EZM (n=8 animals).

F: Since the authors claim a specific involvement of the ILC-BLA pathway in social interactions, it would be crucial to include an analysis that specifically compares changes in local field potentials during social versus non-social interactions (i.e. interaction with food), rather than presenting the change during the entire “encounter” period. Also, were these changes correlated with the social preference index described in Fig. 4 C2?

We sincerely appreciate this comment by the reviewer, and have decided (also based on comments by the other reviewers) to exclude this MEA experiment from the manuscript, as it does lead to more confusion than information at its current state. Thorough and prolonged studies will be required to complete this subset of experiments. After thorough deliberation we concluded it was not crucial for the conclusion and elucidation of the other findings, and will be working on it for a future publication.

Instead, we have performed a new experiment specifically manipulating the BLA-projecting neurons in a social interaction experiment (Fig. 5 F, please see below), which shows the suppressing effect on social interactions after activating these neurons.

Fig. 5 F Social interaction after chemogenetic activation of BLA-projecting neurons originating in the ILC. **F1** Injection scheme depicting the injection of a retrogradely Flp-expressing rAAV into the BLA with simultaneous injection of a Flp-dependent (Frt) Gq-mCherry-expressing rAAV into the ILC to chemogenetically activate BLA-projecting ILC neurons after i.p. application of CNO. **F2** Representative injection site showing expression of the rAAV-hSyn-FRT-Gq-mCherry in cell bodies and fibers in the ILC and projecting fibers in the BLA. Scalebar 250µm. **F3** Chemogenetic manipulation of BLA-projecting neurons via CNO injection before a 5 min social interaction (SI) session with an unknown conspecific in the open field. Activation of BLA-projecting neurons decreases social interaction time (n=11 animals).

General note: when providing the names for Cre-dependent viral vectors (e.g. "rAAV-DLX-hm3Dq-GFP or rAAV-DLX-hm4Di-GFPC" as mentioned in Fig. 4) the Cre/lox cassette should be included.

We thank the reviewer for pointing out this error, and have corrected it in the manuscript.

Reviewer #4 (Remarks to the Author):

We thank the reviewer for their work on our manuscript.

ANSWERS TO REVIEWERS' COMMENTS

We thank the reviewers for dedicating their time and effort to evaluate the revision of our manuscript. For ease of reference, all responses are typed in blue.

Reviewer #1 (Remarks to the Author):

Thank you for thoroughly responding all comments. I'm very pleased with the response and updated manuscript. No further comments. I applaud all the effort by authors and congrat on this manuscript with high quality work.

We thank the reviewer for this very supportive response, and want to thank them again for their feedback that helped us improve the manuscript.

Reviewer #2 (Remarks to the Author):

The authors have thoroughly and adequately answered to all my concerns. I fully support the publication of this manuscript.

We thank the reviewer for their support, and want to thank them again for their feedback that helped us improve the manuscript.

Reviewer #3 (Remarks to the Author):

The authors have addressed all of my comments with additional experiments, data analysis and refinement of statistical tests. I have no further comments on the revised manuscript. Congratulations to the authors on this very important study.

We thank the reviewer for their supportive response, and want to thank them again for their feedback that helped us improve the manuscript.

Reviewer #4 (Remarks to the Author):

We thank the reviewer for their work on our manuscript.